# Predicting proximal tubule failed repair drivers through regularized regression analysis of single cell multiomic sequencing

Nicolas Ledru [1], Parker C. Wilson [2], Yoshiharu Muto [1], Yasuhiro Yoshimura [1], Haojia Wu [1], Dian Li [1], Amish Asthana [3], Stefan G. Tullius [4], Sushrut S. Waikar [5], Giuseppe Orlando[3] & Benjamin D. Humphreys [1,6] ✉

Renal proximal tubule epithelial cells have considerable intrinsic repair capacity following injury. However, a fraction of injured proximal tubule cells fails to undergo normal repair and assumes a proinflammatory and profibrotic phenotype that may promote fibrosis and chronic kidney disease. The healthy to failed repair change is marked by cell state-specific transcriptomic and epigenomic changes. Single nucleus joint RNA- and ATAC-seq sequencing offers an opportunity to study the gene regulatory networks underpinning these changes in order to identify key regulatory drivers. We develop a regularized regression approach to construct genome-wide parametric gene regulatory networks using multiomic datasets. We generate a single nucleus multiomic dataset from seven adult human kidney samples and apply our method to study drivers of a failed injury response associated with kidney disease. We demonstrate that our approach is a highly effective tool for predicting key *cis*- and *trans*-regulatory elements underpinning the healthy to failed repair transition and use it to identify NFAT5 as a driver of the maladaptive proximal tubule state.

Chronic kidney disease (CKD) exacts an immense medical and financial burden, affecting 37 million people in the United States alone[1]. There are limited options for managing the progression from CKD to end-stage renal disease (ESRD), leading to an annual national incidence of 125,000 ESRD cases[1]. A better molecular understanding of kidney pathology could lead to the development of new treatments.

Recent work has increasingly identified different proximal tubule states as key players in the progression of kidney disease across acute and chronic etiologies[2–5]. In acute kidney injury (AKI), proximal tubule cells adopt a dedifferentiated and proliferative phenotype, providing the kidney with some regenerative capacity post-injury[6]. These cells are marked by phosphatidylserine receptor KIM1 (HAVCR1) expression, which binds apoptotic cell fragments to clear debris from the tubular lumen[7]. During the repair process, a small proportion of injured PT cells fail to undergo full repair and restoration of a healthy PT phenotype[2,3,6,9]. These so-called failed repair PT (FR-PT) cells take on a proinflammatory, profibrotic, senescent-associated secretory pathway (SASP) phenotype, thus providing a possible mechanism for AKI to CKD transition[2,3,9].

[1]Division of Nephrology, Department of Medicine, Washington University in St. Louis School of Medicine, St. Louis, MO, USA. [2]Division of Anatomic and Molecular Pathology, Department of Pathology and Immunology, Washington University in St. Louis, St. Louis, MO, USA. [3]Department of Surgery, Wake Forest Baptist Medical Center; Wake Forest Institute for Regenerative Medicine, Wake Forest School of Medicine, Winston Salem, NC, USA. [4]Division of Transplant Surgery and Transplant Surgery Research Laboratory, Department of Surgery, Brigham and Women's Hospital, Harvard Medical School, Boston, MA, USA. [5]Section of Nephrology, Department of Medicine, Boston University Chobanian and Avedisian School of Medicine, Boston Medical Center, Boston, MA, USA. [6]Department of Developmental Biology, Washington University in St. Louis School of Medicine, St. Louis, MO, USA. ✉e-mail: humphreysbd@wustl.edu

 1

A similar cell state is present in control, non-AKI, human kidney[10]. We called one of these cell states PT_VCAM1 based on its expression of VCAM1. PT_VCAM1 resembles the failed repair proximal tubule state that arises following AKI. The PT_VCAM1 population appears to increase with age and in diabetic kidney disease[3,10,11]. These findings raise the possibility that subacute injuries incurred over a patient's life could lead to a gradual accumulation of FR-PT cells, resulting in a progressive increase in proinflammatory and profibrotic stimuli that drives CKD progression. We hypothesize that FR-PT and PT_VCAM1 are similar cell states that fail to repair following sustained PT injury.

Transcriptional control is coordinated through the binding of *trans*-acting transcription factors (TFs) to DNA motifs present in short, nucleosome-free genomic *cis*-regulatory elements (CREs)[12]. These CREs allow the binding of different sets of transcription factors and subsequent recruitment of co-activators and transcription complex factors through dynamic regulatory activity and looping interactions with target gene promoters[13–15]. Misregulation of gene transcription has the potential to drive disease[12,16–18], so we also hypothesize that identifying disease-associated CREs and their gene targets, as well as the key set of transcription factors coordinating this regulatory action, will improve our understanding to identify therapeutic targets that limit kidney function decline by inhibiting or reversing transition to the FR-PT or PT_VCAM1 state.

Single cell sequencing has been successfully applied to characterize the transcriptional changes associated with specific cell types and cell states in a variety of human kidney diseases[2–4,11,19–22]. Several current challenges limit our ability to leverage these datasets for the identification of regulatory factors coordinating gene expression changes. For example, it is difficult to prioritize the handful of transcription factors responsible for a particular cell process among the hundreds of expressed TFs and cognate motifs that can be detected[23]. Single cell multiomic sequencing generates joint RNA- and ATAC-sequencing data from the same cell. In particular, it provides the ability to couple distal *cis*-regulatory element (CRE) activity to gene transcription[24]. We hypothesized that this could be used to identify the regulators of injury and repair responses driving the disease-associated FR-PT state, where a PT cell-specific healthy-to-FR trajectory with associated transcriptional changes is known.

We generated single nucleus multiomic (simultaneous RNA-seq and ATAC-seq) datasets from healthy human adult kidney samples. Although FR-PT arise after acute injury, the transcriptionally similar PT_VCAM1 state is present even in healthy kidneys and we hypothesize it represents a wear and tear or injury in situ state, even in the absence of clinical AKI. We developed a computational tool, RENIN (Regulatory Network Inference), to construct genome-wide parametric gene regulatory networks that generate predicted weights for both CREs and TFs to rank and prioritize the most important regulatory elements. We then applied RENIN to identify regulators of the healthy-PT_VCAM1 transition that may drive fibrosis and inflammation, thereby increasing risk for CKD. Several excellent tools have been developed to accomplish similar tasks[24–29], however we chose to tailor our method to prioritize ranking of regulatory elements for further investigation by applying a parametric adaptive elastic-net estimator for gene regulatory network model training to increase model sparsity and thus improve critical regulatory element prioritization[30]. We then used motif data to identify transcription factors binding to these putative CREs and introduced a second step to correlate expression of predicted binding transcription factors (TFs) to expression of H-FR genes in the proximal tubule population. RENIN therefore simultaneously predicts CRE- and TF-gene regulatory interactions from single cell multiomic datasets to predict detailed genome-wide gene regulatory networks.

Using RENIN, we found evidence supporting the similarity in phenotype and function of the FR-PT and PT_VCAM1 states and the hypothesis that FR-PT formation contributes to CKD risk with partitioning heritability analysis. We also predicted key drivers of the H-FR transition, including protective TFs such as PPARA and pro-FR TFs including NFAT5, HIVEP2, and CREB5. We validated these TFs with siRNA knockdown and CUT&RUN experiments that supported the accuracy of our modeling predictions. We also found that our approach enriched for and predicted more regulatory connections than other tested methods. These findings demonstrate that the FR-PT state is present in control, non-AKI adult human kidneys and provide additional evidence that the FR-PT regulatory elements are associated with increased CKD risk. We also have identified several drivers of the failed repair state that may be targeted therapeutically to treat CKD.

## Results

### Simultaneous single-nucleus transcriptional and chromatin accessibility profiling of the adult human kidney resolves high-quality cell type-specific profiles

We performed single nucleus multiomic sequencing on seven control adult kidney samples. Five samples were nephrectomy-derived, and two samples were back-bench biopsies taken from pre-transplant healthy kidneys. Patient samples were heterogenous in age, sex, and race: median 64 years old (range: 45-76) 2 male and 5 female, and 3 Black and 4 White. Five of these patients had normal kidney function and two of them had advanced chronic kidney disease (Source Data). Histologic scoring of interstitial fibrosis and tubular atrophy (IFTA) was 1-10% for all but one sample which had 10-20% IFTA in the setting of a serum creatinine of 4.73 mg/dL. Nuclei isolation was performed for each sample one at a time, followed by library generation and sequencing. Expression profiles were corrected to remove ambient RNA contamination with CellBender, predicted doublets were removed, then batch correction was performed with Harmony on both the split RNA-seq and ATAC-seq portions of the dataset[31–34]. A total of 50,768 nuclei were annotated with cell types following quality control filtering. Among these nuclei, 29,758 genes and 193,787 accessible chromatin regions were identified. 150,237, or 77.5%, of peaks overlapped with a previously published snATAC-seq dataset[10], confirming quality (Supplementary Fig. 1V). Overall, correlation between snRNA-seq expression and snATAC-seq gene activity was 0.552 (p < 2e-16), but was higher between shared variable features in the RNA and ATAC datasets (r = 0.749, p < 2e-16; Supplementary Fig. 1b). To compare the value added by single cell multiome versus performing snRNA-seq and snATAC-seq on separate samples, we tested Seurat's cell transfer annotation method[35] on our multiome data as though it was generated separately. Performance was high for certain cell types, such as the loop of Henle (LH), but low for other cell types, including podocytes (POD) and mesangial cells (MES). For example, 93.8% of the mesangial cells were predicted to be endothelial cells (Supplementary Fig. 1c). These findings suggest multiome sequencing improves the ability to resolve cell types and states without sacrificing gene and peak detection sensitivity.

We performed weighted nearest neighbor clustering, an approach that learns relative cell type specific mRNAs and chromatin peaks to define cell types and states in the dataset[36]. All major renal cortex cell types described in previous studies were successfully identified by annotating clusters based on snRNA-seq marker expression: proximal convoluted tubule (PCT), proximal straight tubule (PST), *KIM1*+ proximal tubule (KIM1 + PT), parietal epithelial cells (PEC), loop of Henle (LH), distal tubule and collecting tubule (DCT, DCT/CNT, CNT), collecting duct (PC, ICA, ICB), podocytes (POD), endothelium (ENDO), mesangium (MES), fibroblasts (FIB), and immune cells (Immune) (Fig. 1a). Notably, our multiomic clustering allowed separate annotation of PCT and PST clusters, which had been previously difficult to resolve using snRNA-seq alone[10]. Cell type specific marker expression and gene activity, measured by aggregating accessible peaks within the gene body and promoter, were broadly similar between datasets (Fig. 1b).

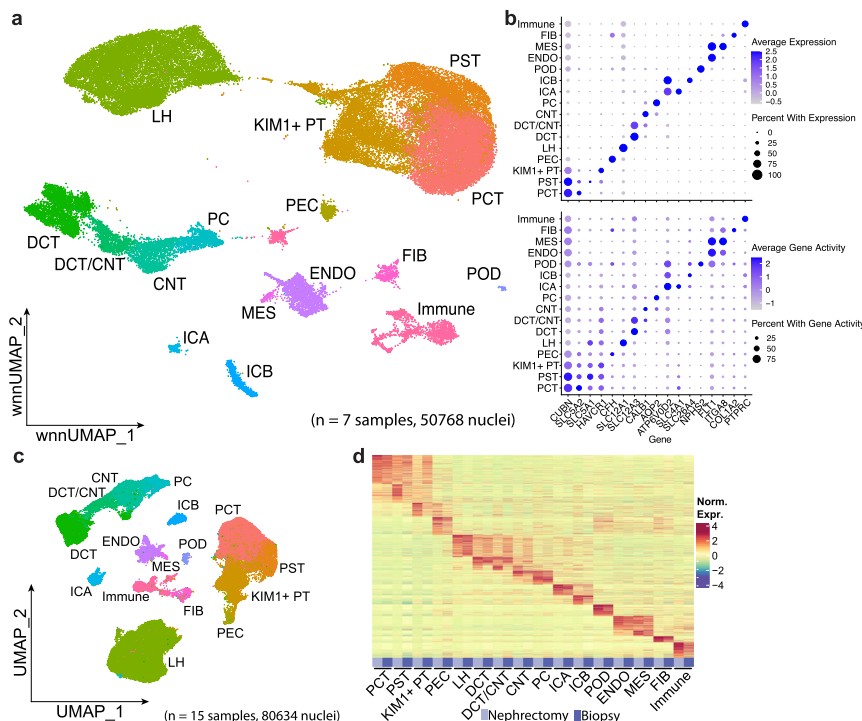

**Fig. 1 | Simultaneous single nucleus multiomic RNA-seq and snATAC-seq of adult human kidney. a** WNN UMAP plot of multiome dataset prepared from 7 samples and totaling 50,768 nuclei. PST, proximal straight tubule; PCT, proximal convoluted tubule; KIM1 + PT, *KIM1*-expressing injured/failed repair proximal tubule, PEC parietal epithelial cell, LH loop of Henle, DCT distal convoluted tubule, CNT connecting tubule, PC principal cell, ICA intercalated alpha, ICB intercalated beta, POD podocyte, ENDO endothelial, MES mesangial, FIB fibroblast. **b** Above, RNA expression of cell type markers by cell type; below, gene activity, of cell type markers by cell type. Gene activity calculated by aggregating promoter and gene body peaks in snATAC-seq dataset. **c** UMAP plot of aggregate snRNA-seq dataset generated from a total of 15 samples (5 from living donor biopsies from 3 individual donors and 10 from nephrectomy tissue), containing 80,634 nuclei. **d** Heatmap of cell type marker expression for each cell type by sample type—nephrectomy or biopsy.

## Doublet calling algorithms identify non-overlapping putative doublets

Droplet-based microfluidic methods commonly result in multiplets when more than one nucleus is encapsulated within the same droplet[32]. When this happens, nucleotide fragments from multiple nuclei are tagged with the same barcode and thus appear as the same individual nucleus in downstream clustering, cell type annotation, and analysis. Several bioinformatic approaches have been developed to identify which barcodes are multiplets, including DoubletFinder, AMULET, and ArchR[32,33,37]. DoubletFinder, using scRNA-seq data, and ArchR, using scATAC-seq data, both simulate artificial doublets by averaging profiles from pairs of randomly selected barcodes. Barcodes are identified as doublets, rather than single nuclei, by a high proportion of artificial doublets to dataset barcodes in neighboring points in a low dimensional projection of the dataset. Unlike DoubletFinder and ArchR, AMULET's algorithm takes advantage of the fact that each individual nucleus contains only two copies of each genomic locus. In the absence of copy number variation, the maximum number of reads mapping to a given region should be two in a scATAC-seq dataset. AMULET determines the number of regions with greater than two overlapping fragments and identifies as multiplets barcodes with higher-than-expected numbers of these regions (Supplementary Fig. 2a). It is difficult and costly to test these approaches with biological ground-truth datasets, so simulated datasets have been used to compare performance of doublet identification methods[38]. We reasoned that simultaneous multimodal single nucleus sequencing provides a unique opportunity to cross-assess doublet calling algorithms using data from different modalities. We performed doublet calling with AMULET and generated artificial doublets from the same pairs of starting nuclei, using either the RNA modality for DoubletFinder or the ATAC modality for ArchR. Unexpectedly, while overlap between the three tested methods was greater than random chance, it was still very low (<20%), consistent with previously reported ArchR-AMULET differences (Supplementary Fig. 2b)[33]. AMULET-derived doublets averaged high ATAC unique molecular identifier (UMI) counts, concordant with the algorithm's approach, whereas ArchR-doublets averaged low ATAC UMI counts (Supplementary Fig. 2c). Unlike ArchR or DoubletFinder, AMULET can call homotypic doublets (Supplementary Fig. 2d), so we removed AMULET-predicted doublets. Furthermore, when comparing doublets detected by DoubletFinder vs. ArchR, doublets predicted by ArchR were evenly spread throughout all clusters whereas those predicted by DoubletFinder were enriched between and on the edges of clusters, where we would expect them to be located (Supplementary Fig. 2d). For this reason, we additionally removed DoubletFinder-predicted doublets.

## Partial nephrectomy kidney samples are similar to live donor samples and both contain FR-PT cells

Failed repair proximal tubule cells (FR-PTC) have been identified in healthy human kidney samples derived from tumor-adjacent tissue after partial nephrectomy[10,11]. We hypothesize that FR-PTC in apparently healthy kidneys is attributable to subacute injury or stress sustained over a patient's lifetime, leading to accumulation of FR-PTC cells and a fibrosis-promoting phenotype. An alternative hypothesis is that tumor mass effect in partial nephrectomy samples elicits an acute injury stimulus that drives FR-PTC accumulation. Since living donor kidney biopsies are the gold standard source of healthy human kidney tissue, we would predict that these samples would not contain FR-PTC if the alternative hypothesis were true.

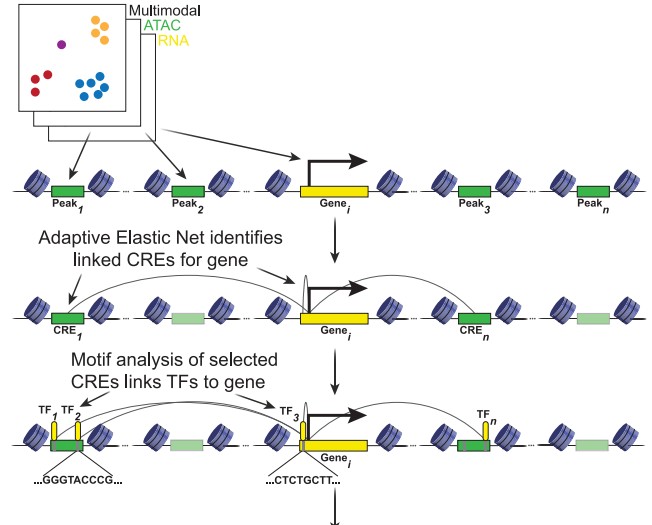

**Fig. 2 | Overview of model design.** Clustered multiome dataset contains chromatin accessibility and gene expression profiles for each nucleus. The model's first step is to learn gene expression predicted by accessibility of peaks within 500kbp of the gene TSS. This step identifies *cis*-regulatory elements (CREs) as peaks with accessibility changes correlated with target gene expression. The second step annotates peaks with potential binding transcription factors (TFs) by scanning for TF motifs. TFs with predicted motifs in predicted CREs are aggregated as putative regulatory TFs for a target gene. The third step is a repeated training step in which the model learns gene expression predicted by expression of TFs selected in the second step. This step identifies putative regulatory TFs based on the correlation between target gene and TF expression in the multiome dataset. For both learning steps, an adaptive elastic-net regression model is used.

We generated snRNA-seq and snATAC-seq (separately, not multiome) libraries from three additional human donors to increase sensitivity. We also added 5 previously published snRNA-seq and snATAC-seq datasets from nephrectomy samples for the same reason. We then compared nephrectomy-derived samples (n = 10) to living donor kidney biopsy samples (*n* = 5 from 3 individual donors) to test this alternative hypothesis. Aggregation of these samples yielded 80,634 cells from snRNA-seq and 120,679 cells from snATAC-seq. We could still clearly detect *KIM1+/VCAM1* + FR proximal tubule cells even when clustering cells solely from living donor kidney biopsies, consistent with recent analysis of living donor kidneys[39], indicating that tumor mass effect alone does not explain FR-PTC accumulation (Fig. 1c, Supplementary Fig. 3a–d). More broadly, cell type-specific RNA expression and ATAC accessibility profiles were similar between living donor kidney biopsies and nephrectomy samples (Fig. 1d, Supplementary Fig. 3e). KIM1 + PT cells derived from either tissue source expressed a set of genes that were not expressed in healthy PT cells. These findings suggest there may be differences between partial nephrectomy and living donor kidney-derived cell populations, like *KIM1+/VCAM1* + PT, although these differences may be influenced by the lower number of living donor samples included and the limited sample size of our dataset. Detection of FR-PT cells in living donor-derived samples supports our usage of both tissue sources in analyzing the FR-PTC multiome.

**Regulatory network inference with adaptive elastic-net model**
A major advantage of simultaneous RNA-seq and ATAC-seq measurements from the same cell is the ability to identifying peak-gene pairs with correlated accessibility and expression. Instead of using overall accessibility or expression levels to predict CREs and TFs, this approach can reduce false positive rates by selecting only elements with accessibility and expression, respectively, correlated to target

gene expression. This correlational analysis generates hypotheses for CREs that are forming distal looping interactions with target gene promoters. We reasoned that TF motifs in candidate CRE provide additional information beyond chromatin accessibility alone. Therefore our model uses a second step, in which putative *trans*-regulatory elements (TFs) are linked to a gene through TF binding motif information using the filtered cisBP motif database provided with the chromVAR package to annotate each CRE and gene promoter region with predicted TF binding sites[23,40]. Then expression of these TFs is used to predict the target gene's expression (Fig. 2). Both modeling steps use an adaptive elastic-net estimator in order to handle collinearity present in single cell datasets and maximize accuracy of regulatory element predictions, leaving fewer but higher quality candidate TFs for biological validation[30]. With this two-step approach, our model can select both key CREs and key TFs to construct gene regulatory networks for a given gene of interest.

**Identifying *cis*-regulatory elements driving healthy-failed repair PT transition**
In the first modeling step, CREs regulating target gene expression are predicted. We modeled marker gene expression of the healthy (PCT and PST) and failed repair (KIM1 + PT) cell clusters to predict key CREs underpinning each PT state (Supplementary Dataset 1). We identified 15,237 unique CREs regulating PT state marker gene expression (Supplementary Dataset 2). In order to validate RENIN's CRE predictions, we performed Cleavage Under Targets and Release Using Nuclease (CUT&RUN) sequencing on cultured RPTECs[41]. CUT&RUN uses histone modification-specific antibodies to target MNase cleavage to nearby DNA, allowing localization of histone modification peaks in the genome. We performed CUT&RUN for two histone modifications marking active chromatin: H3K27ac and H3K4me3. Peaks were called with MACS2 to generate ground-truth PT CRE sets to assess RENIN's performance. Chromatin landscapes in in vivo and in vitro systems can differ substantially, so we first compared RPTEC histone modification CUT&RUN and previously generated ATAC-seq data[11] to proximal tubule cell snATAC-seq profiles to assess whether RPTECs shared enough chromatin accessibility to be used to validate CRE predictions. We generated a PT ATAC profile by taking the top 75% of accessible peaks in PT cells in our multiomic dataset. Approximately 59% of RPTEC ATAC peaks, 63% of H3K27ac peaks, and 87% of H3K4me3 peaks overlapped the snATAC PT peak set (Supplementary Fig. 4a). We also identified H and FR PT differentially accessible regions (DARs) by using Seurat's FindMarkers function to identify peaks with increased accessibility in healthy (PCT and PST) or failed repair (KIM1 + PT) proximal tubule cells, respectively. We found that approximately 52% of snATAC-seq PT peaks, 73% of healthy PT DARs, and 91% of FR PT DARs overlapped RPTEC ATAC peaks (Supplementary Fig. 4b). These findings suggest that while the RPTEC chromatin landscape may contain epigenetic features that are not found in primary kidney tissue, RPTEC accessible peaks include the majority of both healthy and FR PT DARs. Furthermore, the higher representation of FR PT DARs in RPTEC ATAC-seq data supports previous findings that RPTECs cultured on plastic partially recapitulate the FR phenotype[10]. Therefore, despite in vitro differences in chromatin accessibility, these findings supported the use of RPTECs instead of whole kidney epigenetic profiling to validate CRE predictions in order to preserve PT element specificity.

We constructed receiver operating characteristic (ROC) curves of RENIN's ability to recover CUT&RUN peaks. FR-PT (PT_VCAM1) and healthy (PCT and PST) PT predictions both performed better than random chance for recovery of H3K4me3 peaks: area under the curve (AUC) for PT_VCAM1 predictions was 0.696 and 0.607 for healthy PT (Fig. 3a). The higher AUC of PT_VCAM1 predictions again suggests that cultured RPTECs partially recapitulate the FR phenotype. This difference is increased when using the H3K27ac peak set: PT_VCAM1 AUC is 0.694 and healthy AUC is 0.570 (Fig. 3b). Since H3K27ac may be more

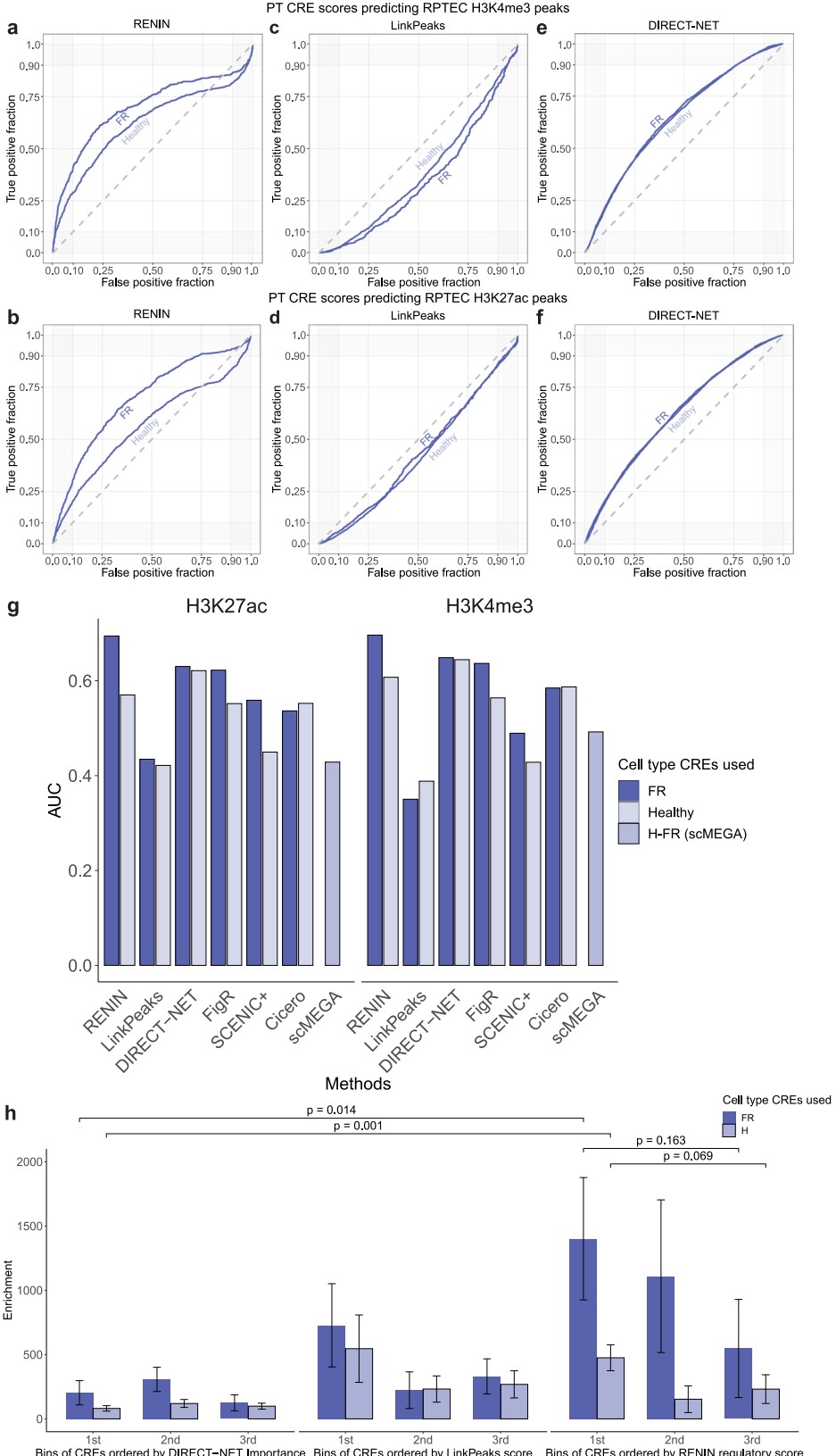

of a marker of enhancer accessibility than H3K4me3[42], RENIN may be particularly useful for identifying distal CREs, which may vary more in accessibility depending on cell states whereas gene body accessibility is more static.

We also compared RENIN with other methods that predict CREs using single cell datasets including Signac's LinkPeaks, DIRECT-NET[25],

FigR[28], SCENIC+[29], Cicero[43], and scMEGA[44]. The univariable approach taken by LinkPeaks resulted in sub-0.500 AUCs with PT_VCAM1 and healthy CREs for both H3K27ac and H3K4me3 peaks (Fig. 3c, d). The sub-0.500 AUC in comparison with the AUC computed with RENIN-predicted CREs confirms that a multivariable approach is best suited for the study of chromatin accessibility due to correlations in

**Fig. 3 | Cell type CREs identified with RENIN. a** ROC curve calculated for RENIN-predicted healthy-failed repair (FR) proximal tubule (PT) *cis*-regulatory elements (CREs) against RPTEC H3K4me3 peaks identified with CUT&RUN performed on $n = 3$ independent samples. FR CREs determined by predicted regulation of a marker gene of the KIM1+ cluster and healthy CREs determined by predicted regulation of a PCT and/or PST cluster marker gene. Source data are provided in the Source Data file. **b** ROC curve calculated for RENIN-predicted healthy-FR PT CREs against RPTEC H3K27ac peaks identified with CUT&RUN performed on $n = 3$ independent samples. Source data are provided in the Source Data file. **c** ROC curve calculated for LinkPeaks-predicted healthy-FR PT CREs against RPTEC H3K4me3 CUT&RUN peaks. Source data are provided in the Source Data file. **d** ROC curve calculated for LinkPeaks-predicted healthy-FR PT CREs against RPTEC H3K27ac CUT&RUN peaks. Source data are provided in the Source Data file. **e** ROC curve calculated for DIRECT-NET-predicted healthy-FR PT CREs against RPTEC H3K4me3 CUT&RUN peaks. Source data are provided in the Source Data file. **f** ROC curve calculated for DIRECT-NET-predicted healthy-FR PT CREs against RPTEC H3K27ac CUT&RUN peaks. Source data are provided in the Source Data file. **g** Area under curve (AUC) calculations for all tested methods against H3K27ac CUT&RUN peaks (left) and H3K4me3 CUT&RUN peaks (right). Each tested method's quantitative metric was used. FigR CREs were scored by rObs, SCENIC+ CREs were scored by

summed R2G_importance_x_abs_rho across all target genes, Cicero CREs were sorted by summed coaccessibility score, and scMEGA CREs were sorted by the TStat metric summed across all target genes. AUCs for H3K27ac peaks: RENIN (FR CRE: 0.694, Healthy CRE: 0.570), LinkPeaks (FR CRE: 0.435, Healthy CRE: 0.422), DIRECT-NET (FR CRE: 0.630, Healthy CRE: 0.621), FigR (FR CRE: 0.622, Healthy CRE: 0.552), SCENIC+ (FR CRE: 0.559, Healthy CRE: 0.450), Cicero (FR CRE: 0.536, Healthy CRE: 0.552), and scMEGA (H-FR trajectory CRE: 0.570). AUCs for H3K4me3 peaks: RENIN (FR CRE: 0.696, Healthy CRE: 0.607), LinkPeaks (FR CRE: 0.350, Healthy CRE: 0.388), DIRECT-NET (FR CRE: 0.649, Healthy CRE: 0.644), FigR (FR CRE: 0.637, Healthy CRE: 0.564), SCENIC+ (FR CRE: 0.489, Healthy CRE: 0.428), Cicero (FR CRE: 0.585, Healthy CRE: 0.587), and scMEGA (H-FR trajectory CRE: 0.513). Source data are provided in the Source data file. **h** Comparison of RENIN, LinkPeaks, and DIRECT-NET by enrichment of partitioned heritability of CKD in model-predicted healthy (PCT + PST) and FR (failed repair−KIM1 + PT) CREs. $N = 7$ biologically independent samples containing 50,768 cells were examined in a joint analysis. Error bars represent standard errors around estimates of enrichment by LDSC with a block jackknife over $n = 200$ equally sized blocks of adjacent SNPs. *P* values shown for two-tailed t-test of difference between enrichment means with degrees of freedom = 199. Source data are provided in the Source Data file.

accessibility between peaks. RENIN also marginally outperformed DIRECT-NET on recall of histone peaks with predicted FR-CREs. DIRECT-NET performed well, predicting PT_VCAM1 and healthy CREs resulting in AUCs of 0.631 and 0.621, respectively, using the H3K27ac peak set and AUCs of 0.649 and 0.644 using the H3K4me3 peak set (Fig. 3e, f). Notably, the PT_VCAM1-healthy AUC difference decreased when using DIRECT-NET predictions. Since RPTECs appear to have an expression profile resembling the FR-PT/PT_VCAM1 state, RENIN may have improved resolution between cell states relevant in disease.

Relative to other methods, FigR, SCENIC+, Cicero, and scMEGA, our configuration had favorable results as well. Most notably, RENIN-predicted FR CREs had the highest recall of RPTEC histone modification peaks (Fig. 3g, Supplementary Fig. 5). To favor enrichment of regulatory function, we favored a smaller value of $\overline{\lambda_2}$, 0.25, to increase the relative weighting of the adaptive $\overline{\lambda_1}$ regularization for this modeling step. As expected, larger values of $\overline{\lambda_2}$ reduce AUC but increase the number of predicted CREs (Supplementary Fig. 6). Therefore, for research questions that require larger numbers of CREs, other methods such as the gradient boosting-based DIRECT-NET should be considered as well.

We confirmed RENIN's ability to identify important CREs by partitioning heritability of kidney related GWAS loci into predicted healthy and FR PT CREs. Proximal tubule cells have been previously reported to enrich for heritability of CKD and eGFR[11]. Cell type partitioning is closely related to cell-specific function and gene expression. We hypothesized that RENIN-identified CREs of cell type-specific genes would enrich for heritability of these traits[45]. We found statistically significant enrichment of both traits in both healthy and PT_VCAM1 CREs, supporting RENIN's utility in identifying functional CREs for traits with a genetic component (Fig. 3h, Supplementary Fig. 7). We binned healthy and PT_VCAM1 CREs by decreasing regulatory score (1st bin to 3rd bin, splitting the CRE set into equally sized thirds) and calculated partitioned heritability into each bin, finding a trend of increasing enrichment of CKD heritability[45] with increasing PT_VCAM1 CRE score. We also binned DIRECT-NET-identified CREs by Importance score and LinkPeaks-identified CREs by score. RENIN predictions had a statistically significant increase in enrichment relative to DIRECT-NET and a trending increase relative to LinkPeaks. We also observed a trending decrease in enrichment from the 1st to 3rd bin for both PT_VCAM1 and healthy CREs ($p = 0.163$ and $p = 0.069$, respectively, by two-tailed t-test of difference between enrichment means), suggesting the parametric ranking may prioritize CKD-relevant CREs. Relative to other methods, RENIN predictions had the highest enrichment of

partitioned heritability, although FigR and SCENIC+ also had high enrichment for both CKD and eGFR heritability (Supplementary Fig. 8). Taken together, these findings validate our approach to enrich for important regulatory regions and highlight the potential relevance of the PT_VCAM1 state in renal disease and declining kidney function.

## Cell type-specific regulatory elements can be predicted with RENIN

Beyond our main focus on modeling regulators of PT state, we asked whether RENIN could be applicable to other more general uses. In our multiomic dataset, we identified an average of 893 (range: 612-1304) cell type-specific genes per cell type (Supplementary Dataset 1). For each of these genes, our modeling identified 32,304 total unique CREs, with an average of 4263 CREs predicted to be active (range: 3016-11006) per cell type (Supplementary Dataset 2). Most (27,934) CREs had one target gene, with 4,370 having two or more targets (Supplementary Fig. 9a). Of the 6,105 modeled genes, 5,038 had at least one linked CRE, with the majority having multiple predicted CREs, suggesting that CRE accessibility is an important regulatory mechanism in human adult kidney (Supplementary Fig. 9b). As expected, predicted CREs were not uniform across cell types, suggesting predictions were able to preserve cell type differences (Supplementary Fig. 9c). Predicted regulatory scores of CREs annotated as promoters (peaks within 2kbp of the target gene TSS) were higher on average than either gene body or intergenic peaks, suggesting our approach assigns quantitative regulatory scores appropriately, although higher baseline accessibility of promoter regions may partially contribute to higher scores (Supplementary Fig. 9d). Beyond the promoter, regulatory scores for intergenic and gene body CREs did not decline with distance, suggesting that RENIN identifies functional CREs rather than regions that are accessible simply due to proximity to an open and transcribed TSS.

We then performed the second modeling step and calculated cell type-specific regulatory scores for each TF by summing their regulatory coefficients for cell type marker genes weighted by their mean expression in each cell type. This allowed us to identify cell type-defining TFs for each renal cell type that replicated known biology (Supplementary Fig. 10a). For example, WT1, TCF21, and MAFB are known podocyte-specific TFs[46–48] and were identified as key podocyte TFs. PPARA and HNF4A, both well-characterized as driving PT differentiation and function[49,50], were top key proximal convoluted and straight tubule TFs. In contrast, PPARA and HNF4A had lower regulatory scores in FR-PTCs, consistent with their dedifferentiated PT phenotype. Notably, RENIN could also be used to predict TFs for rarer cell types: for example, candidate PEC-specific TFs include RFX2 and

TP63. Intriguingly, evidence implicates PECs as progenitor cells and TP63 is a p53 family member implicated in regulation of stem cell state[51–54]. We could also use this approach to identify broader tubular and glomerular TFs, such as HNF1B, suggesting this approach is not restricted to cell type-specific analyses (Supplementary Fig. 10b).

In order to determine whether RENIN's regularized approach provided additional information by integrating motif and expression data, we quantitated differences between top predictions by RENIN, ChromVAR[40], a powerful tool that can be used to identify cell type-specific TFs from scATAC-seq sequence data alone, and TF RNA expression. To do so, we calculated the Jaccard index (ranges from 0 to 1, denoting zero and perfect overlap, respectively) on the top 25 cell type TF predictions by each method. This revealed a low overlap between RENIN and ChromVAR (Supplementary Fig. 10c). The Jaccard index was higher between RENIN predictions and the top 25 TFs per cell type based on their RNA expression alone, in line with our weighting of TFs by mean cell type expression, but under 1, indicating RENIN still predicted cell type TFs that would not have been prioritized with TF expression alone. So, while RENIN's regression approach still relies on high quality expression data, it may still be applicable to lowly expressed genes or rare cell types due to its design and integration of multiple modalities. Motif enrichment approaches such as ChromVAR are very powerful, particularly with access only to scATAC-seq data alone, and may be useful for identifying factors that affect CRE accessibility. These findings demonstrate that RENIN or other similar approaches may be a complementary addition to single cell analysis by integrating both TF expression data and motif data to generate high quality hypotheses about key cell type-defining TFs if multiomic data are available.

## CRE analysis of healthy-failed repair axis reveals coordinated regulatory element remodeling

Next, we applied RENIN to study the gene regulatory networks involved in the healthy to failed repair transition. We hypothesized that expression changes along this transition would allow RENIN to identify *cis*- and *trans*-regulatory elements driving the change in cell state. Using the proximal tubule subset of our dataset, we identified 1666 genes differentially expressed (adjusted *p*-value < 0.05) between healthy (PCT and PST) PT and FR-PTC (Supplementary Dataset 3). Of these, 1516 were non-mitochondrial and present in our EnsDb.Hsapiens.v86 genome annotation reference. We generated multiomic pseudocells using a version of VISION's micropooling algorithm[55], modified to use multiomic WNN graphs to counter limitations introduced by the sparsity of snATAC-seq datasets, and performed the first modeling step. RENIN's first step identified 15,434 Healthy to FR-PTC CREs regulating 1,488 of 1,516 input genes, with an average of 11.4 CREs per differentially expressed gene (Supplementary Fig. 11a). These findings strongly implicate distal chromatin remodeling in the development of the FR-PT state. On average, RENIN explained a proportion of variance of 0.417 for pseudocell gene expression with at least one linked CRE, with a negative $r^2$ for only 89 modeled genes (Supplementary Fig. 11b). Thus RENIN was able to generate a list of candidate CREs implicated in the H-FR transition.

We then asked what factors might be driving changes in accessibility of these CREs, resulting in differential expression along the H-FR transition. We classified CREs as either healthy- (H CREs) or failed repair-promoting (FR CREs) based on their predicted enhancer or repressor role on differentially expressed genes; a CRE was labeled healthy-promoting if it enhanced a healthy-upregulated gene or repressed a FR-upregulated gene and vice-versa. We further hypothesized that motif enrichment might identify factors that were candidate pioneering TFs driving healthy- and failed-repair-promoting CRE accessibility. We calculated motif enrichment within H and FR CREs. The most enriched H CRE motifs were HNF4A/G, while the most enriched FR CRE motifs were AP-1 subunit motifs (Supplementary Fig. 11c). HNF4A/G and AP-1 subunit motifs were the least enriched in

the other CRE peak set, suggesting they may be involved in adversarial processes promoting normal differentiation or fibrosis and a failed repair-associated phenotype. Stimuli altering the balance between the two may influence the likelihood of H-FR transition through the opening of new regulatory TF-FR gene links.

We also performed footprinting analysis to determine whether Tn5 insertion enrichment around motifs in H CREs and FR CREs was different. We found that Tn5 insertion was higher in regions adjacent to HNF4A and PPARA motifs in H CREs relative to those found in FR CREs, while Tn5 insertion was enriched near JUN motifs in FR CREs relative to those in H CREs (Supplementary Fig. 11d). NFKB1, previously identified as a likely factor in the H-FR transition[10], motifs also showed slight increased adjacent Tn5 enrichment relative to H CRE NFKB1 motifs, supporting the identification of these CREs as FR-associated or promoting (Supplementary Fig. 11e). We also performed footprinting analysis for other TF motifs, and did not find the same differential insertion enrichment, demonstrating that this differential insertion enrichment is not due to non-specific differences in H and FR CRE accessibility (Supplementary Fig. 11f). One possible mechanism affecting differences in binding of certain transcription factors may be methylation. We found that RENIN-predicted CREs were overlapped with CKD-associated methylated regions at a higher rate than the set of all called peaks in our dataset with an odds ratio of 1.322 (95% CI = 1.229–1.421, *p* = 2.138e-13 by Fisher's exact test, Supplementary Dataset 4). Our dataset thus contains evidence that epigenetic remodeling of specific CREs, such as methylation, can lead to altered TF binding associated with the H-FR transition.

## Identification of healthy- and FR-promoting TFs with RENIN
The failed repair PT cell state is distinct from healthy proximal tubule cells and the aberrant phenotype is stable long after recovery from renal injury[2,3]. We hypothesized that gene regulatory network activity in the proximal tubule could play a role in establishing and/or maintaining the failed repair population, and that there is a subset of key TFs regulating the transition. We constructed parametric gene regulatory networks for genes differentially expressed between healthy PTs and PT_VCAM1 cells to prioritize candidate TFs for this H-FR axis. We ranked TFs by the sum of their regulatory coefficients across all input genes, weighted by whether the target gene was FR-upregulated (negative score) or H-upregulated (positive score) and TF average expression (Fig. 4a, Supplementary Dataset 5). Top healthy-promoting TFs included canonical PT TFs, HNF4A and PPARA, as well as other factors that have been demonstrated to be protective against kidney disease: ESRRG and RREB1[49,56–58]. The top FR-promoting predictions included NFAT5, a TF that has been previously identified as upregulated following ischemia-reperfusion injury in a mouse model of AKI[3], and KLF6, previously shown to contribute to proximal tubule injury[59]. Thus, RENIN prioritizes plausible TFs involved in the H-FR transition for further investigation.

Next, we assembled predicted gene regulatory networks into a directed graph representation (Fig. 4b). We ranked TFs by two measures of centrality, betweenness and PageRank, to prioritize most central TFs in the H-FR gene regulatory landscape (Fig. 4c–e, Supplementary Fig. 12a). Top central TFs included similar predictions of TF importance—e.g. ESRRG, PPARA, HNF4A, RREB1, NFAT5, CREB5, NFKB1, KLF6, HIVEP2 were all within the top 25 central TFs by both measures. Simulated upregulation of the top 5 predicted failed-repair promoting TFs caused simulated healthy PCT and PST cells to approach KIM1 + PT cells, while simulated upregulation of the top 5 predicted healthy-promoting TFs in KIM1 + PT cells caused simulated cells to approach healthy PCT and PST cells, providing further computational evidence that these top predicted TFs are relevant in the H-FR transition (Supplementary Fig. 12b, c). We designed RENIN to prioritize key TFs and the consistency between multiple methods of ranking supports its usage in identifying disease-relevant TFs for biological validation.

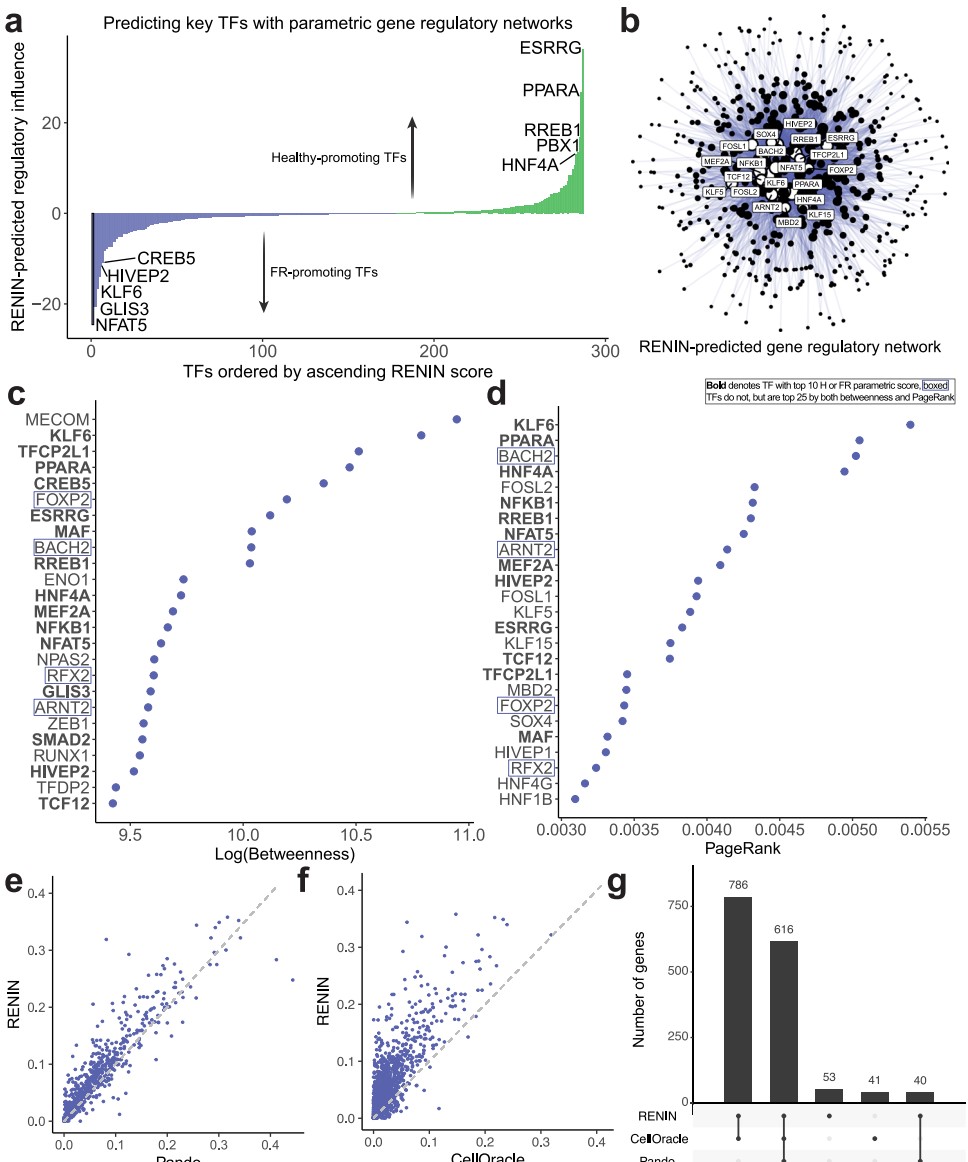

**Fig. 4 | Predictions of key TFs involved in healthy to failed repair PT transition. a Transcription factors** (TFs) sorted by regulatory score, computed as the sum of predicted regulatory coefficients for healthy-FR PT DEGs multiplied by mean TF expression in PT (PCT, PST, KIM1 + PT) clusters. Negative scores indicate FR-promoting TFs—positive regulation of DEGs upregulated in FR PT or negative regulation of DEGs downregulated in FR-PT—and positive scores indicate H-promoting TFs—positive regulation of DEGs upregulated in H PT (PCT and PST) or negative regulation of DEGs downregulated in H PT. Similar TF rankings and scores replicated over $n = 5$ independent trials. **b**. Graph visualization of gene regulatory networks predicted by RENIN. TF node size represents centrality of TFs computed by PageRank, top 20 TFs are labeled. Source data are provided in the Source Data file. **c** Top 25 TFs ranked by betweenness. Source data are provided in

the Source Data file. **d** Top 25 TFs ranked by PageRank. Source data are provided in the Source Data file. **e** $r^2$ calculated for RENIN- and Pando-predicted H-FR gene expression compared to target gene expression in an independent KPMP snRNA-seq dataset for genes that were successfully modeled by both methods. For shared genes, mean RENIN $r^2$ was .080 and mean Pando $r^2$ was .065. Source data are provided in the Source Data file. **f** $r^2$ calculated for RENIN- and CellOracle-predicted H-FR gene expression compared to target gene expression in an independent KPMP snRNA-seq dataset for genes that were successfully modeled by both methods. For shared genes, mean RENIN $r^2$ was .055 and mean CellOracle $r^2$ was .026. Source data are provided in the Source Data file. **g** Number of H-FR differentially expressed genes modeled by each method. Source data are provided in the Source Data file.

We also assessed the ability of RENIN and other regression-based methods to predict H-FR gene expression in an independent dataset generated by the Kidney Precision Medicine Project[60] (KPMP, n = 29 specimen snRNA-seq dataset) after being trained on our multiome dataset to compare performance with less dataset-specific overfitting. We compared the mean $r^2$ of genes that were successfully modeled by each method in each comparison. For modeled genes by both Pando and RENIN, RENIN predictions yielded a mean $r^2$ of .080, relative to .065 for Pando. For genes modeled by both RENIN and CellOracle, the mean $r^2$ was .055 and .026, respectively (Fig. 4e, f).

RENIN was also able to predict more gene regulatory interactions than the other methods (Fig. 4g). We also trained the three methods on the Gerhardt 2023 multiome dataset. Results were similar, with RENIN outperforming both methods in predicting expression in the independent Kirita 2020 dataset (Supplementary Fig. 13a, b). RENIN and CellOracle fit models for similar numbers of genes, while Pando fit models for far fewer genes (Supplementary Fig. 13c). It is very likely that different methods perform better or worse in different contexts and for different research questions. However, for the identification of gene regulatory networks and key TFs

underpinning the H-FR PT transition, these findings demonstrate RENIN is a viable option.

Modeling large datasets can be computationally intensive, therefore we explored whether the use of pseudocell aggregation could be used without deleterious changes to the main predictions. Reassuringly, the same top 5 healthy- and FR-promoting TFs were predicted when aggregating 5 and 10 cells per pseudocell (Supplementary Dataset 5). The top 20 healthy- and FR-promoting TFs were also highly consistent with the use of pseudocells, measured by Jaccard index, with FR TF predictions showing less consistency with a Jaccard index of 0.74 between predictions with no pseudocells and targeting 10 cells per pseudocell (Supplementary Fig. 13d–e). These findings suggest the use of pseudocells has a small effect on TF rankings, particularly for the most highly predicted TFs, so we used 10 cells per pseudocell when testing and exploring different model configurations. If unable to use individual cells due to computational resource restraints, then pseudocell aggregation is an effective way to reduce computational burden during exploratory analysis with only small changes to model predictions.

Many gene regulatory network modeling tools use motif enrichment to identify TFs of interest. This has utility in omitting TF-gene putative links from less frequent computationally predicted motifs, which may be more likely to be false positive connections. While this is a very powerful approach, particularly in developmental contexts, there may be active TF-gene interactions that do not rely on enriched concentrations of conjugate motifs to regulate target genes, particularly for disease processes that may not undergo positive selective pressure. So we modified RENIN to select for enriched, rather than all, motifs present in a gene's predicted set of CREs, relative to a GC-matched background peak set to determine whether top TF predictions differed. Of the 1,516 modeled genes with at least one predicted CRE, 660 were not able to be modeled with this approach, which is consistent with the usage of the more stringent requirement of motif enrichment rather than simply presence. Despite fewer predicted TF-gene regulatory interactions, the top predicted TFs on successfully modeled genes were very similar to our original list: the top five FR TFs were the same between rankings with only differences in ordering and four of five H TFs were shared between rankings. (Supplementary Fig. 14, Supplementary Dataset 6). Since we applied a regularized regression specifically to reduce the number of linked CREs to minimize false positive predictions and because overall top predictions are similar, these results suggest our approach is viable in the adult kidney context.

Given the coordinated chromatin remodeling previously described, we hypothesized that non-promoter CREs comprising a *cis*-coaccessibility network (CCAN) for a target gene have non-redundant regulatory influence due to the presence of additional TF binding sites that regulate the H-FR transition. We compared RENIN's performance when using motifs present in CREs, as well as the promoter, to the promoter alone. We found that regulatory scores of ESRRG, PPARA, NFAT5, and HNF4A decreased, while TCF12 and RREB1 scores increased (Supplemental Fig. 15a). The addition of non-promoter CREs improves the proportion of variance of pseudocell H-FR gene expression explained by RENIN, measured by coefficient of determination (Supplemental Fig. 15b). While the model training results in a sparse list of putative TFs for each gene (mean: 33.2, range: 4-125, relative to initial list based solely on motif presence within a linked CRE averaged 392.2 TFs per gene, range: 18-602. Supplemental Fig. 15c), the improved proportion of gene expression variance explained could be partially due to overfitting as a result of the inclusion of more TFs when scanning distal CREs for putative TFs. So, we compared the likelihoods of RENIN models when incorporating all CRE information versus promoter peak information alone by Akaike Information Criterion (AIC). For 1,323 genes that could be modeled with promoter-only and CCAN configurations, the vast majority—1,184 genes—were modeled more

accurately with the distal CRE inclusion, and we calculated the mean DAIC to be −1293 (Supplementary Fig. 15d, Supplementary Dataset 7). These findings strongly support the relevance of distal regulation, as the inclusion of the information they provide improves model fit. For the remaining 165 genes without any accessible promoter peaks using our reference annotation, RENIN was able to identify CREs and subsequently TFs with the inclusion of distal interactions. These results demonstrate the added utility of implementing the first step of RENIN to identify both distal and proximal CREs in improving accurate modeling of gene regulatory networks with single cell datasets.

While our model configuration seemed reasonable, it remained unclear whether top predicted TFs were generalizable or specific to our dataset of limited ($n = 7$) sample size. We therefore sought to apply RENIN to other datasets to determine whether any of the top predicted TFs were shared across single cell datasets. We still performed the first CRE-gene prediction step with our multiomic dataset, as we do not have access to additional single nucleus multiomic datasets. First, we included additional snRNA-seq datasets generated from the human biopsies in this manuscript ($n = 3$) and from a previous study on control human adult kidney ($n = 5$)[10]. After modeling the healthy (PCT and PST)-FR (KIM1 + PT) transition in this aggregated $n = 15$ dataset, we found highly similar top TF predictions (Supplementary Fig. 16a, b, Supplementary Dataset 8). Eight of the ten original top FR-promoting predictions were within the top 10 predictions from modeling the merged dataset: MEF2A, NFAT5, GLIS3, TCF12, KLF6, FOXP1, HIVEP2, and SOX6. To better test the external validity of these TF predictions, we also modeled the healthy-FR transition in the independent $n = 29$ specimen KPMP snRNA-seq dataset. We preserved the KPMP investigators' cell type annotations, rather than transferring cell type annotations from our datasets to avoid introducing any bias due to differences in preprocessing strategies. Therefore we assessed whether any cell type annotations in the KPMP dataset resembled our KIM1 + PT cluster. The aPT cluster had a similar expression profile, with reduced healthy PT marker expression and increased expression of *VCAM1*, *TPM1*, and *HAVCR1*, so we modeled differential gene expression between the aPT cluster and other PT clusters (Supplementary Fig. 16c, d). Here too, we found highly similar top TF predictions, with the top ten predicted FR TFs including GLIS3, TCF12, FOXP1, SOX6, NFAT5, CREB5, MEF2A, and KLF6. For both datasets, top predicted healthy-promoting TFs included ESRRG, PPARA, RREB1, and HNF4A (Supplementary Dataset 8). Repeat top predicted TFs may therefore drive shared gene regulatory mechanisms promoting FR formation across human populations with different demographic characteristics.

We also asked whether any TFs were conserved across species and in more acute modes of kidney injury. We modeled regulation of differentially expressed genes between cells annotated as healthy and injured PT in a chronic ischemia-reperfusion injury (IRI) mouse single cell multiomic dataset (referred to as Gerhardt 2023 dataset)[9]. Again, we preserved the investigators' original clustering and cell type annotations, modeling regulation of the differentially expressed genes between the PTS1-3 clusters and the Injured PT cluster, which showed expression of FR-PT markers, *Tpm1* and *Vcam1*, and lower expression of *Havcr1* (Supplementary Fig. 17a). Top FR predictions for the Gerhardt 2023 dataset included many targets identified in the human datasets: Glis3, Nfat5, Foxp1, Nfkb1, Pax8, Tcf12, and Klf6 (Supplementary Fig. 17b, Supplementary Dataset 8). We also modeled a separate snRNA-seq mouse IRI dataset (referred to as Kirita 2020 dataset)[2]. As above, we used the CRE-gene regulatory connections from the multiomic Gerhardt 2023 dataset modeling. We removed control samples to enrich for post-acute injury repair mechanisms. We then modeled differential gene expression between healthy PT clusters and the NewPT2 cluster, which has a FR expression profile (Supplementary Fig. 17c). Top predictions for the Kirita 2020 dataset included Pax8, Nfat5, Glis3, and Nfkb1 (Supplementary Fig. 17d, Supplementary Dataset 8). Between the human and mouse datasets, there

were some differences that may represent repair mechanism differences in different species and modes of injury or differences in subpopulation composition of different investigators' annotations. For example, Pax8 was the 6th and 1st predicted FR TF in these two datasets. So while the presence of shared, predicted top transcription factors across both human and mouse datasets indicates conservation of transcriptional programs across modes of injury and species, there may be important differences in mechanisms between different species and modes of injury.

We also assessed overall similarity between top predictions across all 5 modeled datasets by calculating pairwise Jaccard indexes for each dataset's top 10 predicted FR and healthy TFs. Similarity between human datasets was high: at least 7 top 10 FR and H TFs were shared between human datasets (Supplementary Fig. 18a, b). Similarity between human and mouse IRI datasets was lower, likely due to different mechanisms in different species and modes of injury. We also counted the number of times any top 10 TF was predicted in any of the 5 modeled datasets. Twelve of 24 FR-associated TFs and 15 of 21 healthy PT-associated TFs were top TFs in at least 2 datasets (Supplementary Fig. 18c, d). Two FR TF predictions, NFAT5 and GLIS3, and two healthy TF predictions with well-described function in the proximal tubule, HNF4A and PPARA, were top predictions in all 5 datasets[18,49]. TF predictions conserved across datasets are of interest in therapeutically targeting mechanisms of proximal tubule injury and failed repair in kidney disease. Furthermore, shared TFs and gene regulatory mechanisms provide further support that the PT_VCAM1 state present in control, non-AKI human adult kidney and the FR-PT state that develops after AKI share similar transcriptional regulation mechanisms and may thus be targetable by similar strategies. Although the use of additional single cell multiomic datasets would be the best approach to assessing external validity of our predictions here, we believe that the consistency in top TF predictions across datasets, species, and modes of injury is encouraging and increases the likelihood that some of these repeatedly predicted TFs are relevant to the healthy-failed repair transition.

### NFAT5 knockdown partially reverts failed repair phenotype

RENIN identified NFAT5 as the top TF driving the H-FR transition. We next used the RENIN-predicted NFAT5 gene regulatory network to simulate NFAT5 upregulation in healthy PCT and PST cells to predict the resulting effect on cell phenotype. Visualizing cells in the low dimensional PCA space computed by Seurat, simulated upregulation of NFAT5 in a sample of PCT and PST cells caused them to move towards KIM1 + PT cells, consistent with NFAT5's predicted FR-promoting effect (Supplementary Fig. 19). We then asked whether actual siRNA knockdown of NFAT5 in RPTECs would reduce the FR-PT phenotype in culture. We targeted NFAT5 for siRNA knockdown and achieved a 78% reduction in NFAT5 mRNA levels, which was associated with decreased expression of the FR-PTC marker VCAM1, consistent with its predicted role in contributing to the FR phenotype (Fig. 5a). Next, we performed bulk RNA-seq on NFAT5 siRNA knockdown samples compared to control. We found reduced expression of NFAT5 targets involved in one of its roles in driving transcriptional expression in response to hyperosmolar conditions[61–63], suggesting that experimental knockdown levels were sufficient to disrupt NFAT5 regulatory activity (Supplementary Fig. 20). We also found reduced expression of fibrosis-associated genes including TGFB1, TGFB1R, COL1A1, and COL4A1 (Fig. 5b). Knockdown of two other top predicted FR TFs, CREB5 and HIVEP2, resulted in similar changes in expression. CREB5 knockdown by siRNA resulted in decreased expression of VCAM1, TGFB1, and IL6 and increased expression of healthy PT markers such as PPARA, HNF4A, and SLC4A4 (Supplementary Fig. 21a). HIVEP2 knockdown resulted in decreased expression of another FR marker, CCL2,[2,3] and increased expression of healthy PT-expressed genes SLC4A4, MAF, and ACSM2B (Supplementary Fig. 21b). We also tested the results of knocking down PPARA, a top

predicted healthy PT TF. We saw transcriptional changes including reduced expression of proximal tubule-expressed solute-carrier genes SLC3A1, SLC16A12, and SLC22A2 and increased expression of pro-inflammatory genes such as IL1A and IL1B (Supplementary Fig. 21c). For each of the four tested TFs, we also performed KEGG enrichment analysis on RENIN-predicted direct target genes and on the genes differentially expressed in siRNA knockdown samples relative to non-targeting siRNA samples. RENIN predictions identified enriched inflammatory pathways that have been associated with failed repair and kidney injury such as ferroptosis, necroptosis, and glutathione metabolism (Supplementary Fig. 22a–d)[64–66]. In siRNA knockdown samples, we identified enrichment of several shared pathways, including ECM-receptor interaction, TGF-beta signaling, cellular senescence, necroptosis, ferroptosis and other inflammation-related pathways (Supplementary Fig. 22e–h). These findings are consistent with the hypothesis that these TFs drive expression patterns with knock-on effects promoting the FR phenotype. For each tested top TF, siRNA knockdown had the anticipated effect based on modeling predictions. Knockdown of PPARA had the anticipated effect of reducing healthy PT marker gene expression and knockdown of NFAT5, HIVEP2, and CREB5 each reduced FR-associated gene expression and increased healthy PT-associated gene expression in RPTEC culture.

We also tested whether siRNA targeting of these specific TFs validated predicted TF-gene regulatory interactions. Overall, 156 of 445 predicted NFAT5 targets, 84 of 439 predicted HIVEP2 targets, 94 of 329 predicted CREB5 targets were significantly downregulated when knocking down the respective TF (Supplementary Fig. 23a). Only 27 of 472 PPARA-predicted targets were significantly downregulated, which may be because the RPTEC phenotype is closer to FR than it is to healthy PT, with low healthy PT TF expression levels. Since differentially expressed genes with siRNA treatment may be due to different cellular context in vitro and secondary regulatory effects, we also performed CUT&RUN for NFAT5 in RPTECs in order to identify directly bound NFAT5 targets. Of 445 RENIN-predicted NFAT5 targets, 379 were bound by NFAT5 (Fig. 5c). Relative to other tested methods that predict TF-gene regulatory connections, this was the highest number of recovered genes, although all methods had high precision. Results were similar for HIVEP2 CUT&RUN peaks: of 439 RENIN-predicted HIVEP2 targets, 368 were bound by HIVEP2 (Supplementary Fig. 23b). Pando predicted fewer targets but had similar levels of precision, and other methods either did not predict any H-FR differentially expressed genes to be regulated or did not model HIVEP2 as a potential regulator. While most ATAC-seq peaks containing computationally predicted motifs did not overlap with CUT&RUN peaks for either NFAT5 or HIVEP2, approximately half of NFAT5 motif- or HIVEP2 motif-containing FR PT DARs were bound by each respective TF in our CUT&RUN data (Supplementary Fig. 24). These results demonstrate cell type-specific NFAT5 and HIVEP2 activity. Further improvements to bioinformatic representations of TF binding motifs and/or binding cofactor requirements may improve regulatory model performance by reducing the number of false initial TF-motif-gene connections. We were also able to identify specific distal CREs for key predicted targets with CUT&RUN binding peaks such as the previously identified FR gene, TPM1 (Fig. 5d)[10]. These findings support the use of computational gene regulatory network modeling for the study of transcriptional regulation and identification of targetable mechanisms in disease.

Finally, we sought to confirm NFAT5 expression and binding in adult human kidney cortex, as previous work on NFAT5 has focused on the medulla. Immunostaining for NFAT5 showed a patchy expression pattern, consistent with the hypothesis that FR-PTCs accumulate progressively in a scattered fashion over time. NFAT5 expression was anticorrelated with LTL staining, with LTL-low cells and tubules being highest in NFAT5 expression (Fig. 5e). While NFAT5 expression in adult human kidney proximal tubules is low in control kidneys, NFAT5

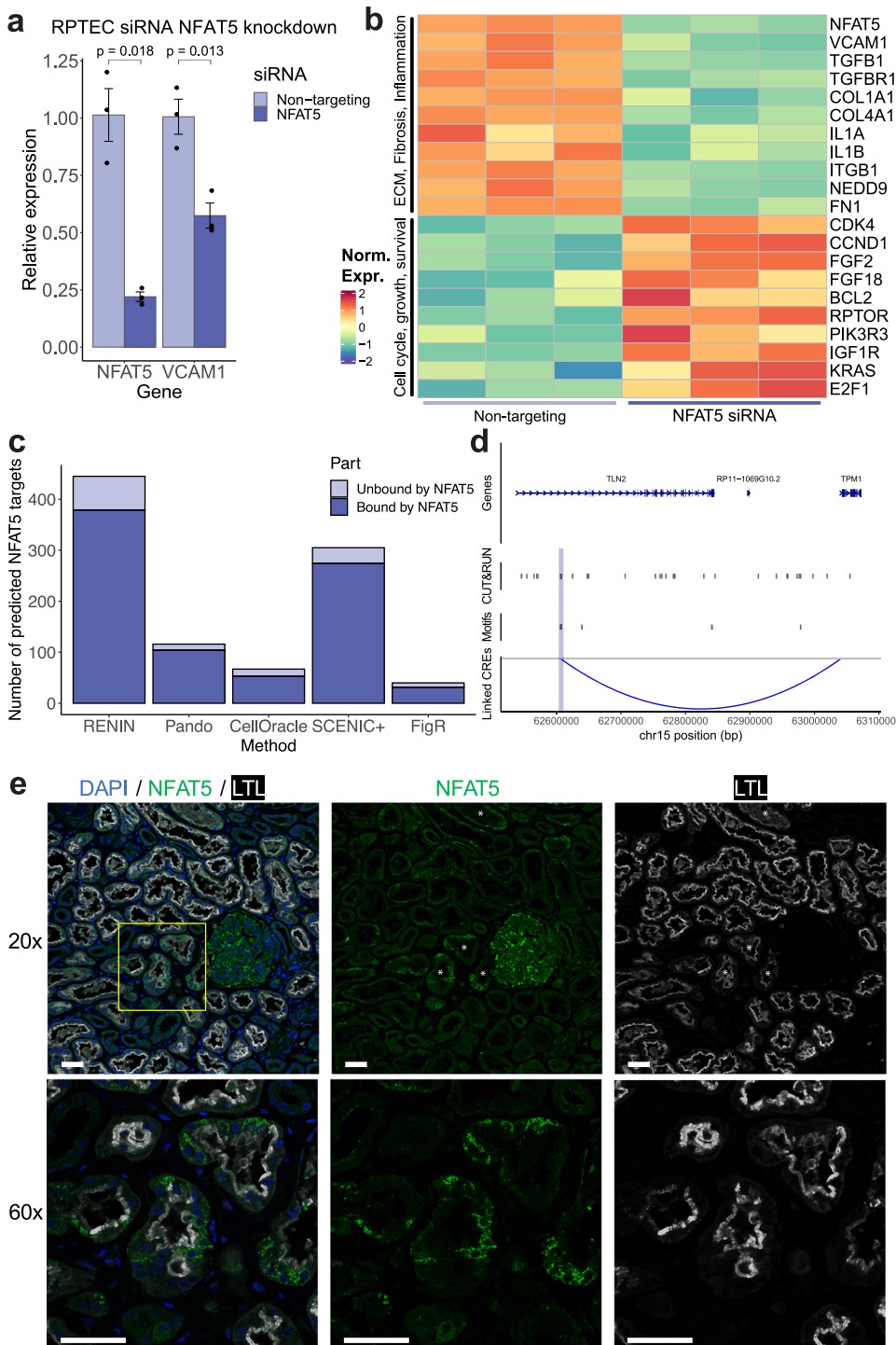

**Fig. 5 | NFAT5 promotes FR expression phenotype in cultured RPTECs.**
**a** Expression of *NFAT5* and *VCAM1* in cultured RPTECs treated with (*n* = 3 independent samples) non-targeting (NT) or (*n* = 3 independent samples) *NFAT5*-targeting small interfering RNA (siRNA). RNA levels measured by quantitative reverse transcription PCR (RT-qPCR) and normalized to *GAPDH* expression. *NFAT5* siRNA-treated cells had 22% of the *NFAT5* RNA and 57% of the *VCAM1* RNA levels of non-targeting-siRNA-treated cells. *P* values calculated with two-tailed t-test with unequal variance. Data are presented as mean ± standard deviation. Source data are provided in the Source Data file. **b** Heatmap of select differentially expressed genes by RNA-seq in NT and *NFAT5* siRNA-treated RPTECs. **c** Number of predicted NFAT5 targets by each method, separated into target genes that were bound versus

unbound on NFAT5 CUT&RUN-seq performed on *n* = 2 independent RPTEC samples. 379/445 RENIN-predicted targets, 104/116 Pando-predicted targets, 53/67 CellOracle-predicted targets, 274/305 SCENIC + -predicted targets, and 31/40 FigR-predicted targets were bound by NFAT5 assessed by CUT&RUN-seq on RPTEC culture. **d** Distal predicted CRE for *TPM1*, predicted to be NFAT5 target FR gene and downregulated with siRNA NFAT5 knockdown, bound by NFAT5.
**e** Immunofluorescent labeling of NFAT5 in adult human kidney. DAPI is a nucleus marker and LTL is an apical proximal tubule marker. * denotes examples of tubules with low LTL intensity. Representative image of *n* = 3 independently analyzed samples. Sample clinical data in Source Data. Scale bars are 50 μm in length.

appears to colocalize with VCAM1 in CKD/ESRD (Supplementary Fig. 25). The presence of NFAT5+, VCAM1-, LTL- tubules is consistent with NFAT5's known role in the kidney against osmotic stress in distal tubular cells[62,63,67]. Given this known role, NFAT5 likely binds many genes and regulatory elements that are not involved in the H-FR transition. To test the hypothesis that NFAT5 is involved in the H-FR process, we performed NFAT5 CUT&RUN in primary healthy and CKD human kidney and calculated the enrichment of FR PT-associated genes, peaks, and NFAT5 motifs in whole kidney NFAT5 CUT&RUN peaks relative to their representation in the multiomic dataset. There were 1,666 genes identified as H-FR DEGs out of the 18,262 genes (8.3%) with genomic annotations and expression in our dataset. Of 1070 genes bound by NFAT5, 237 were in the H-FR DEG list (22.2%, Supplementary Fig. 26a). Chromatin peaks with increased accessibility in KIM1 + PT cells (FR PT DARs) represented 10,082 of the 193,787 peaks (5.2%) in our dataset, and 437 of 1845 NFAT5-bound peaks that overlapped with our dataset peaks were FR PT DARs (23.7%, Supplementary Fig. 26b). Finally, 17,499 peaks in our dataset had at least one computationally predicted NFAT5 motifs. Of these, 1,543 lie within FR PT DARs (8.8%). 824 NFAT5 motif-containing peaks overlapped with NFAT5 CUT&RUN binding, and 198 of these were FR-PT DARs (24%, Supplementary Fig. 26c). NFAT5-bound genes, chromatin accessibility peaks, and predicted motifs were all significantly enriched for H-FR-association. Therefore, although NFAT5 is involved in several processes in different renal cell types, a significant portion of its binding activity and regulatory function is represented in H-FR regulatory processes, supporting its role in promoting the failed repair PT state. These results suggest that NFAT5, as well as other top predictions such as HIVEP2 and CREB5, could be a therapeutic target. They also demonstrate a template to identify key TFs driving disease processes using multiomic single-cell sequencing datasets.

## Discussion

Profibrotic, proinflammatory PT cell states have been increasingly predicted to play a role in CKD development following incomplete PT repair after acute injury or during chronic progressive disease[3,10,11,68]. Here we performed joint single nucleus RNA and ATAC sequencing on adult human kidney samples to study the PT_VCAM1 state in adult human kidney. We found that the presence of PT_VCAM1 was not exclusively explained by sample origin, as populations were also detected in biopsy-derived samples. These results support the use of nephrectomy samples as adequate control tissue in human kidney research, including for the study of the role of PT_VCAM1 cells in CKD. While the detection of PT_VCAM1 in clinically healthy biopsy samples could indicate the accumulation of subacute, subclinical injury over time, an alternative hypothesis is that this represents a progenitor population that can proliferate post-injury to replenish tubular cell populations[69]. While we have previously found evidence in opposition to this hypothesis[6,70], we envision that RENIN could easily be applied to a single-cell multiomic dataset of the proliferation process in order to identify regulatory elements that promote or inhibit progenitor proliferation.

We applied RENIN to our multiomic dataset to study both *cis*- and *trans*-regulatory machinery in the proximal tubule. We found that predicted CRE regulatory influence was associated with enrichment of CKD heritability, further implicating PT_VCAM1 in CKD. Risk variants open in PT_VCAM1 CREs may exert their effects through amplifying profibrotic, proinflammatory signaling or increasing the accumulation and/or persistence of the PT_VCAM1 state. We also identified a small set of TFs predicted to regulate the healthy-PT_VCAM1 transition. Our model suggests targeting some candidate TFs such as NFAT5, HIVEP2, and CREB5 may attenuate the PT_VCAM1 phenotype. Examples of protective TFs include ESRRG and PPARA, which may regulate mitochondrial function and metabolism in a protective capacity[49,56,71]. NFAT5 has previously shown to play a protective role in AKI and

against hypertonic stress[72]. Our results suggest that chronic expression of NFAT5 may instead promote a proinflammatory phenotype in PT_VCAM1 cells.

Gene regulatory network modeling is a powerful application of single-cell sequencing datasets. Many well-designed methods for gene regulatory network modeling have been created, each offering unique developments and strategies[73]. DIRECT-NET recovers maximal numbers of CREs with good true and false positive rates. Methods like FigR and scMEGA could use pseudotemporal or temporal ordering of trajectories to study regulators of chromatin accessibility and developmental contexts. SCENIC+ introduced an expansive motif database and a new method of motif enrichment analysis. CellOracle introduced TF perturbation simulation methods. Our multiomic dataset and catalog of cell type-specific regulatory elements will serve as a resource for modeling renal gene regulatory networks with these methods, but we also found RENIN's parametric design had great utility in highlighting both CREs and TFs that are relevant for our renal disease research interest through full integration of information from both expression and chromatin accessibility data. Our adaptive elastic-net-based approach had competitive performance in selection of key regulatory elements while minimizing the risk of overfitting and misallocation of priority relative to other methods. Model sparsity may reduce false positive rates for predicted regulators and explain the higher enrichment for risk variants and CUT&RUN peaks. We have tailored our approach so that the output is highly interpretable in order to facilitate intuitive understanding of regulatory network composition and selection of candidates for biological validation. The ability to effectively prioritize candidate regulatory factors for further investigation makes RENIN, available at https://www.github.com/nledru/RENIN, a highly effective computational tool for hypothesis generation and a tool to consider using to complement other gene regulatory network methods. As computational tool development continues, it will be important to continue building on the strengths of different algorithms to improve modeling of gene regulation.

With just two sequencing modalities, our modeling explained a large portion of variance in the expression of H-FR genes of interest. In agreement with previous studies, we found strong evidence that distal regulatory interactions play significant roles in regulating gene expression, which RENIN could leverage accurately to characterize gene regulatory networks with high resolution. This work demonstrates that increased profiling resolution can be derived from integration of different data streams, including gene expression, chromatin accessibility, and GWAS. In the future, ranking CREs by predicted regulatory influence may be integrated with GWAS to increase risk variant detection and elucidate new disease mechanisms. Spatiotemporal control of transcription is achieved through layers of regulatory control that tune the accessibility of CREs. For example, histone modifications reduce DNA-histone interaction and facilitate DNA access by transcription factors or other chromatin remodeling proteins[74]. Other epigenetic modifications, such as DNA methylation, may alter the ability of transcription factors to bind their cognate motifs[75]. Due to the kidney's high cellular complexity, single-cell methods are essential for studying specific cell type or state processes, which limits our ability to profile all epigenetic and transcriptional regulation factors with cell type specificity. Future work on gene regulatory networks will benefit greatly from single-cell profiling of epigenetic features such as methylation and histone modifications[15], in addition to the sequence and accessibility information used in this study. Additional single-cell epigenetic modalities will improve CRE identification while preserving cell type specificity without the need for in vitro cell culture models, which do not fully recapitulate in vivo biology. Other improvements to modeling may include efforts to model translational regulation or non-steady-state RNA dynamics[76]. Computational representations of TF binding motifs that better

model cofactor and sequence requirements may boost modeling performance as well by reducing false positive motif annotations. Single-cell sequencing technology continues to improve and capture more complex data. Continued development of methods incorporating these data will facilitate exciting discoveries in disease processes and new therapies.

## Methods

### Declarations
This research complies with all relevant ethical regulations and has been approved by the Washington University Institutional Review Board (IRB ID# 201601020).

### Tissue procurement
Non-tumor kidney cortex samples were obtained from patients undergoing partial or radical nephrectomy for renal mass at Brigham and Women's Hospital (Boston, MA) under an established Institutional Review Board protocol approved by the Mass General Brigham Human Research Committee and at Barnes Jewish Hospital (St. Louis, MO) under an established Institutional Review Board protocol approved by the Washington University Institutional Review Board. Additional kidney cortex samples were obtained from living kidney donors prior to surgical implantation at Wake Forest University under an established protocol approved by the Wake Forest University Institutional Review Board. Additional samples from kidneys rejected for transplantation were obtained from deceased organ donors provided by Mid America Transplant under an established protocol approved by the Washington University Institutional Review Board. All samples were deidentified and laboratory data were abstracted from the medical record. All participants provided written informed consent in accordance with the Declaration of Helsinki. All appropriate consents, including to publish, have been obtained in the original consent document. Samples were frozen for storage.

### Nuclear dissociation and library preparation
For single nucleus multiomic RNA- and ATAC-seq, samples were first cut into <2 mm pieces and homogenized using Dounce homogenizers and the loose head pestle (885302-0002; Kimble Chase) in 2 ml of Nuclei EZ Lysis buffer (NUC-101; Sigma-Aldrich) with protease inhibitor (5892791001; Roche) at 4 °C. Samples were then filtered through a 200 μm cell strainer (43-50200; pluriSelect) and homogenized in the Dounce homogenizers with the tight head pestle. Samples were incubated on ice for 5 minutes in 4 ml of EZ Lysis buffer, then filtered through a 40 μm cell strainer (43-50040; pluriSelect) and centrifuged at 500 g for 5 minutes at 4 °C. The resuspended pellet was then washed with 4 ml of lysis buffer and incubated for 5 minutes at 4 °C. Then the sample was centrifuged again and resuspended in Diluted Nuclei Buffer (PN-2000153; 10X Genomics), then filtered through a 5 μm cell strainer (43-50005; pluriSelect). After counting, nuclei suspensions were diluted if needed to target 10,000 nuclei per lane and loaded into a thermal cycler to begin the transposition reaction, following 10X Genomics' protocol. The manufacturer protocol was followed for the completion of library preparation.

Paired single nucleus RNA and ATAC libraries were prepared from biopsies with a modification of the protocol detailed above. Biopsies were homogenized as above, incubated for 5 minutes, filtered through a 40 μm cell strainer, then spun down at 500 g for 5 minutes at 4 °C. The pellet was then resuspended in Diluted Nuclei Buffer and strained through a 5 μm cell strainer. An aliquot of the resulting nuclei suspension was used to construct the snATAC-seq library following 10X Genomics' protocol. The remainder was diluted with 1x DPBS (14190144; Gibco – Thermo Fisher) and used for snRNA-seq library construction following 10X Genomics' protocol.

### Multiomic sequencing bioinformatics workflow
Seven single nucleus multiomic libraries were generated from 5 control human nephrectomy samples and 2 pre-transplant human biopsies. Libraries were sequenced on an Illumina Novaseq platform with 28-10-10-150 bp configuration for RNA-seq libraries. ATAC-seq libraries were sequenced with either a 50-8-16-50 bp configuration (N1 and N2) or 2x150 bp configuration (rest of samples). Libraries were counted with cellranger-arc 2.0.0 using the 10X-provided GRCh38-2020-A-2.0.0 reference genome and were aggregated for each sample with cellranger-arc aggr. The variational autoencoder CellBender 0.2.0 was used to reduce ambient RNA signal in each sample. The expected number of cells for each sample was estimated from cellranger-arc output, and cell probabilities were calculated for approximately 10,000 additional barcodes per sample. CellBender was run using with the following parameters: --epochs 150, --z-dim 100, --z-layers 500, --learning-rate 0.0001. For two samples, b6 and b8, the learning rate was reduced to 0.00005 to reduce training instability. Prior to doublet removal, ArchR 1.0.1 was used for preliminary filtering (TSSEnrichment >= 4, BlacklistRatio <= 0.01, NucleosomeRatio <= 4, nFrags >= 3000, and nFrags <= 100000). Seurat 4.0.2 was also used for preliminary filtering (nCount_ATAC > 3000, nCount_ATAC < 100000, nFeature_ATAC > 1000, nCount_RNA < 50000, nCount_RNA > 1000, nFeature_RNA > 500, percent.mt <5, percent.rps <2, percent.rpl <2), and cell barcodes passing both sets of filtering criteria were retained.

After the removal of low-quality barcodes, predicted doublets, and remaining small likely-doublet clusters, SCTransform was used for normalization of the snRNA-seq component, and Harmony 0.1.0 was used for batch effect correction on the SCT output assay. Seurat's FindMarkers function was used on the data slot of the SCT assay with min.pct = 0.1 to identify differentially expressed genes, both those upregulated in each cell type relative to the rest of the dataset and between healthy and failed repair PT cell clusters. To process the snATAC-seq component, term-frequency inverse-document-frequency (TFIDF) was computed, then dimensional reduction was performed on the TFIDF matrix with singular value decomposition. Harmony was used for batch effect correction on the resulting latent semantic indexing reduction. A gene activity matrix was computed with the GeneActivity function, including the 2000 bases upstream of the TSS. Cell type-specific peaks were called with MACS2 2.2.7.1 using the Signac CallPeaks (1.2.1) function with default parameters for CRE analysis. Differentially accessible regions (DARs) for proximal tubule cell states were identified with Seurat's FindMarkers function with the logistic regression test, min.pct = 0.1, and increased accessibility in the PT cell cluster relative to other cells in our dataset. The healthy PT DAR set was generated as the union of PCT and PST DARs. We removed the bottom 25% of accessible peaks in the PCT, PST, and KIM1 + PT clusters to derive a whole kidney proximal tubule chromatin accessibility profile for comparison with RPTEC chromatin accessibility. A weighted shared nearest neighbor graph (SNN) was constructed on the RNA and ATAC batch-corrected dimensional reductions. Clustering of the weighted SNN graph was performed with smart local moving. A WNN UMAP reduction was calculated with the RunUMAP function for visualization.

### Doublet detection algorithm comparison
The three algorithms tested were AMULET 1.1, ArchR 1.0.1, and DoubletFinder 2.0.3. Output metadata of cellranger-arc count was modified for compatibility with AMULET with a custom script. AMULET was run on the ATAC modality for each sample, omitting genomic regions in the ENCODE hg38 blacklist[77]. Barcodes with $q < 0.05$ were identified as doublets. For doublet removal with ArchR, default settings of the addDoubletScores function are to perform 5 trials of simulating a number of doublets equal to the number of cells in the sample. ArchR's addDoubletScores function was modified to use previously generated

paired lists of randomly sampled barcodes. DoubletFinder was modified to use the same paired list of randomly sampled barcodes.

## Biopsy versus nephrectomy comparison

We aggregated our multiomic library with previously generated single nucleus RNA-seq and ATAC-seq libraries from 5 control nephrectomies (GEO accession number GSE151302), as well as snRNA-seq libraries generated from 3 pre-transplant biopsies and 2 pre-transplant biopsy-derived snATAC libraries. The 3 snRNA-seq libraries were processed in a similar manner to the multiome dataset, removing low quality cells with the following filtering parameters: nFeature_RNA > 500, nFeature_RNA < 6000, nCount_RNA < 16000, percent.mt <0.8, percent.rps <0.4, percent.rpl <0.4. For the ATAC modalities, we aggregated fragments in the non-multiome samples into the 193,787 accessible peaks called on the multiome samples. We treated the multiome cell type annotations as ground truth cell type identities and performed label transferring with Seurat to the single modality snRNA-seq and snATAC-seq datasets in order to generate unified complete datasets. We retained all nuclei with maximum prediction scores greater than 0.5, then used Seurat's FindMarkers function to identify RNA and ATAC cell type markers. We then split each dataset into subsets of cell types by origin−pre-transplant biopsy or nephrectomy.

## The adaptive elastic-net for single-cell datasets

In snATAC-seq datasets, chromatin accessibility values between nearby peaks are not always fully independent, a feature that has been leveraged previously to assemble networks of *cis*-coaccessible peaks[43]. Therefore for most genes, a multivariable regression model is more appropriate for modeling the influence of CRE accessibility on gene expression. Additionally, it stands to reason that adjusting for the influence of all CREs within a regulatory network facilitates more accurate selection of key regulatory elements with lower false discovery rates relative to a univariable modeling approach for a given gene's expression.

However, single cell multiome datasets have some characteristics that may impede model performance. The sparse and high-dimensional nature of single cell datasets makes it difficult to select a minimal set of regulatory elements for a given gene, with hundreds of possible peaks and TFs that may target any given gene. Within gene regulatory networks, groups of genes can be regulated by the same regulatory elements, and the resulting collinearities increase the difficulty of selecting the most important regulatory elements. The adaptive elastic-net has several features that are well-suited for single cell datasets[30]. First, model complexity is penalized, which enables variable selection in high-dimensional datasets. An adaptive L1 penalization reduces false positives in selecting regulatory elements for a given gene. Two critical weaknesses of the lasso estimator, a common choice for modeling this kind of dataset, are instability in variable selection and lack of oracle property. The adaptive L1 penalty addresses these weaknesses by introducing adaptive weights for penalizing each coefficient[78]. An additional L2 penalization term increases model stability in high dimensional datasets with collinear predictor variables, which is a defining feature of single cell datasets. Finally, parametric model construction allows for ranking of regulatory elements, thus improving yield when selecting potential key regulatory elements for biological follow-up and validation. The adaptive elastic-net estimator is therefore well-suited for identification of regulatory elements of target genes with single cell datasets.

## Gene regulatory network modeling

Nuclei can first be aggregated into pseudocells using a modified version of VISION's micropooling algorithm[55]. We recommend this for the first step due to high sparsity of snATAC-seq datasets and to increase computational tractability during exploratory analysis. Briefly, the WNN UMAP graph is used to perform Louvain clustering, then nuclei are partitioned into pseudocells with k-means, targeting a maximum of 100 nuclei for peak aggregation, due to the increased sparsity of snATAC-seq data, and 10 nuclei for RNA aggregation. As cells are only grouped with WNN graph neighbors, this approach aims to reduce the effects of sparsity of single cell datasets while maximizing preservation of variance and heterogeneity across individual cells and cell types. Peak accessibility and RNA expression are then averaged within pseudocells. We did not use pseudocell aggregation when attempting to predict specific gene targets of predicted key TFs, because aggregation may cause some regulatory interactions to be averaged out. We recommend the use of individual cells for the second step, when possible, to take full advantage of individual cell resolution in single cell datasets for the most accurate inference of gene regulatory networks for simulations and validation of TF-gene regulatory relationships.

Accessibility and expression matrices are then used to perform two regression steps. In the first, accessible peaks within 500kbp of each modeled gene are identified as putative CREs, a window that earlier CRE prediction approaches used[43]. An adaptive elastic-net model is trained to predict the gene's expression with non-binarized peak accessibility in order to select for CREs regulating each modeled gene. Model-predicted CREs and the gene promoter region are then scanned for motifs in chromVAR's filtered version of the cisBP v0.2 database (chromVARmotifs 0.2.0)[23,40]. Motifs were added to the Seurat object with the AddMotifs function and BSgenome.Hsapiens.UCSC.hg38 1.43.0 for human datasets or BSgenome.Mmusculus.UCSC.mm10 1.43.0 for mouse datasets. Signac's RunChromVAR function and chromVAR 1.14.0 were used to compare RENIN and chromVAR results. TFs predicted to bind a significant CRE or a peak within the gene's promoter region (−2000 to 0 bp from TSS) are aggregated as putative regulators of each modeled gene. A second adaptive elastic-net model is then trained for each gene, using the expression of TFs with potential to bind somewhere in the CRE or promoter set. In each adaptive elastic-net step, we model the response vector $y = (y_1, y_2, \ldots y_n)^T$, where $y_i$ is the expression of the target gene, $y$, in pseudocell $i \in \{1, 2, \ldots n\}$ and $n$ is the number of pseudocells, as $y = X\beta + \epsilon$. The predictor matrix $\mathbf{X}$ contains $n$ rows, for each pseudocell $i$, $\mathbf{x} = (1, x_1, \ldots x_p)$, where $p$ is the number of regulatory elements, CREs for the first step and TFs for the second step, used to predict target gene expression. Then, the following optimization problem is solved:

$$\hat{\beta} = \left(1 + \frac{\lambda_2}{n}\right) \left\{ \arg\min_{\beta} ||y - X\beta||_2^2 + \lambda_2 ||\beta||_2^2 + \lambda_1^* \sum_{j=1}^{p} \hat{w}_j |\beta_j| \right\} \quad (1)$$

The adaptive elastic-net estimator, $\hat{\beta} = \{\beta_1, \beta_2 \ldots \beta_p\}$, is computed for all target genes, assigning predicted weights to all included regulatory elements for each gene, with the adaptive weights, $\hat{w}_j = (|\hat{\beta}_j(\text{enet})|)^{-\gamma}$ with $\gamma > 0$, first calculated with the elastic-net estimator[79], resulting in higher penalties for regulatory elements with lower first estimated weights. Cross-validation is used to determine the optimal $\lambda_1$ values for the elastic-net and adaptive elastic-net estimators. Lower values of $\lambda_2$ increase model sparsity. We used a value of 0.25 during CRE prediction in order to increase enrichment of regulatory importance and partitioned heritability. For TF prediction, we used a value of 0.5. This estimator has the benefits of both the adaptive weighted L1 penalty and the elastic-net penalty: increasing model sparsity, producing more interpretable gene regulatory networks with fewer factors implicated, while maintaining the oracle property[30]. It also allows us to simulate the effects of knocking down or upregulating select TFs, $X_{perturb}$, as $\hat{y} = X_{perturb}\widehat{\beta_{TF}}$. In order to determine significance, we model $\hat{\beta}_j \sim \text{Normal}(\hat{\mu}_j, \hat{\sigma}_j)$. We bootstrap 1000 and 100 training sets for RENIN's first and second steps, respectively, to estimate standard errors of each estimated regulatory weights, and retain weights predicted to be nonzero with $p < 0.05$. Runtime for each step with these parameters was under an hour on our dataset with multithreading.

## Regulatory element prioritization

The output of the two modeling steps is a set of functions of the form:

$$\mathrm{Expr}_{gene_i} = \sum_{p=1}^{n} \beta_p \cdot \mathrm{CRE}_p \quad \text{and} \quad \mathrm{Expr}_{gene_i} = \sum_{p=1}^{n} \beta_p \cdot \mathrm{TF}_p \quad (2)$$

In each step, each modeled gene's expression is the weighted sum of regulatory elements plus an intercept and error term. The weights of each regulatory element are the learned coefficients, and are a measure of that regulatory element's influence on that gene's expression. We take two complementary approaches to regulatory element prioritization. First, we chose to multiply each regulatory weight by the mean accessibility and expression of predicted CREs and TFs, respectively, as this would allow us to rank TFs by absolute regulatory effect on a given gene. Since this approach may be biased towards highly expressed genes, we also rank TFs by measures of centrality. After constructing a graph containing all modeled gene regulatory networks, TFs most central in the graph are prioritized as key candidate regulatory TFs. Centrality measures used for this work were PageRank and betweenness.

## Comparison with other gene regulatory network methods

We followed each method's tutorial to model gene regulatory networks. For CRE modeling and benchmarking, we compared CRE detection for marker genes of healthy (PCT and PST) and FR (KIM1+) PT clusters. Signac's LinkPeaks (1.2.1) function was used with default parameters, and links for PT cluster marker genes were retained. The score for each CRE was used, and were summed for any CREs with multiple predicted linked genes. We ran DIRECT-NET for PT cluster marker genes, and Importance scores were used and summed as for LinkPeaks for CREs with multiple predicted linked genes. For Cicero, we generated coaccessibility scores for all pairs of coaccessible peaks in our dataset, then generated cis-coaccessibility networks with Cicero's (1.3.9) default settings. Cicero uses snATAC-seq data, so peak accessibility is correlated to other peaks' accessibility instead of gene expression. Therefore we identified PT marker genes with 2kbp upstream promoter or gene body peaks, then any CCAN peaks were retained as putative PT marker gene CREs. Multiple coaccessibility scores were summed for any peaks with multiple predicted coaccessible peaks, then overall coaccessibility scores were used to rank peaks by predicted cis-regulatory function. For FigR (0.1.0) and SCENIC+ (1.0.1), we used pycisTopic (1.0.3) to perform topic modeling, using a 10,000-cell subset of our dataset due to computational limitations. For both methods, default settings were used and predicted CREs linked to PT marker genes were kept. For FigR, the rObs for each predicted CRE was used for ranking. For SCENIC+, the R2G_importance_x_abs_rho was used to rank predicted CREs, summing multiple values for CREs with multiple linked genes. For scMEGA (0.2.0), MOJITOO 1.0 was used to integrate RNA and ATAC modalities, followed by ArchR's AddTrajectory function to generate a pseudotime trajectory. The TStat method was used, with multiple values summed if needed, as for previous methods.

For TF modeling and benchmarking, we identified NFAT5 and HIVEP2 predicted targets in the set of H-FR differentially expressed genes with each method that generates TF-gene regulatory predictions. For Pando (1.0.3) and CellOracle (0.12.1), we used default settings to predict regulators of the H-FR differentially expressed genes in our dataset. For FigR and SCENIC+, all genes are modeled, so we filtered the predicted TF-gene regulatory interactions for H-FR genes. FigR and CellOracle come with their own motif databases, cisBP_human_pfms_2021 and cisBP_ver2_Homo_sapiens, respectively, which did not contain HIVEP2 motifs. Finally, for regression-based approaches, we also used the TF-gene regulatory network models to predict H-FR gene expression in the $n = 29$ KPMP dataset[60] or in the $n = 18$ Kirita 2020 dataset[2]. For RENIN-Pando and RENIN-CellOracle comparisons, $r^2$ was calculated for each gene that was successfully modeled by both datasets.

## Cell culture

Human primary proximal tubular cells (RPTECs; Lonza, CC-2553) were cultured with renal epithelial cell growth medium (REGM; Lonza, CC-3190). Cell cultures were maintained in humidified 5% $CO_2$ at 37 °C.

## CUT&RUN sequencing

CUT&RUN on primary RPTEC culture was performed with the CUTANA kit (EpiCypher, 14-1048) according to the manufacturer's instructions. The primary RPTEC with early passages were seeded at $8 \times 10^5$–$8 \times 10^6$ cells per 10 cm culture dish 24 h prior to CUT&RUN assay. 37% formaldehyde (Sigma-Aldrich, 25259) was directly added to the medium of the RPTEC to achieve a final concentration of 0.5%, and then the medium in the dish was swirled and incubated for 1 min in room temperature. Fixation reaction was quenched by adding glycine to a final concentration of 125 mM. Subsequently, the cells were scraped from culture dishes and centrifuged at $500 \times g$ for 5 min. Pellets were resuspended in PBS with 1% BSA and counted. The cells were centrifuged at $500 \times g$ for 5 min, and resuspended with wash buffer. 500,000 cells in 100 ul wash buffer were mixed and incubated with Concanavalin A (ConA) conjugated paramagnetic beads. Bulk human kidney CUT&RUN was performed on one healthy and one CKD sample (sample data in Source Data), Nuclei were isolated as described above, without fixation. Antibodies were added to each sample (0.5 µg of H3K27ac antibody [Epicypher, 13-0045, 1:50], H3K4me3 antibody [Epicypher, 13-0041, 1:50], or rabbit IgG negative control antibody [Epicypher, 13-0042, 1:50]). For transcription factor CUT&RUN libraries, duplicates were prepared with 1 ug of NFAT5 antibody (Thermo Fisher Scientific, PA1-023) or 1 ug of HIVEP2 antibody (Thermo Fisher Scientific, PA5-100756). The remaining steps were performed according to the manufacturer's instructions for cross-linked samples. Library preparation was performed using the NEBNext Ultra II DNA Library Prep Kit for Illumina (New England BioLabs, E7645S) with the manufacturer's instructions, including minor modifications indicated by CUTANA described above. CUT&RUN libraries were sequenced on a NovaSeq instrument (Illumina, 150 bp paired-end reads). Fastq files were trimmed with Trim Galore (Cutadapt 2.8) and aligned with Bowtie2 2.3.5.1 (parameters: --local --very-sensitive-local --no-unal --no-mixed --no-discordant --phred33 -I 10 -X 700) using hg38. Peak calling was performed using MACS2 2.2.7.1 with default parameters using samtools (1.9) and DeepTools (3.5.0). For TF CUT&RUN libraries, TFs were determined to be bound to a given gene if a binding peak overlapped the 2kbp promoter region or gene body. Analysis of overlapping peaks between snATAC-seq, RPTEC ATAC-seq, and CUT&RUN peaks was performed with GenomicRanges' findOverlaps (1.44.0).

## Partitioning heritability analysis of CREs

CREs that met the $p$-value threshold of 0.05 were sorted by the absolute value of their total predicted regulatory score, then binned into tertile peak sets. Bed files for each peak set were converted to hg19 with UCSC's liftOver (UCSC utilities 1.04.00). For each modeling algorithm, the LDSC (1.0.1) workflow was followed to partition heritability into each healthy or FR PT CRE set using the 1000G Phase 3 reference[80]. GWAS summary statistics for eGFR and CKD were downloaded from the publicly available CKDGen database and formatted with munge_sumstats.py[45].

## Jaccard index

The Jaccard index is a measure of similarity between sets. It is computed as the intersection of the two sets divided by the union:

$$\mathrm{Jaccard}(A,B) = \frac{|A \cap B|}{|A \cup B|} \quad (3)$$

Values range from 0 in the case of no overlap to 1 in the case of perfect overlap.

## Akaike information Criterion

The Akaike Information Criterion (AIC) can be used to evaluate model goodness of fit while penalizing overfitting[81]. We adapt it by calculating $\mathrm{AIC} = -2 \cdot l\left(\hat{\beta}, |, \mathrm{dataset}\right) + 2k$, where $k$ is the number of nonzero parameters for the given gene regulatory model and $l$ is the log-likelihood, calculated by summing $\sum_{i}^{n} f(y_i - \hat{y})$, where $f$ is the probability density function of the normal distribution. A smaller AIC value indicates a model with higher likelihood for a given dataset, however absolute values of AIC are meaningless without a comparison value[82]. Therefore we calculate relative likelihoods between CCAN and promoter models by calculating $\Delta\mathrm{AIC} = \mathrm{AIC}_{\mathrm{CCAN}} - \mathrm{AIC}_{\mathrm{Prom}}$, with more negative values indicating a larger likelihood of the CCAN-based model relative to the promoter-only-based model. This value can be further converted into a relative probability, $e^{\frac{\Delta\mathrm{AIC}}{2}}$.

## Motif analysis of healthy-FR PT CREs

CREs predicted to regulate differentially expressed genes between healthy and FR PT cells—selected with Seurat's FindMarkers function—were identified as healthy-promoting if they positively regulated a gene upregulated in healthy PT or negatively regulated a gene upregulated in FR-PT, and failed repair-promoting if otherwise. Enriched motifs were identified with Signac's FindMotifs for each CRE set. Footprinting analysis was performed on the subset of accessible peaks that were either healthy or FR PT CREs using a modified function adapted from Signac's footprint functions. We downloaded a previously compiled list of hypermethylated regions associated with CKD[11], added a flanking 1 kb window, and identified overlapping peaks with GenomicRanges' findOverlaps using RENIN-predicted CREs and all 193,787 peaks called in the multiome dataset.

## Motif enrichment-based gene regulatory network modeling

The first CRE-gene modeling step was performed to identify CREs for each H-FR gene. We then used Signac's FindMotifs function to identify motifs that were enriched in each gene's CRE set relative to a GC-matched background peak set. Enriched motifs with Benjamini and Hochberg adjusted $p$ values < 0.05 were retained as putative TFs for the second TF-gene modeling step.

## Modeling of other datasets

In order to model TF-gene regulatory interactions with snRNA-seq datasets, we identified differentially expressed genes between the healthy and failed repair clusters in each dataset using Seurat's FindMarkers function. We then used our multiomic dataset or the Gerhardt 2023 multiomic dataset[9] to generate CRE-gene regulatory networks for H-FR differentially expressed genes. TFs with motifs within each gene's linked CRE set were aggregated, then the second modeling step was performed using expression values for TFs and target genes from the snRNA-seq dataset to generate TF-gene regulatory networks.

## siRNA knockdown of NFAT5 in RPTECs

P2 RPTECs were switched to starvation medium for 24 hours to synchronize cell cycle. RPTECs were passaged into 6-well plates at 1*10⁵ cells and 2.5 mL of REGM per well. Fifteen uL of Lipofectamine RNAiMAX (Thermo Fisher, 13778075) was diluted in 125 uL OptiMEM (Thermo Fisher, 31985070) and 2.5 uL of ON-TARGETplus SMARTpool siRNA (Horizon Discovery, L-009618-00 for NFAT5 targeting, L-003434-00 for PPARA targeting, L-008436-00 for CREB5 targeting, L-015324-00 for HIVEP2 targeting, and D-001810-10 for non-targeting negative control pool) 50uM stock was diluted in 125 uL. The diluted siRNA and diluted Lipofectamine RNAiMAX were combined and incubated for 5 minutes, then 250 ul of the siRNA-lipid complex solution was added each well. Medium was replaced at 24 hours post-transfection, then at 48 hours, cells were harvested.

## Quantitative PCR

RNA was extracted from RPTECs with RNeasy Mini Kit (Qiagen, 74104). Reverse transcription was performed on the RNA with High Capacity cDNA Reverse Transcription Kit (Thermo Fisher, 4368813) to prepare cDNA libraries. The Bio-Rad CFX96 Real-Time System was used for quantitative PCR, using the iTaq Universal SYBR Green Supermix (Bio-Rad). Expression of target genes were normalized to GAPDH expression, and the $2^{-\Delta\Delta Ct}$ method was used to analyze results. A two-sample t-test was performed to compare non-targeting and siRNA treatment groups, with a $p$ value < 0.05 determined to be statistically significant. Data are presented as mean ± standard deviation. Primer sequences used are GAPDH: Fw 5´- GACAGTCAGCCGCATCTTCT−3´; Rv 5´-GCGCCCAATACGACCAAATC−3´; NFAT5: Fw 5´- CCTAATGCCCTGAT GACTCCAC−3´; Rv 5´- GTTTGCTGAGTTGATCCAACAGAC−3´; VCAM1: Fw 5´- GATTCTGTGCCCACAGTAAGGC−3´; Rv 5´- TGGTCACAGA GCCACCTTCTTG−3´.

## RNA sequencing

Total RNA integrity of extracted RNA was determined by Agilent 4200 Tapestation. Library preparation was performed with 0.5 to 1 ug of total RNA. Ribosomal RNA was removed using RiboErase kits (Kapa Biosystems), then mRNA was fragmented in reverse transcriptase buffer, heating to 94° for 8 minutes. Reverse transcription was performed with SuperScript III RT enzyme (Life Technologies) following manufacturer instructions. Illumina sequencing adapters were ligated, dual index tags were incorporated, and fragments were sequenced on an Illumina NovaSeq 6000 using paired end reads extending 150 bases. Base calls and demultiplexing were performed with Illumina's bcl2fastq and a custom Python demultiplexing program with a maximum of one mismatch in the indexing read. Reads were aligned to the Ensembl release 101 primary assembly with STAR 2.7.9a[83] and gene counts were calculated from the number of uniquely aligned, unambiguous reads by Subread:featureCount 2.0.3[84]. For each TF of interest, DESeq2 1.32.0 was used to identify differentially expressed genes with siRNA treatment ($n = 3$ samples) versus non-targeting control treatment ($n = 3$ samples). Genes were prefiltered for normalized counts greater than 0.5 per million library reads in at least 3 of 6 samples. DESeq was run on the unnormalized gene counts for filtered genes and results were extracted with alpha = 0.05. Differentially expressed genes were identified by Benjamini and Hochberg adjusted $p$ values < 0.05.

## NFAT5 immunofluorescence

Kidney cortex from non-tumor human kidney cortex samples (sample information in Source Data) was fixed with 10% formalin overnight, embedded in paraffin, and cut at 5-μm thicknesses. Antigen retrieval in antigen unmasking solution (Vector Laboratories, H-3301-250) using was performed before staining. The sections were blocked with 1% bovine serum albumin in PBS(−) for 60 minutes at room temperature, followed by incubation with primary antibodies for NFAT5 at 1:200 dilution (Thermo Fisher Scientific, PA1-023) and biotinylated Lotus Tetragonolobus Lectin (LTL) at 1:200 dilution (Vector Laboratories, B-1325) at 4 °C overnight. Next, the sections were incubated with secondary antibodies at 1:200 dilution (Invitrogen A21206 and S21374) for 90 minutes at room temperature. Nuclei were counterstained with DAPI (4,6-diamidino-2-phenylindole) and mounted in ProLong gold antifade mountant (Thermo Fisher Scientific, P36930). Fluorescence images were captured by Nikon C2+ Eclipse confocal microscopy and processed using Nikon Elements-AR 5.42.02 and FIJI (2.9.0/1.53t).

We modified the protocol to costain for NFAT5 and VCAM1 in ESRD kidney cortex samples (sample information in Source Data). After antigen retrieval, we blocked in 2.5% normal horse serum (Vector Laboratories, MP-7401-15) for 60 minutes at room temperature. Primary antibody incubation for NFAT5 at 1:10000 dilution was performed overnight at 4 °C. After three 5 min washes in PBS(-),

incubation with anti-rabbit ImmPRESS secondary antibody reagent (Vector Laboratories, MP-7401-15) was performed for 30 minutes at room temperature, followed by three PBS(-) 5 min washes. Incubation with Tyramide-AF488 (Thermo Fisher Scientific, B40953) diluted 1:100 in 0.0015% $H_2O_2$ was then performed for 10 minutes at room temperature, followed by three PBS(-) washes. After blocking for 60 minutes at room temperature in 1% BSA, anti-VCAM1 antibody was added at 1:200 dilution (Abcam, ab134047). Secondary antibody incubation was performed for 90 minutes at room temperature at 1:200 dilution (Invitrogen, A10042 and S21374).

## Statistics and reproducibility

No statistical method was used to predetermine sample size, and no data were excluded from analysis. Experiments were not randomized, and investigators were not blinded during experiments or outcome assessment.

## Reporting summary

Further information on research design is available in the Nature Portfolio Reporting Summary linked to this article.

## Data availability

The datasets generated in this study have been deposited in GEO under accession codes GSE220289. A processed Seurat R object for the multiome dataset is available at Zenodo under record number 10444715. Previously generated datasets that were analyzed during the current study are available in GEO under accession codes GSE151302 and GSE195443. The results here are also in part based upon data generated by the Kidney Precision Medicine Project. Data accessed March 24, 2024 (https://www.kpmp.org); the Seurat object was downloaded from the KPMP repository (https://www.kpmp.org/doi-collection/10-48698-yyvc-ak78). GRCh38-2020-A-2.0.0 reference genome, released on May 3, 2021, was downloaded from 10X Genomics (https://support.10xgenomics.com/single-cell-multiome-atac-gex/software/downloads/latest). Source data are provided with this paper.

## Code availability

Package code is available in a public repository at https://www.github.com/nledru/RENIN (https://doi.org/10.5281/zenodo.10524911)[85]. Analysis code for this manuscript is available at Zenodo under record number 10444715.

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

## Acknowledgements

These experiments were funded by Seed Networks Grant CZF2019-002430 from the Chan Zuckerberg Initiative and by UC2DK126024 from the National Institutes of Health to B.D.H. and F30DK132862 from the National Institutes of Health to N.L. The authors acknowledge the Washington University Genome Technology Access Center and Center for Genome Sciences & Systems Biology for sequencing support.

## Author contributions

N.L. and B.D.H. conceived, coordinated, and designed the study. N.L., Y.M., Y.Y., H.W., and D.L. performed experiments. N.L. and P.C.W. analyzed data. S.G.T., S.S.W., A.A., and G.O. provided human samples. N.L. and B.D.H. wrote the manuscript. All authors read and approved the final manuscript.

## Competing interests

B.D.H. is a consultant for Janssen Research & Development, LLC, Pfizer and Chinook Therapeutics, holds equity in Chinook Therapeutics and grant funding from Janssen Research & Development, LLC and Pfizer; all interests are unrelated to the current work.
