## [Peer Review File · Nature Communications]

Predicting proximal tubule failed repair drivers through regularized regression analysis of single cell multiomic sequencingReviewer #1 (Remarks to the Author):

The manuscript by Ledru et al describes primarily the transcriptional/open chromatin status of human kidney cells using single cell RNA-seq and ATAC-seq. The main focus of the manuscript is to predict key transcription factors of kidney epithelia cells and of failed repair cells. The conclusions that can be drawn from the author's work are (i) that by using their modified analysis pipeline they define cell type specific regulatory elements, and (ii) predicted transcription factors.

Main concern is the lack of validation, with similar data sets and more important with experiments (alternative methods to define regulatory elements, binding of transcription factors at regulatory elements and on a functional level). The strength of the study is that the experiments were done in primary tissues from humans, which is an interesting resource for the research community. However, the same group already published the same kind of dataset in Nature communications 2021 (Reference 27), although with a different analysis pipeline. Additionally several limitations must be taken into account by using human tissue such as different genetic background, age, gender, health, etc. which might not make it the right model to predict drivers of epithelia cell state. Or in other words as also stated by the authors such drivers of kidney epithelia cell status must be conserved between species or at least most of them. Therefore by only using single cell sequencing from seven human kidney specimens might not capture the real picture. This might not be the right dataset to validate a 'new' bioinformatics tool. Further, several significant weaknesses of the manuscript are that it is entirely descriptive, predictive, not validated in other data sets and with alternative methods and very limited functional validation of one predicted transcription factor.

Specific comments:

1. It seems that the TF prediction is driven in a higher proportion by RNA-seq compared to ATAC-seq (Figure 4). Is this due to the fact that RENIN is derived from SCENIC. Did the authors also use HOMER for motif analysis – how does it compare?
2. At the moment, it is a hypothesis that the identified FR-PTs in 'healthy' human kidneys are indeed failed repair cells and more important are drivers of disease progression. Can you find such cells in very young healthy kidneys? I know its very hard to come by such tissue especially from an ethical point of view, but in kidney transplantation it might be possible or tumor nephrectomy.
3. HNF1B, a very important transcription factor in kidney epithelia cells is not coming up in the TF prediction (Figure 4). Is it due to the analysis, which is not appreciating the fact that there are transcription factors who are working in several types of kidney epithelia cells and/or in combination with a different group of TFs in each cell type. It seems the focus is only on unique TFs in each cell type.
4. The NFAT5 siRNA experiment does not prove that NFAT5 promotes an in-vitro phenotype of FR-PTCs. I was not aware that RPTECs are a model of FR-PTCs. Is NFAT5 active / binds at failed repair genes / regulatory elements near failed repair genes. With the siRNA approach, 25% of NFAT5 expression is left. Is this enough to prevent binding of NFAT5 at regulatory elements. Are any known NFAT5 target genes down-regulated by the siRNA exp. Is there a binding motif of NFAT5 near VCAM1. Figure 7e: what about co-staining with VCAM1, KIM1 and NFAT5. Overexpression NFAT5 in RPTECs?
5. The title is misleading, it does not indicate that the authors used only kidney tissue and data from scRNA-seq and scATAC-seq
6. The CUT&RUN technology should be used in kidney biopsies for the cell type specific TFs. For some of the TFs antibodies are available.

Reviewer #2 (Remarks to the Author):

Description of the paper: In this work, the authors propose an approach termed Renin, a gene regulatory network algorithm to identify regulators of gene expression using multiomic data. RENIN is built on previous approaches that identify individual cis-regulatory elements and a gradient boosting-approach. RENIN has two steps. In the first one, pseudo cells are computed and gene expression is predicted by peak accessibility using elastic net regularization, here, the link between chromatin and transcription is done on the same pseudocells. The second step, uses another elastic net regularization algorithms to deconvolve what TF (binding in each regulatory

element of step 1) is regulating gene expression. The first part of the paper defines cell types in the kidney with a claim of unprecedented precision. This is due to using multiomic data and external joint analysis algorithms. Next, the paper compares doublet algorithms. Then, they use RENIN to study Kidney disease and correlate the results from the algorithm to known biology. I am not an expert on Kidney biology, so my comments on the impact on this field are minimal, just centered on algorithmic validation. They use partial nephrectomy and live donor samples to study kidney diseases and focused on FR-PTC cells (failed repair proximal tubule), which they are likely to be important clinically and they ruled out hypothesis about the origin of these cells. Next, they used CUT&RUN experiments as a validation of inferred regulatory interactions and a knock down experiment.

Findings and their significance: Authors claim two main novel ideas at play in this paper: 1) the introduction of RENIN (GRN inference algorithm) and 2) findings regarding Kidney biology. The title tries to reconcile both while the abstract stresses RENIN, as if kidney biology was less important and do not mention validation experiments as part of the results.

Regarding the computational finding, I do not believe that there is enough novelty in the work to merit publication. First, there are a plethora of gene regulatory network algorithms (scenic, CellOracle, the inferelator as being the most recent ones that harness single cell information and ATAC and/or RNA info, I refer the editors to a simple google search to highlight the many algorithms in the field). Second, many of these algorithms use scRNA and scATAC data, to create GRN estimates. Thirdly, some of the current methods even work at the single cell level (RENIN uses pseudo-cells, i.e. aggregation of cells). Fourthly, using elastic net regularization is a method widely used in the GRN field, so there is no novelty here. Fifthly, given the amount of GRN algorithms available, validation of novel algorithms should follow a more careful approach using many CUT&RUN and knock-out experiments.

Lastly, the method lacks any comparison against similar GRN inference approaches. The authors compared against LinkedPeaks and Direct-net. Even more, experiments performed in sup. Fig. 5 are not clear. How were the metrics computed? Is this computing the "intersection" of peaks identified by RENIN as regulators of gene expression and MACS2 calls from CUT&RUN experiments? What happens if there was not peak before RENIN analysis in the position of a CUT&RUN peak? If this is the case, RENIN will never be able to identify the peak as being regulated. In any case, this is correlating functional activity of peaks but not linking of TFs against genes as a GRN.

Are the authors claiming that all regulatory information in cell types is due to gene expression? That seems to be the conclusion from their analysis in figure 4. In any case, there does not seem to be new biology.

The identification of cell-type specific regulators in figure 5 is speculative, authors just show a correlation to accessibility enrichment around known motifs. Again, it is showing positive correlation, which is encouraging for the method but it is not proof of performance.

The most interesting results of the paper is given in figure 7 in which authors predict a regulatory interaction and validated it by a knockdown experiment.

I refer the authors to reconsider previous approaches like scenic, cellOracle as similar ones, and this work (Nature 2021, Genetic and epigenetic coordination of cortical interneuron development) in which, single-cell chromatin information was used to identify promoters and distal regulatory elements and applied GRN algorithms that have a built in elastic-net regularization penalty. Even more, in the work mentioned, validation of candidate TF regulating gene expression was performed by knocking out experiments and predicting resulting gene expression (not just a handful of genes (a task far more challenging than the presented on this paper)).

Comments:

1) There is an alternative hypothesis regarding the origin of FR-PTC cells, in which they exist in low number in healthy donors and they reproduce or convert when there is injury. I believe this is not proposed as a putative explanation of why these cells were found on the healthy donors' tissue.

2) The sentence:

"Of the 6,105 modeled genes, 5,038 had at least one linked CRE, suggesting that CRE accessibility is an important regulatory mechanism in human adult kidney."

If I am understanding correctly, ~1000 genes or ~16% of the genes have no CRE linked to them. Isn't this a failure of the algorithm?

3) Sentence:

"Out of the total 62,255 predicted CRE-gene links, 13,489 links had negative predicted regulatory scores, suggesting this approach may also be useful in the study of silencers."

Predicting the sign of regulation in penalized approaches is really challenging. There has to be validation to pressure that this is true.

4) Sentence:

"Predicted regulatory weights of CREs annotated as promoters (peaks within 2kbp of the target gene TSS) were higher on average than either gene body or intergenic peaks, suggesting our approach assigns quantitative regulatory scores appropriately (Fig. 3B)."

Can this be actually explain by existence of higher accessibility at promoter regions than intronic.

Clarity and context. The paper is clearly written. I believe that the paper can be summarize as a brief communication. Figure 1,2,3 can be summarize into 1 and the rest into 2.

Reviewer #3 (Remarks to the Author):

Ledru and colleagues describe in this study a novel multiome data on kidney disease tissues and a novel approach for inference of regulatory networks from multiome data. For this, it uses a multivariate model, which predict the expression by considering several CREs/TFs in parallel, and explores for the first time an adaptative elastic net model. The proposed model is compared with a few competing methods showing favorable results. However, benchmarking is limited to few competing methods and on few problems scenarios. This should be improved. The study explores a rich kidney disease multiome data and show how RENIN can be used to find a regulator of kidney injury. The paper has several interesting aspects, but focus is unclear. It sometimes focus on the computational problem or on a carefull analysis of this interesting kidney disease data set. This should also be addressed.

Major points

Authors also do an excellent job in data QC. The use of a strict doublet calling procedure is welcomed and show some troublesome results, i.e. doublet calling methods have a low agreement. The conservative author choice is a good one. I fee however that this technical aspect has poor relation with the manuscript main aim (regulatory network inference) and could go for the supplement (and be explored in futurte work).

Line 173, authors mention additional snRNA and snATAC. I could not find the snATAC-seq data. Is this one shown in the supplement. In the comparison of signatures (figure 1F and S3E) it would be best to measure the correlation between clusters of distinct modalities. Currently, it looks like that some PT populations are not fully matching between biopsy and nephrectomy samples.

While I agree that the use of multivariate / sparse models are crucial for the problem in CRE-gene link inference, the advantage of using adaptative elastic net are unclear from the text. The current explanation is highly technical. Authors should improve this motivation by using more intuitive examples of the power of adaptative elastic net.

I also wonder the impact of penalizing corelated features by the L2 term. Genes are known to be controlled by several enhancers. What is the impact of selecting λ_2 on the number of CREs/TFs. Would this impact the recovery of H3K4me3/H3K27ac peaks?

The information/motivation behind figure 3A is unclear to me.

The analysis based on H3K4me3 and H3K27ac (and Kidney related GWAS locus) is an interesting approach to validate their peak selection. However, it is currently only compared with a single competing method (Linkpeaks). There is a large number of tools for similar task in the literature, some using simple univariate models (FigureR: doi.org/10.1016/j.xgen.2022.100166; scMEGA: <https://doi.org/10.1093/bioadv/vbad003>) and some using multivariate models for CREs (SCENIC+: <https://www.biorxiv.org/content/10.1101/2022.08.19.504505v1>) and some for TF-gene associations (CellOracle: <https://doi.org/10.1038/s41586-022-05688-9>, Pandos: <https://doi.org/10.1038/s41586-022-05279-8>). Authors should improve their benchmarking to compare models approaching similar problems and discuss methods focusing on other problems.

Analysis of detected regulators TFS (Fig 4.) is welcomed, but currently only explores the use of ChromVAR, which only uses snATAC-seq information. This is not a fair comparison. It is not surprising that RENIN predicted scores correlates with TF expression, as this was used as a criterion for filtering TFs for RENIN (see line 220). Most of the methods listed above do combine gene expression in their analysis and would be more suitable here.

Line 338-339, the need to perform enrichment analysis on CRE is unclear. RENIN as a model to find gene-TF interactions, so why not focus on the RENIN based predictions described in the next section?

The use of networks methods for analyse the GRNs is interesting. It is also welcomed that authors validate their top predicted gene (NFAT5). I could not find, however, any details on how the insilico perturbation experiments were computed.

REVIEWER COMMENTS

Reviewer #1 (Remarks to the Author):

The manuscript by Ledru et al describes primarily the transcriptional/open chromatin status of human kidney cells using single cell RNA-seq and ATAC-seq. The main focus of the manuscript is to predict key transcription factors of kidney epithelia cells and of failed repair cells. The conclusions that can be drawn from the author's work are (i) that by using their modified analysis pipeline they define cell type specific regulatory elements, and (ii) predicted transcription factors.

Main concern is the lack of validation, with similar data sets and more important with experiments (alternative methods to define regulatory elements, binding of transcription factors at regulatory elements and on a functional level).

The strength of the study is that the experiments were done in primary tissues from humans, which is an interesting resource for the research community. However, the same group already published the same kind of dataset in Nature communications 2021 (Reference 27), although with a different analysis pipeline.

We agree with this comment that this dataset is similar to the one published in Ref. 27. One major difference is that our ability to predict cis-regulatory elements of genes of interest is substantially increased with simultaneous RNA and ATAC assaying from individual cells, because we can correlate chromatin accessibility directly with target gene expression, which is impossible to do with split single cell RNA- and ATAC-seq libraries. For example, with split single cell libraries, discovery of cis-chromatin interactions in Ref. 27 was performed with Cicero, which measures coaccessibility of ATAC peaks with one another. As noted in Ref. 27, correlation between Cicero gene activity and gene expression was low ($r^2 = 0.12$), so the ability to identify peaks as cis-regulatory elements of genes of interest is limited. Furthermore, as Cicero's coaccessibility metric only measures correlation of accessibility between different peaks, it has less quantitative predictive utility for cis-regulatory elements that regulate gene expression. To demonstrate this, we applied the same benchmarking approach in Figure 3 to these coaccessibility scores by summing coaccessibility scores for each peak. We assigned Cicero-linked peaks to healthy PT or FR-PT marker genes with which each peak intersected. For recall of H3K27ac peaks, Cicero peak scores achieved AUCs of 0.536 for FR marker genes and 0.552 for healthy PT marker genes. For H3K4me3, scores were 0.585 and 0.587 for FR and healthy marker genes, respectively (**New Supplementary Figure 4**).

New Supplementary Figure 4. Enrichment of CKD and eGFR partitioned heritability using other CRE-predicting methods. **a.** From left to right, ROC curve calculated for FigR, SCENIC+, Cicero, and scMEGA-predicted healthy-FR PT CREs against RPTEC H3K27ac peaks identified with CUT&RUN. **b.** From left to right, ROC curve calculated for FigR, SCENIC+, Cicero, and scMEGA-predicted healthy-FR PT CREs against RPTEC H3K4me3 peaks identified with CUT&RUN. **c.** AUCs for all tested methods against H3K27ac CUT&RUN peaks (left) and H3K4me3 CUT&RUN peaks (right). scMEGA CREs calculated along H-FR pseudotime trajectory.

We also calculated enrichment of partitioned heritability for CKD and eGFR for Cicero-predicted CREs. Here, our approach considerably outperformed Cicero, which showed limited quantitative enrichment of partitioned heritability. (Figure 3e, Supplementary Figure 6, New Supplementary Figure 7).

New Supplementary Figure 7. Enrichment of CKD and eGFR partitioned heritability using other CRE-predicting methods. a. Enrichment of partitioned heritability of CKD in model-predicted healthy (PCT + PST) and FR (failed repair—KIM1+ PT) CREs by FigR, SCENIC+, Cicero, and scMEGA. **b.** Enrichment of partitioned heritability of eGFR in model-predicted healthy (PCT + PST) and FR (failed repair—KIM1+ PT) CREs by FigR, SCENIC+, Cicero, and scMEGA.

Therefore, while Cicero and split single cell analysis are certainly very powerful computational approaches, we do find that there is considerable improvement in regulatory inference with our method, allowing for better selection of base gene regulatory networks for the second TF-gene prediction step and, as shown in Figure 3, enriching for disease heritability with the quantitative coefficients generated.

Additionally several limitations must be taken into account by using human tissue such as different genetic background, age, gender, health, etc. which might not make it the right model to predict drivers of epithelia cell state. Or in other words as also stated by the authors such drivers of kidney epithelia cell status must be conserved between species or at least most of them. Therefore by only using single cell sequencing from seven human kidney specimens might not capture the real picture. This might not be the right dataset to validate a ‘new’ bioinformatics tool.

We thank the reviewer for this important point. We agree that 7 human tissue samples may not be a sufficient sample size to support some of the claims we have made. In particular, we have decided to deemphasize the predictions of cell type-specific transcription factors, moving Figure

4 to the supplement and instead focus on our main interest, the healthy-failed repair transition, as proof of concept. Therefore, we also applied our approach to additional datasets, generated by us and other labs. First, we included additional snRNA-seq datasets generated from the human biopsies in this manuscript (n=3) and from Ref. 27 (n=5). We still performed the first CRE-gene prediction step with our multiomic dataset, as we do not have access to additional single nucleus multiomic datasets. After modeling the healthy-FR transition in this aggregated dataset, we found highly similar top TF predictions (**New Supplementary Figure 14A-B**). Eight of the ten top FR-promoting predictions were within the top 10 predictions from modeling the merged dataset: MEF2A, NFAT5, GLIS3, TCF12, KLF6, FOXP1, HIVEP2, and SOX6. We also modeled the healthy-FR transition in a snRNA-seq dataset from the Kidney Precision Medicine Project (KPMP) consisting of 27 samples (**New Supplementary Figure 14C-D**). We preserved the KPMP investigators' cell type annotations, rather than transferring cell type annotations from our datasets to preserve its utility as an independent, external dataset. Therefore we assessed whether any cell type annotations in the KPMP dataset resembled our KIM1+ PT cluster. The aPT cluster had a similar expression profile, with reduced healthy PT marker expression and increased expression of VCAM1, TPM1, and HAVCR1, so we modeled differential gene expression between the aPT cluster and other PT clusters. Here too, we found highly similar top TF predictions, with the top ten predicted FR TFs including GLIS3, TCF12, FOXP1, SOX6, NFAT5, CREB5, MEF2A, and KLF6.

New Supplementary Figure 14. Modeling H-FR transition in other human datasets.

a. Expression of healthy PT (CUBN, LRP2, SLC5A12) and FR PT (VCAM1, TPM1, HAVCR1) markers by cell type annotation in merged n=15 snRNA-seq dataset. **b.** TFs sorted by regulatory score, calculated by applying RENIN to merged n=15 snRNA-seq dataset. FR-healthy comparison was between KIM1+ PT-labeled cells and PST- or PCT-labeled cells. **c.** Expression of healthy PT (CUBN, LRP2, SLC5A12) and FR PT (VCAM1, TPM1, HAVCR1) markers by cell type annotation in merged n=27 snRNA-seq KPMP dataset. **d.** TFs sorted by regulatory score, calculated by applying RENIN to n=27 snRNA-seq KPMP dataset. FR-healthy comparison was between the aPT cluster and all other PT clusters.

For a different species, we modeled regulation of differentially expressed genes between cells annotated as healthy and injured PT in a chronic IRI mouse single cell multiomic dataset (Gerhardt 2023). Again, we preserved the investigators' original clustering and cell type annotations, modeling regulation of the differentially expressed genes between the Injured PT cluster and the PTS1-3 clusters. The Gerhardt 2023 dataset was enriched for Ki67-expressing cells after induction of injury, so we also modeled a separate snRNA-seq mouse IRI dataset (Kirita 2020). As above, we used the CRE-gene regulatory connections from the multiomic dataset modeling. We removed Control samples and modeled differential gene expression

between healthy and FR (NewPT2) clusters to enrich for post-IRI repair mechanisms. In both mouse datasets, top transcription factor predictions were highly similar to human predictions. Top predictions for the Gerhardt 2023 dataset included Glis3, Nfat5, Foxp1, Nfkb1, Pax8, Tcf12, and Klf6 (**New Supplementary Figure 15A-B**). Top predictions for the Kirita 2020 dataset included Pax8, Nfat5, Glis3, and Nfkb1 (**New Supplementary Figure 15C-D**). There were some differences that may represent repair mechanism differences in different species and post-IRI injury. For example, Pax8 was the 6th and 1st predicted FR TF in these two datasets. Additionally, Esrrg was associated with the Injured PT annotation in the Gerhardt dataset, in contrast to every other tested dataset, which may indicate that the Gerhardt 2023 Injured PT cluster, which may represent differences in subpopulation composition of different investigators' annotations.

New Supplementary Figure 15. Modeling H-FR transition in other mouse datasets.

a. Expression of healthy PT (Cubn, Lrp2, Slc5a12) and FR PT (Vcam1, Tpm1, Havcr1) markers by cell type annotation in Gerhardt 2023 multiomic dataset. **b.** TFs sorted by regulatory score, calculated by applying RENIN to Gerhardt 2023 multiomic dataset. FR-healthy comparison was between Injured PT-labeled cells and PTS1, PTS2, and PTS3-labeled cells. **c.** Expression of healthy PT (Cubn, Lrp2, Slc5a12) and FR PT (Vcam1, Tpm1, Havcr1) markers by cell type annotation in Kirita 2020 snRNA-seq dataset. **d.** TFs sorted by regulatory score, calculated by applying RENIN to Kirita 2020 snRNA-seq dataset. FR-healthy comparison was between the NewPT2 cluster and all other PT clusters.

We also assessed overall similarity between top predictions across all 5 modeled datasets by calculating pairwise Jaccard indexes for each dataset's top 10 predicted FR and healthy TFs. Similarity between human datasets for FR indicated that 70-80% of top 10 TFs were shared between datasets (**New Supplementary Figure 16A**). For healthy-promoted TF predictions, 70-90% of the top 10 TFs were shared (**New Supplementary Figure 16B**). Similarity between human and mouse IRI datasets was lower, likely due to different mechanisms in different species and modes of injury. We also counted the number of times any top 10 TF was predicted in any of the 5 modeled datasets. Twelve of 24 FR-associated TFs were top TFs in at least 2 datasets. (**New Supplementary Figure 16C**). Notably, NFAT5 and GLIS3 were top predicted TFs in all 5 datasets. Fifteen of 21 healthy-promoting TFs were top TFs for at least 3 datasets, and HNF4A and PPARA were top predictions in all 5 datasets, which is consistent with their known functions in the proximal tubule (**New Supplementary Figure 16D**).

New Supplementary Figure 16. Comparison of predictions between five modeled datasets. a. Matrix of Jaccard indexes calculated between the top ten predicted FR-associated TFs for each dataset. **b.** Matrix of Jaccard indexes calculated between the top ten predicted healthy-associated TFs for each dataset. **c.** Frequency of prediction as a top ten FR-associated TF for all TFs in at least one top 10 FR list. TFs that were top FR predictions in all 5 modeled datasets are GLIS3 and NFAT5. **d.** Frequency of prediction as a top ten healthy-associated TF for all TFs in at least one top 10 healthy list. TFs that were top healthy predictions in all 5 modeled datasets are HNF4A and PPARA.

For the modeling of each of these additional datasets, we identified differentially expressed genes between healthy and FR PT clusters de novo, so that TF rankings were not biased by expression patterns specific to our n=7 dataset. We also preserved the original clustering and cell type annotations of each dataset in order to minimize bias associated with our dataset or preprocessing pipeline. Therefore, although a different human single cell multiomic dataset would be the best approach to assessing external validity of our predictions here, we believe that the consistency in top TF predictions across datasets, species, and chronic versus acute modes of injury is encouraging and increases the likelihood that some of these repeatedly predicted TFs are relevant to the healthy-failed repair transition. We have also commented on the limitations of the human dataset we generated with the lack of additional single cell multiomic datasets for more robust validation of both modeling steps.

Further, several significant weaknesses of the manuscript are that it is entirely descriptive, predictive, not validated in other data sets and with alternative methods and very limited functional validation of one predicted transcription factor.

Thank you for these suggestions for improvement. In addition to validation in other datasets, we also add siRNA knockdown experiments of other predicted transcription factors HIVEP2, CREB5, and PPARA. We also expand our validation of NFAT5 and HIVEP2 with CUT&RUN performed on RPTEC cultures. We calculated the fraction of predicted target genes for each TF that are bound by NFAT5 or HIVEP2, measured by NFAT5 CUT&RUN-seq performed on RPTECs. We find the five tested methods have similar accuracies, but RENIN recovers more targets (RENIN – 379/445, Pando – 104/116, CellOracle – 45/56, SCENIC+ 274/305, FigR – 31/40) (**New Figure 5C**). For HIVEP2, only RENIN and Pando predicted any targets (RENIN – 368/439, Pando – 70/85, CellOracle – 0/0, SCENIC+ 0/0, FigR 0/0) (**New Supplementary Figure 23B**). For regression-based methods, we also assessed each method's ability to predict gene expression in an independent dataset (KPMP, n=27 snRNA-seq) after being trained on our multiome dataset. Here we found that RENIN had competitive performance as well, with mean r^2 of .080, relative to .065 for Pando and .016 for CellOracle (**New Figure 4E-G**). It is very likely that different methods will have better or worse performance in different contexts, however, for the focus of our work, the design of our approach appears to be the best option among available methods.

New Figure 4E-G. Benchmarking model TF-gene regulatory predictions . e. r^2 calculated for RENIN- and Pando-predicted H-FR gene expression compared to target gene expression in an independent KPMP snRNA-seq dataset for genes that were successfully modeled by both methods. For shared genes, mean RENIN r^2 was .080 and mean Pando r^2 was .065. **f. r^2** calculated for RENIN- and CellOracle-predicted H-FR gene expression compared to target gene expression in an independent KPMP snRNA-seq dataset for genes that were successfully modeled by both methods. For shared genes, mean RENIN r^2 was .055 and mean CellOracle r^2 was .016. **g.** Number of H-FR differentially expressed genes modeled by each method.

We also attempted CUT&RUN for CREB5 and PPARA. We were unsuccessful at generating libraries with CREB5 (Abcam, ab277098). We were able to generate and submit libraries for PPARA CUT&RUN (ThermoFisher, MA1-822). However, we recovered significantly fewer peaks (1,864 PPARA peaks versus 48,671 NFAT5 and 28,822 HIVEP2 peaks) and all models performed extremely poorly on prediction (**New Figure R1**). Given PPARA's well known role in the proximal tubule and failed validation across all tested methods, this may represent low PPARA activity in the context of a FR phenotype in RPTEC culture.

New Figure R1. Model predictions are not validated by PPARA CUT&RUN in RPTEC culture. Number of predicted PPARA targets by each method, separated into target genes that were bound versus unbound on PPARA CUT&RUN-seq. 2/472 RENIN-predicted targets, 0/82 Pando-predicted targets, 0/78 CellOracle-predicted targets, 3/509 SCENIC+-predicted targets, and 0/40 FigR-predicted targets were bound by PPARA.

Specific comments:

1. It seems that the TF prediction is driven in a higher proportion by RNA-seq compared to ATAC-seq (Figure 4). Is this due to the fact that RENIN is derived from SCENIC. Did the authors also use HOMER for motif analysis – how does it compare?

Thank you for this comment and suggestion. Many tools—such as HOMER, chromVAR, and SCENIC+—use motif enrichment to identify active TFs of interest, for example in different cell types. While this is certainly a powerful approach, particularly in developmental contexts, there may be active TFs that do not rely on enriched concentrations of conjugate motifs to regulate target genes, particularly for disease processes that may not undergo positive natural selection.

We adapted the HOMER approach to select for enriched, rather than all, motifs present in a gene's predicted set of CREs, relative to a GC-matched background peak set. Of the 1,516 modeled genes with at least one predicted CRE, 660 were not able to be modeled, which is consistent with the usage of the more stringent requirement of motif enrichment rather than simply presence. Despite fewer predicted TF-gene regulatory interactions, the top predicted TFs on successfully modeled genes were very similar to our original list: the top five FR TFs were the same between rankings with only differences in ordering and four of five H TFs were shared between rankings. (New Supplementary Figure 13, New Supplementary Data 6). While overall regulatory element prioritization appears consistent, there were considerably fewer TF-gene links. For example, motif enrichment-based modeling predicted 38 NFAT5 target genes, of which 0 were bound by NFAT5 in our CUT&RUN dataset. Since we applied a regularized regression specifically to reduce the number of linked CREs in order to minimize false positive

predictions, motif enrichment for TF selection may be more effective with other modeling approaches that are less penalized, resulting in greater numbers of putative linked regulatory elements.

New Supplementary Figure 13. RENIN predictions using motif enrichment. **a.** TFs sorted by predicted regulatory score. For each modeled H-FR differentially expressed gene, TFs with significantly enriched motifs within its set of linked CREs were identified and used for regression modeling. **b.** Overlap of unique predicted TF-gene links between the two modeling configurations. For the All motifs configuration, any motif appearing at least once in a gene's linked CREs is considered. For the Enriched motifs only configuration, only TFs with significantly enriched motifs are considered.

2. At the moment, it is a hypothesis that the identified FR-PTs in 'healthy' human kidneys are indeed failed repair cells and more important are drivers of disease progression. Can you find such cells in very young healthy kidneys? I know its very hard to come by such tissue especially from an ethical point of view, but in kidney transplantation it might be possible or tumor nephrectomy.

We thank the reviewer for this suggestion and agree that this is an important remaining question. We unfortunately do not have any very young, healthy kidneys, but have attempted to explore this question in related samples. In a single cell multiomic dataset prepared from kidney organoids (Yoshimura 2023), VCAM1 expression in developing tubule populations was nonexistent (**Figure R2, for reviewing purposes only**). Similarly, in a human 17 week fetal kidney dataset (Lindström 2018), tubular precursors also did not demonstrate VCAM1 expression.

Figure R2. VCAM1 expression in young kidney samples. a. UMAP plot showing no VCAM1 expression in developing tubule lineages (dev_TUB_1 and dev_TUB_2) nor in PT cells. (Yoshimura 2023) **b.** Image taken from KIT (<http://humphreyslab.com/SingleCell>) showing no/very low VCAM1 expression in tubular precursor cells (cluster 15) in human 17-week fetal kidney snRNA-seq dataset (Lindström 2018).

In mice, VCAM1+ PT cells in uninjured kidneys of 3-month-old mice were also extremely sparse or not present, relative to the number of VCAM1+ proximal tubule cells in 18-month-old mice (Gerhardt 2019). While these findings support our hypothesis that VCAM1+ populations are associated with or drivers of disease progression, we cannot conclusively rule out alternative hypotheses, for example, that PAX2+ intratubular progenitor populations can restore tubular populations following injury. We have added discussion that alternative hypotheses such as this one can explain the presence of FR populations in ‘healthy’ kidneys. We did not predict PAX2 to have much effect on the differentially expressed genes we modeled, so other datasets may be needed to better study PAX2 in tubule repair.

3. HNF1B, a very important transcription factor in kidney epithelia cells is not coming up in the TF prediction (Figure 4). Is it due to the analysis, which is not appreciating the fact that there are

transcription factors who are working in several types of kidney epithelia cells and/or in combination with a different group of TFs in each cell type. It seems the focus is only on unique TFs in each cell type.

We thank the reviewer for this important point. We were trying to specifically highlight cell type-specific TFs with this Figure, so our selection of TFs to include did not include HNF1B. We appreciate the point the reviewer has made that many important TFs are not cell type-specific. In order to demonstrate that our approach is not limited to only cell type-specific TF prediction, we have also added examples of top glomerular and tubular TF predictions, including HNF1B, ranked by summed normalized scores across the POD, PEC, ENDO, MES, and FIB clusters and the PCT, PST, LH, DCT, CNT, PC, ICA, and ICB clusters, respectively.

New Supplementary Figure 9B. Tubular and glomerular TF prediction. Regulatory scores calculated by summing normalized cell type RENIN scores for tubular or glomerular cell type marker gene set and multiplying by mean expression of that TF in the given cell type. Selected TFs for plotting identified by summing RENIN scores across tubular (PCT, PST, LH, DCT, CNT, PC, ICA, ICB) or glomerular (POD, PEC, ENDO,

For this reason and other comments by this reviewer and others, we have decided to deemphasize Figure 4, as we do not believe we can validate and/or benchmark all of the predicted cell-type-specific TFs sufficiently for this paper and instead wish to tighten our focus on the failed repair PT phenotype. We believe that our validation of failed repair PT drivers demonstrates proof of concept that our approach, as with other methods, may be useful to identify cell type-specific TFs with subsequent validation.

4. The NFAT5 siRNA experiment does not prove that NFAT5 promotes an in-vitro phenotype of FR-PTCs. I was not aware that RPTECs are a model of FR-PTCs. Is NFAT5 active / binds at failed repair genes / regulatory elements near failed repair genes.

We have observed that RPTECs have low expression of PT marker genes such as SLC5A1, SLC5A2, SLC5A12, CUBN, LRP2, and HNF4A and expression of inflammatory genes that are not expressed in healthy PT, such as VCAM1 (Ref. 27). We therefore think in vitro culture may induce a partial injury/dedifferentiation phenotype. NFAT5 is predicted to regulate 445 FR genes, so our modeling predicts it plays a central role. We performed CUT&RUN for NFAT5, finding that it binds 85% of predicted NFAT5 target failed repair genes (**New Figure 5C**), including EGR1, HIVEP2, TGFBR2, NF1, ITGB1, YAP1, LAMC1, COBLL1, IL4R, TNFSF10, DAPK1, PXDN, and TPM1. TPM1 has previously been identified as FR-associated genes in Ref. 27. A similar result was seen for another predicted FR TF we profiled for additional validation, HIVEP2 (**New Supplementary Figure 22B**).

Figure 5C-D. c. Number of predicted NFAT5 targets by each method, separated into target genes that were bound versus unbound on NFAT5 CUT&RUN-seq. 379/445 RENIN-predicted targets, 104/116 Pando-predicted targets, 45/56 CellOracle-predicted targets, 274/305 SCENIC+-predicted targets, and 31/40 FigR-predicted targets were bound by NFAT5 assessed by CUT&RUN-seq on RPTEC culture. **d.** Distal predicted CRE for TPM1, predicted to be NFAT5 target FR gene and downregulated with siRNA NFAT5 knockdown, bound by NFAT5.

Supplementary Figure 22B. Number of predicted HIVEP2 targets by each method, separated into target genes that were bound versus unbound on HIVEP2 CUT&RUN-seq. 368/439 RENIN-predicted targets and 70/85 Pando-predicted targets were bound by HIVEP2 assessed by CUT&RUN-seq on RPTEC culture. SCENIC+ does not predict any H-FR genes to be HIVEP2 targets. CellOracle and FigR do not use motif databases that include HIVEP2 motif necessary for modeling.

With the siRNA approach, 25% of NFAT5 expression is left. Is this enough to prevent binding of NFAT5 at regulatory elements. Are any known NFAT5 target genes down-regulated by the siRNA exp.

We thank the reviewer for their careful review of our data and this point. We have reanalyzed the RNA-seq data from this experiment for evidence that the moderate knockdown of NFAT5 we achieved could reduce NFAT5 binding at target regulatory elements. NFAT5 has a well-described role in the literature in protecting against hypertonicity. NFAT5 targets that mediate an osmoprotective response that have been reported in the literature include SLC6A12 (BGT1), AQP1, and RNF183, which have significant decreases in expression with siRNA knockdown. We believe the knockdown of these targets demonstrate that NFAT5 regulatory activity has been reduced with siRNA knockdown.

New Supplementary Figure 19. Known targets of NFAT5 are significantly downregulated with NFAT5 knockdown in RPTEC culture. SLC6A12 (BGT1), RNF183, and AQP1 are significantly downregulated with NFAT5 siRNA treatment. Adjusted p values: 1.293e-3, 1.241e-14, and 3.525e-02, respectively.

Is there a binding motif of NFAT5 near VCAM1.

We do not detect any motifs of NFAT5 near VCAM1, and NFAT5 is not predicted by our modeling to drive VCAM1 expression. Our modeling predicts 16 TFs that contribute to VCAM1 regulation: BACH2, HIVEP1, ARID5A, YBX1, PKNOX2, NFKB2, RELB, NFKB1, ZBTB47, BCL6, ZEB1, IRF8, IRF1, FOSL2, FOSL1, TGIF1. Of these 16, NFAT5 is predicted to regulate NFKB1, an interaction that has been described in the literature.

We think that this supports the role of NFAT5 in driving an inflammatory in vitro phenotype partially resembling the in vivo failed repair phenotype. Direct NFAT5-VCAM1 regulation would confound our ability to use VCAM1 expression as a marker for overall failed repair phenotype. VCAM1 expression being indirectly downregulated by NFAT5 knockdown, suggests NFAT5 target genes include inflammatory mediators that lead to VCAM1 expression.

Our modeling predicts that NFAT5 positively regulates TPM1 expression, another FR-associated gene described in Ref. 27. We found that TPM1 was downregulated with siRNA treatment and was also bound by NFAT5, measured by CUT&RUN (**New Figure 5D**).

Figure 7e: what about co-staining with VCAM1, KIM1 and NFAT5.

We thank the reviewer for this suggestion. We chose to co-stain for NFAT5 and VCAM1, because VCAM1 is a more specific marker for failed repair PT, instead of injured PT, of which the vast majority undergoes successful repair. In an ESRD sample, we found many low LTL+ proximal tubules expressing both NFAT5 and VCAM1 (**New Supplementary Figure 24**). This suggests that NFAT5 expression in proximal tubules is associated with the FR state.

New Supplementary Figure 24. VCAM1 and NFAT5 colocalization in ESRD kidney. Immunofluorescent labeling of NFAT5 (green) and VCAM1 (red) in adult human kidney. DAPI (blue) is a nucleus marker and LTL (white) is a proximal tubule marker. Arrows denote examples of NFAT+, VCAM1+ proximal tubules. Scale bars are 50 μ m in length.

Overexpression NFAT5 in RPTECs?

Our RNA-seq showed that NFAT5 is highly expressed in RPTECs, consistent with both its role in promoting FR and the FR-like phenotype of RPTECs cultured on plastic. We stained RPTEC culture for NFAT5, finding expression in every cell imaged (**Figure R3, for review only**).

Therefore we do not believe it is likely that overexpression of NFAT5 in RPTECs would produce stronger FR/VCAM1 expression.

Figure R3. NFAT5 expression in RPTECs. Immunofluorescent labeling of NFAT5 (green) expression in RPTECs. DAPI (blue) is a nucleus marker. Scale bar is 100 μm in length.

5. The title is misleading, it does not indicate that the authors used only kidney tissue and data from scRNA-seq and scATAC-seq

Thank you for this point, we realize that the title was incorrect. We have changed it to mention specifically the proximal tubule. The data was obtained from simultaneous (multiomic) scRNA- and ATAC-seq profiling. We have also clarified the description of our dataset in the abstract and introduction.

6. The CUT&RUN technology should be used in kidney biopsies for the cell type specific TFs. For some of the TFs antibodies are available.

We thank the reviewer for the suggestion to use CUT&RUN to validate TF predictions. In the interest of focusing on cell type- and state-specific TF activity and tightening the focus of our manuscript, we performed CUT&RUN for NFAT5, HIVEP2, CREB5, and PPARA in RPTECs to validate our model's predictions for the H-FR transition. We found that 0.5% fixation substantially outperformed no fixation in number of MACS2-called peaks. NFAT5 and PPARA were predicted to be cell type-specific TFs as well. However, when we attempted CUT&RUN on human kidney biopsies, we found the 0.5% fixation to limit our recovery of nuclei for these antibodies and were unsuccessful at generating libraries. We instead focus on validation of PT state-associated transcription factors with our RPTEC CUT&RUN as proof-of-concept that such an approach can be considered for cell type-specific TF prediction with validation still required.

Reviewer #2 (Remarks to the Author):

Description of the paper: In this work, the authors propose an approach termed Renin, a gene regulatory network algorithm to identify regulators of gene expression using multiomic data. RENIN is built on previous approaches that identify individual cis-regulatory elements and a gradient boosting-approach. RENIN has two steps. In the first one, pseudo cells are computed

and gene expression is predicted by peak accessibility using elastic net regularization, here, the link between chromatin and transcription is done on the same pseudocells. The second step, uses another elastic net regularization algorithms to deconvolve what TF (binding in each regulatory element of step 1) is regulating gene expression. The first part of the paper defines cell types in the kidney with a claim of unprecedented precision. This is due to using multiomic data and external joint analysis algorithms. Next, the paper compares doublet algorithms. Then, they use RENIN to study Kidney disease and correlate the results from the algorithm to known biology. I am not an expert on Kidney biology, so my comments on the impact on this field are minimal, just centered on algorithmic validation. They use partial nephrectomy and live donor samples to study kidney diseases and focused on FR-PTC cells (failed repair proximal tubule), which they are likely to be important clinically and they ruled out hypothesis about the origin of this cells. Next, they used CUT&RUN experiments as a validation of inferred regulatory interactions and a knock down experiment.

Findings and their significance: Authors claim two main novel ideas at play in this paper: 1)the introduction of RENIN (GRN inference algorithm) and 2) findings regarding Kidney biology. The title tries to reconcile both while the abstract stresses RENIN, as if kidney biology was less important and do not mention validation experiments as part of the results.

We thank the reviewer for their careful reading of our manuscript and suggestion on focus. We have edited the title, abstract, and introduction to shift focus to the kidney biology.

Regarding the computational finding, I do not believe that there is enough novelty in the work to merit publication. First, there are a plethora of gene regulatory network algorithms (scenic, Celloracle, the inferelator as being the most recent ones that harness single cell information and ATAC and/or RNA info, I refer the editors to a simple google search to highlight the many algorithms in the field). Second, many of these algorithms use scRNA and scATAC data, to create GRN estimates.

We thank the reviewer for this comment, and agree that there are many other great methods beyond the ones we mentioned in this manuscript. We do not believe ours, or any single method, is uniformly better across all gene regulatory network applications. We were specifically interested in predicting the most important cis- and trans-regulatory elements in the healthy-failed repair transition and believe the configuration we developed was best suited for it, as demonstrated by new benchmarking results. We also add a discussion on other methods that address similar and different, but related problems in gene regulatory network modeling.

Thirdly, some of the current methods even work at the single cell level (RENIN uses pseudo-cells, i.e. aggregation of cells).

We appreciate this point and have carefully considered the risks of losing the advantages of single cell resolution with pseudocells. All of the mentioned algorithms, and the approach taken in this manuscript, have the theoretical capacity to run at single cell resolution. However we believe the use of pseudocells or another common tactic, imputation, in single cell datasets is a strength rather than a weakness. The benefits of pseudocell aggregation or another common tactic, imputation, include considerably improved computational tractability, lowering required

RAM and runtime, and reduced noise due to sparsity, therefore improving our ability to discover CRE- and TF-gene regulatory connections. For example, CellOracle uses Cicero for the construction of CRE-gene networks, which bins similar cells to reduce the effects of scATAC-seq sparsity on Cicero's ability to discover coaccessible peaks. SCENIC also performs pseudobulking for its regulon detection step. For this manuscript, we decided to update VISION's micro-pooling algorithm for multiomic data to generate pseudocells because pooling cells only with nearby neighbors, rather than random cells of the same cell type, can reduce the effects of sparsity and dropout while preserving the heterogeneity of the single cell dataset.

An alternative approach to address sparsity is imputation, which SCENIC+ uses during processing of the scATAC modality. We decided that pseudocells are more flexible in allowing different degrees of smoothing, rather than a binary decision on imputation. Additionally, as we used a regression approach, we thought the reduction in n when using pseudocells more accurately represented the uncertainty introduced by the sparsity of the single cell dataset, instead of artificially deflating the variances of the estimates of regulatory coefficients. As we bootstrap our approximations of regulatory coefficients, this allows us to maintain accurate regulatory element-gene link identification.

In order to illustrate these points, we have added Supplementary Figures showing predictions using single cells and target bin sizes of 5 and 10. Although the order changes slightly, top TF predictions remain identical through the top 5 H and FR TFs, and highly similar through the top 20, suggesting our approach to pseudocell binning preserves variation present in single cell datasets (**New Supplementary Figure 12, New Supplementary Dataset 5**). We have updated **New Figure 5** using the modeling results without the use of pseudocells. Because the use of a target bin size of 10 precipitously drops computational time and RAM requirements, we have continued using a bin size of 10 when testing different settings and have clarified that we have done so. We have also updated the manuscript to discuss the advantages and disadvantages of pseudocells,

Supplementary Figure 12. Similarity of TF predictions using different sizes of pseudocells. a-b. Matrix of pairwise Jaccard indexes calculated for the top 20 model predicted (a) FR and (b) H TFs using target bin sizes of 1 (no pseudocell binning), 5, 10.

Fourthly, using elastic net regularization is a method widely used in the GRN field, so there is no novelty here.

We recognize that an elastic-net is a commonly used regularized estimator and we agree that the simple change from elastic-net to adaptive elastic-net is not a considerable advance on its own. We have added details on additional specifications we have made, including the careful use of pseudocells as detailed above, treatment of ATAC data as non-binary, bootstrapping, cross-validation, and split CRE-gene / TF-gene modeling steps on the full dataset versus PT subset that allows us to focus on understanding the differential expression between the different PT states.

Fifthly, given the amount of GRN algorithms available, validation of novel algorithms should follow a more careful approach using many CUT&RUN and knock-out experiments.

We appreciate this suggestion. We have added additional knock-out experiments for CREB5, HIVEP2, and PPARA. We also added CUT&RUN experiments for NFAT5 and HIVEP2, which we used to compare other methods' performance.

Lastly, the method lacks any comparison against similar GRN inference approaches. The authors compared against LinkedPeaks and Direct-net. Even more, experiments performed in sup. Figure 5 are not clear. How were the metrics computed? Is this computing the "intersection" of peaks identified by RENIN as regulators of gene expression and MACS2 calls from CUT&RUN experiments? What happens if there was not peak before RENIN analysis in the position of a

CUT&RUN peak ? If this is the case, RENIN will never be able to identify the peak as being regulated. In any case, this is correlating functional activity of peaks but not linking of TFs against genes as a GRN.

We thank the reviewer for this point. We have clarified that this Figure was intended only to test CRE linking to genes, and not TFs. We have performed additional benchmarking with newly generated NFAT5 and HIVEP2 CUT&RUN-seq data from RPTECs. We agree that our method, and any other single cell modeling method, is limited only to peaks that are called in the single cell dataset, which is why we tested for recall of histone modification peaks. Furthermore, we have only attempted to predict CREs of a fraction of the expressed genes in our dataset, proximal tubule cluster marker genes, instead of all genes expressed in the proximal tubule. Therefore, we use the called peaks from CUT&RUN experiments as a ground-truth dataset, in order to validate the proximal tubule marker gene-specific predictions we attempted to make.

Are the authors claiming that all regulatory information in cell types is due to gene expression ? That seems to be the conclusion from their analysis in Figure 4. In any case, there does not seem to be new biology.

We thank the reviewer for raising this point and recognize that we designed this Figure poorly. The intention was to show that RENIN incorporated both ATAC and RNA information. In the original Figure 4D, we show that regulatory information is not simply due to high gene expression, as measured by Jaccard index. In order to clarify the focus of the manuscript and to remove this figure as attempted validation of our method, we have move it to the supplement and explained that this is only an example of one use case.

The identification of cell-type specific regulators in Figure 5 is speculative, authors just show a correlation to accessibility enrichment around known motifs. Again, it is showing positive correlation, which is encouraging for the method but it is not proof of performance.

We have deemphasized Figure 5 to focus on more rigorous validation with siRNA and CUT&Run experiments.

The most interesting results of the paper is given in Figure 7 in which authors predict a regulatory interaction and validated it by a knockdown experiment.

We thank the reviewer for this point and have extended this approach to other predictions.

I refer the authors to reconsider previous approaches like scenic, cellOracle as similar ones, and this work (Nature 2021, Genetic and epigenetic coordination of cortical interneuron development) in which, single-cell chromatin information was used to identify promoters and distal regulatory elements and applied GRN algorithms that have a built in elastic-net regularization penalty.

Even more, in the work mentioned, validation of candidate TF regulating gene expression was performed by knocking out experiments and predicting resulting gene expression (not just a handful of genes (a task far more challenging than the presented on this paper).

We have added similar analysis. We calculated the proportion of targets that were significantly downregulated with siRNA knockdown for each TF (**New Supplementary Figure 22A**). These showed ~20-35% of targets for three of our top predicted FR TFs were significantly downregulated as predicted. This may not be directly comparable to the citation provided, given differences in the *in vitro* and adult disease contexts, but are consistent with the RPTEC phenotype being driven more by FR TFs than healthy PT TFs such as PPARA, which had fewer predicted targets downregulated with siRNA knockdown.

Supplementary Fig. 22. Proportions of predicted TF targets that are experimentally validated by siRNA knockdown and CUT&RUN. a. NFAT, HIVEP2, CREB5, and PPARA predicted targets (light purple) and the proportion that is significantly downregulated (adjusted p value < 0.05) with siRNA knockdown in RPTECs (dark purple).

We also performed CUT&RUN as a more direct validation of predicted targets. We were unsuccessful at generating CREB5 and PPARA CUT&RUN libraries, but could do so for NFAT5 and HIVEP2. (**New Figure 5C, Supplementary Figure 22B**). Here we found that ~85% of predicted targets were bound by either TF, which indicates NFAT5 and HIVEP2 have the potential to regulate predicted target genes in the right biological context.

Figure 5C. Number of predicted NFAT5 targets by each method, separated into target genes that were bound versus unbound on NFAT5 CUT&RUN-seq. 379/445 RENIN-predicted targets, 104/116 Pando-predicted targets, 45/56 CellOracle-predicted targets, 274/305 SCENIC+-predicted targets, and 31/40 FigR-predicted targets were bound by NFAT5 assessed by CUT&RUN-seq on RPTEC culture.

Supplementary Figure 22B. Number of predicted HIVEP2 targets by each method, separated into target genes that were bound versus unbound on HIVEP2 CUT&RUN-seq. 368/439 RENIN-predicted targets and 70/85 Pando-predicted targets were bound by HIVEP2 assessed by CUT&RUN-seq on RPTEC culture. SCENIC+ does not predict any H-FR genes to be HIVEP2 targets. CellOracle and FigR do not use motif databases that include HIVEP2 motif necessary for modeling.

Comments:

1) There is an alternative hypothesis regarding the origin of FR-PTC cells, in which they exist in low number in healthy donors and they reproduce or convert when there is injury. I believe this is not proposed as a putative explanation of why these cells were found on the healthy donors' tissue.

We thank the reviewer for pointing out this omission. We have added discussion this hypothesis. We did not predict PAX2 to have much effect on the differentially expressed genes we modeled, so other datasets may be needed to better study PAX2 in tubule repair.

2) The sentence:

“Of the 6,105 modeled genes, 5,038 had at least one linked CRE, suggesting that CRE accessibility is an important regulatory mechanism in human adult kidney.”

If I am understanding correctly, ~1000 genes or ~16% of the genes have no CRE linked to them. Isn't this a failure of the algorithm?

We certainly do not propose to have solved all of gene regulation in the kidney with one dataset and a regression model. Sparsity of single cell datasets, particularly for scATAC-seq datasets, and low mRNA expression or representation of genes hinder our ability to explain all of gene expression. As part of the added comparisons this reviewer and others recommended, we have also assessed other models' ability to identify CRE-gene links. Cicero (CellOracle's method for

CRE-gene identification) identified CREs for 4,564 of the 6,105 genes. LinkPeaks identified CREs for 1,277 of these genes. DIRECT-NET identified CREs for 5212 genes.

In addition to other benchmarking comparisons we have added, we also counted the number of CREs and modeled genes for the other methods we added. These methods take slightly different approaches. For example, scMEGA's approach is to model the top 10% most variable genes. It identified 18,408 CREs, and 2,652 of the 2,975 most variable genes had at least one linked CRE (and 5,108 of the top 20% most variable genes). FigR and SCENIC+ model all 29,758 genes in the dataset. FigR linked 80,127 CREs to 16,262 of these genes. SCENIC+ selected 37,362 CREs, linked to 10,713 genes.

These numbers are not fully comparable, as successful CRE linking is more likely for more variable genes. However, we think these comparisons indicate our model's performance is not necessarily a failure and is competitive with other approaches. We would also like to add that raw number of CRE-gene links is not necessarily indicative of good performance. For example, our bootstrapping approach to attempt to eliminate some false positives reduces the final total of CRE-gene links reported. On the other hand, investigators interested in identifying a maximal number of CREs may find DIRECT-NET or other similar methods to be more useful, at the expense of increased risk of overfitting with the XGBoost model, and diminishing enrichment of regulatory significance (Figure 3).

3) Sentence:

“Out of the total 62,255 predicted CRE-gene links, 13,489 links had negative predicted regulatory scores, suggesting this approach may also be useful in the study of silencers.” Predicting the sign of regulation in penalized approaches is really challenging. There has to be validation to pressure that this is true.

We thank the reviewer for their close reading and this comment. We agree and have removed the speculative sentence.

4) Sentence:

“Predicted regulatory weights of CREs annotated as promoters (peaks within 2kbp of the target gene TSS) were higher on average than either gene body or intergenic peaks, suggesting our approach assigns quantitative regulatory scores appropriately (Figure 3B).”

Can this be actually explain by existence of higher accessibility at promoter regions than intronic.

We thank the reviewer for this point, and we have added this as a possible interpretation. However higher promoter accessibility uniformly across all cells in the dataset would not necessarily always lead to a higher score if it was not correlated with gene expression.

Clarity and context. The paper is clearly written. I believe that the paper can be summarize as a brief communication. Figure 1,2,3 can be summarize into 1 and the rest into 2.

We appreciate the positive remark and suggestions on making the manuscript more concise. Thanks to this and other comments, we have reduced the number of main figures from 7 to 5.

Reviewer #3 (Remarks to the Author):

Ledru and colleagues describe in this study a novel multiome data on kidney disease tissues and a novel approach for inference of regulatory networks from multiome data. For this, it uses a multivariate model, which predict the expression by considering several CREs/TFs in parallel, and explores for the first time an adaptative elastic net model. The proposed model is compared with a few competing methods showing favorable results. However, benchmarking is limited to few competing methods and on few problems scenarios. This should be improved. The study explores a rich kidney disease multiome data and show how RENIN can be used to find a regulator of kidney injury. The paper has several interesting aspects, but focus is unclear. It sometimes focus on the computational problem or on a carefull analysis of this interesting kidney disease data set. This should also be addressed.

We thank the reviewer for their close reading and constructive criticism. Regarding the focus, we have made changes to the manuscript to tailor its focus to the study of the FR regulatory elements.

Major points

Authors also do an excellent job in data QC. The use of a strict doublet calling procedure is welcomed and show some troublesome results, i.e. doublet calling methods have a low agreement. The conservative author choice is a good one. I fee however that this technical aspect has poor relation with the manuscript main aim (regulatory network inference) and could go for the supplement (and be explored in futurte work).

We thank the reviewer for the thoughtful suggestion and agree that it muddles the main aim. We have moved the figures to **Supplementary Figure 2**.

Line 173, authors mention additional snRNA and snATAC. I could not find the snATAC-seq data. Is this one shown in the supplement. In the comparison of signatures (Figure 1F and S3E) it would be best to measure the correlation between clusters of distinct modalities. Currently, it looks like that some PT populations are not fully matching between biopsy and nephrectomy samples.

We thank the reviewer for their careful reading of our manuscript and apologize for the confusion. We have clarified that the snATAC-seq data is the one shown in the supplement. We agree that there are likely some differences between certain cell type populations in our dataset. Pearson correlations between the average expression of each cell type delineated by biopsy or nephrectomy origin were all greater than 0.98 (**Figure R4, for review only**). Due to this and our n=7 sample size, we do not feel we can make strong conclusions about differences with this multiomic dataset alone but do agree that it is likely there may be slight changes.

Figure R4. Correlation between average gene expression profiles of biopsy-derived and nephrectomy-derived cell types. Average expression of each cell type, grouped by either biopsy or nephrectomy origin was calculated. Then Pearson correlation was calculated between pairs of cell types for biopsy versus nephrectomy comparison.

While I agree that the use of multivariate / sparse models are crucial for the problem in CRE-gene link inference, the advantage of using adaptive elastic net are unclear from the text. The current explanation is highly technical. Authors should improve this motivation by using more intuitive examples of the power of adaptive elastic net.

We thank the reviewer for this suggestion on improving clarity and motivation. We move the technical explanation to the Methods and add a less technical, more intuitive explanation.

I also wonder the impact of penalizing correlated features by the L2 term. Genes are known to be controlled by several enhancers. What is the impact of selecting λ_2 on the number of CREs/TFs. Would this impact the recovery of H3K4me3/H3K27ac peaks?

We thank the reviewer for this point and agree that increasing the weighting of L2 is an interesting idea. We add **New Supplementary Figure 5** showing the results of increasing L2. As expected, the number of predicted regulatory connections increases as the relative L1 weighting decreases and model prediction accuracy decreases. We have added text explaining the expected results of changing relative L1 and L2 weighting. For example, for enrichment analysis, a lower L2 may be better, as sparse models should be more enriched for regulatory function/histone modifications/heritability.

New Supplementary Figure 5. The effects of increasing λ_2 on CRE predictions. a. Calculated AUCs for H3K27ac (purple) and H3K4me3 (green) peaks in RPTECs with FR and H (healthy PT) CRE predictions with increasing λ_2 . Number of unique CREs predicted with increasing λ_2 . Random number generator seeds and pseudocell matrices used were kept the same across trials.

The information/motivation behind Figure 3A is unclear to me.

We aim to show cell type-specific CRE distributions do indeed differ. However we appreciate the point that there is little information provided by this figure. As such, we deemphasize it and move it to the **New Supplementary Figure 8**.

The analysis based on H3K4me3 and H3K27ac (and Kidney related GWAS locus) is an interesting approach to validate their peak selection. However, it is currently only compared with a single competing method (Linkpeaks). There is a large number of tools for similar task in the literature, some using simple univariate models (FigureR: doi.org/10.1016/j.xgen.2022.100166; scMEGA: <https://doi.org/10.1093/bioadv/vbad003>) and some using multivariate models for CREs (SCENIC+: <https://www.biorxiv.org/content/10.1101/2022.08.19.504505v1>) and some for TF-gene associations (CellOracle: <https://doi.org/10.1038/s41586-022-05688-9>), Pandos: <https://doi.org/10.1038/s41586-022-05279-8>). Authors should improve their benchmarking to compare models approaching similar problems and discuss methods focusing on other problems.

We thank the reviewer for this point. We have performed the same benchmarking steps with FigureR, SCENIC+, and scMEGA. We find that our method is competitive against all other methods in both RPTEC CRE recall and enrichment of partitioned heritability, in both magnitude of enrichment and quantitative gradation of CRE bins. Notably, FigR and SCENIC+ also predict FR CREs to recall RPTEC H3K27ac and H3K4me3 peaks more accurately relative to H CREs, supporting our hypothesis that RPTECs may have a partial FR phenotype. Similarly, they also predict FR CREs to be more enriched for CKD heritability than H CREs, which is consistent with our hypothesis as well.

New Supplementary Figure 4. Enrichment of CKD and eGFR partitioned heritability using other CRE-predicting methods. **a.** From left to right, ROC curve calculated for FigR, SCENIC+, Cicero, and scMEGA-predicted healthy-FR PT CREs against RPTEC H3K27ac peaks identified with CUT&RUN. **b.** From left to right, ROC curve calculated for FigR, SCENIC+, Cicero, and scMEGA-predicted healthy-FR PT CREs against RPTEC H3K4me3 peaks identified with CUT&RUN. **c.** AUCs for all tested methods against H3K27ac CUT&RUN peaks (left) and H3K4me3 CUT&RUN peaks (right). scMEGA CREs calculated along H-FR pseudotime trajectory.

New Supplementary Figure 7. Enrichment of CKD and eGFR partitioned heritability using other CRE-predicting methods. **a.** Enrichment of partitioned heritability of CKD in model-predicted healthy (PCT + PST) and FR (failed repair—KIM1+ PT) CREs by FigR, SCENIC+, Cicero, and scMEGA. **b.** Enrichment of partitioned heritability of eGFR in model-predicted healthy (PCT + PST) and FR (failed repair—KIM1+ PT) CREs by FigR, SCENIC+, Cicero, and scMEGA.

Analysis of detected regulators TFS (Fig 4.) is welcomed, but currently only explores the use of ChromVAR, which only uses snATAC-seq information. This is not a fair comparison. It is not surprising the RENIN predicted scores correlates with TF expression, as this was used as a criterial for filtering TFs for RENIN (see line 220). Most of the methods listed above do combine gene expression in their analysis and would be more suitable here.

We thank the reviewer for this point and agree that it is not a fair comparison. ChromVAR is an incredibly useful tool, especially when working with non-multiomic datasets. Moreover, as stated, it is not necessarily striking or interesting that RENIN predictions correlate with TF expression, because our approach to ranking TFs weights by average expression. The intention with this Figure was to demonstrate that RENIN was not simply replicating either approach, and was using gene expression information on top of motif analysis to yield new predictions (Figure 4D) . We agree that this Figure should not be presented as a central benchmarking finding and have deemphasized this Figure, moving it to **New Supplementary Figure 9** and clarifying that this should not be interpreted as a validation of our method. As recommended above, we have added TF benchmarking with the other mentioned methods. We first calculate the fraction of NFAT5 target genes predicted by each model that are bound by NFAT5, measured by NFAT5 CUT&RUN-seq performed on RPTECs. We find the methods have similar accuracies, but RENIN recovers more targets (RENIN – 379/445, Pando – 104/116, CellOracle – 45/56,

SCENIC+ 274/305, FigR – 31/40). We performed CUT&RUN-seq for another top predicted FR TF, HIVEP2, and did the same comparison. For HIVEP2, only RENIN and Pando predicted any targets (RENIN – 368/439, Pando – 70/85, CellOracle – 0/0, SCENIC+ 0/0, FigR 0/0).

Figure 5C. Number of predicted NFAT5 targets by each method, separated into target genes that were bound versus unbound on NFAT5 CUT&RUN-seq. 379/445 RENIN-predicted targets, 104/116 Pando-predicted targets, 45/56 CellOracle-predicted targets, 274/305 SCENIC+-predicted targets, and 31/40 FigR-predicted targets were bound by NFAT5 assessed by CUT&RUN-seq on RPTEC culture.

Supplementary Figure 22B. Number of predicted HIVEP2 targets by each method, separated into target genes that were bound versus unbound on HIVEP2 CUT&RUN-seq. 368/439 RENIN-predicted targets and 70/85 Pando-predicted targets were bound by HIVEP2 assessed by CUT&RUN-seq on RPTEC culture. SCENIC+ does not predict any H-FR genes to be HIVEP2 targets. CellOracle and FigR do not use motif databases that include HIVEP2 motif necessary for modeling.

For the regression-based methods, we also assessed each method's ability to predict gene expression in an independent dataset (KPMP, n=27 snRNA-seq) after being trained on our multiome dataset. Here we found that RENIN had competitive performance as well, with mean r^2 of .080, relative to .065 for Pando and .016 for CellOracle (**New Figure 4E-G**). It is very likely that different methods will have better or worse performance in different contexts, however, for the focus of our work, the design of our approach appears to be the best option among available methods.

New Figure 4E-G. Benchmarking model TF-gene regulatory predictions . e. r^2 calculated for RENIN- and Pando-predicted H-FR gene expression compared to target gene expression in an independent KPMP snRNA-seq dataset for genes that were successfully modeled by both methods. For shared genes, mean RENIN r^2 was .080 and mean Pando r^2 was .065. **f.** r^2 calculated for RENIN- and CellOracle-predicted H-FR gene expression compared to target gene expression in an independent KPMP snRNA-seq dataset for genes that were successfully modeled by both methods. For shared genes, mean RENIN r^2 was .055 and mean CellOracle r^2 was .016. **g.** Number of H-FR differentially expressed genes modeled by each method.

Line 338-339, the need to perform enrichment analysis on CRE is unclear. RENIN as a model to find gene-TF interactions, so why not focus on the RENIN based predictions described in the next section?

We thank the reviewer for this point and intend for the RENIN-based predictions to be the main focus of the manuscript. For this part we asked whether motif enrichment performed on RENIN-predicted CREs may reveal putative factors that drive their changes in expression, that may subsequently have an effect on gene expression. We believe our model design is suited only to CRE to gene expression prediction, rather than gene expression to CRE prediction, as we do not consider any temporal aspect. Beyond motif enrichment analysis, an approach such as scMEGA that leverages pseudotime may be best here. We have moved this Figure to **New Supplementary Figure 10** in an effort to further focus our manuscript, and add clarifying text of these points as well.

The use of networks methods for analyse the GRNs is interesting. It is also welcomed that authors validate their top predicted gene (NFAT5). I could not find, however, any details on how the insilico perturbation experiments were computed.

We thank the reviewer for their close reading of the manuscript. We have moved the description of the perturbation testing out of the Gene regulatory network modeling section into its own Methods section for clarity.

Reviewer #1 (Remarks to the Author):

Also the authors did extensive work in the revised version, the main results are predictive as reported in many other manuscripts and provide not enough evidence that RENIN is superior compared to other methods, especially in other datasets.

As cis-regulatory elements are identified by ATAC-seq the functionality of the open chromatin regions remains unclear. In fact most of the predicted regulatory elements might not have any function at all. Normally additional experiments are performed to reduce the false positive rate eg. marker of active chromatin (H3K27ac) or closed chromatin, Pol2 and RNA-seq (active transcription), mediator binding, transcription factor binding,... This fact may diminishes the advantage of measuring ATAC- and RNA-seq in the same cell. In silico methods to reduce the false positive rate has not proofed very successful.

The quality of the CUT-RUN experiments remains unclear or rather it seems they are of poor quality? Do you see NFAT5 binding in open chromatin regions? For this the CUT-RUN experiments should be performed in primary human tissue. This would also allow for comparison in RPTECs. This is also true for ATAC-seq only the other way around. (Perform ATAC-seq in RPTECs or any other marker of open/active chromatin). The chromatin landscape can be quite different between in-vivo and in-vitro systems. Therefore, RPTECs might not a good model to validate drivers of failed repair important in the human kidney in-vivo.

NFAT5 is also expressed in VCAM1 negative, LTL negative tubules - expression in non-failed repair, non proximal tubules (Suppl. Fig. 24) What is the role of NFAT5 in these tubules. Authors should discuss.

I am still not convinced that RPTECs are the best model for failed repair. Most of the methods can be performed in human kidney tissue, authors should do that.

Reviewer #2 (Remarks to the Author):

I am satisfied with the answer to my concerns. The paper is much improved and merits publication in N.Comms.

1) Authors have argued somewhat convincingly that pseudobulk aggregation reduce noise and compared against aggregation with 1, 5 and 10 cells (S.F.12). Not sure what the jacquard similarity metrics represent in terms of inferential results. Are the results with 1 cell more accurate than with 10 ?

2) The authors have made the work of including additional experiments to validate GRN inference, such as:

- Additional knock-out experiments for CREB5, HIVEP2, PPARA,
- Added Cut&Run experiments for NFAT2,5, HIVEP2
- tried Cut&Run for CREB5 and PPARA but failed.
- together with previous siRNA experiments

and used to benchmark model's predictions against Pando, CellOracle, Scenic+ and FigR. Results on new figure 5c and S.F 22b shows credible improvements in their methodology compared to competing approaches. Their results seem convincing and other reviewers commented positively on the analysis of the data sets and the significance for kidney biology.

3) They also addressed my comments throughout the text, deemphasize many of the claims when needed and modified the title to emphasize their focus on kidney biology.

4) To address another reviewer, they have added co-staining experiments showing that NFAT5 is associated with the repair state.

5) Additionally, they have changed the focus from cell type specificity to H-FR transtion and added reanalysis of additional human data sets.

6) I like Figure 2 depicting some of the math and the link between CREs and genes.

Reviewer #3 (Remarks to the Author):

I am mostly satisfied with the revision of the manuscript describing RENIN. The comparison of RENIN with additional methods for regulatory network inference reinforces the relevance of the proposed method. A few points still require some additional clarifications.

1. I was not able to find the example data to run RENIN. Authors should make its location explicit. Also, they should provide R objects (of complete data) at zenodo.

2. The finding of a small overlap between the CUT& RUN and the sentence "a finding that has been detailed before in the context of gene regulatory network modeling and may demonstrate cell type- specific TF activity⁶⁷" are puzzling. Authors should expand and explain this analysis with more details. How were the NFAT5 motifs exactly defined? Aren't these findings cell specific? It would be crucial to further clarify this observation.

Minor Points:

It is unclear why most of the novel analysis went to the supplement. Some of the interesting results (graph with AUCs of all methods in Sup. 4c), are a main part of the benchmarking and could be shown in figure 3.

The reference to panel 3E (page 10) seems wrong. This possibly refer to 3H.

Reviewer #1

Also the authors did extensive work in the revised version, the main results are predictive as reported in many other manuscripts and provide not enough evidence that RENIN is superior compared to other methods, especially in other datasets.

We agree that benchmarking with other multiomic datasets would be useful. In this second revision we therefore trained Pando, CellOracle, and RENIN on the Gerhardt 2023 mouse multiome dataset in order to predict expression in the Kirita 2020 dataset. As with the results predicting expression in the KPMP dataset, we found that RENIN had superior prediction results relative to both methods. Between RENIN- and Pando-shared genes, the mean RENIN r^2 was .068, while the mean Pando r^2 was 0.42. For RENIN- and CellOracle-modeled genes, the mean RENIN r^2 was .012 and the mean CellOracle r^2 was .0035 (**New Supplementary Figure 13A-B**). RENIN and CellOracle fit models for the majority of H-FR differentially expressed genes, while Pando fit models for just 140 genes (**New Supplementary Figure 13C**). We have updated the manuscript to discuss these benchmarking results as well.

New Supplementary Figure 13A-C. Benchmarking RENIN predictions. **a.** r^2 calculated for RENIN- and Pando-predicted H-FR gene expression by training on Gerhardt 2023 multiomic dataset compared to target gene expression in an independent Kirita 2020 dataset for genes that were successfully modeled by both methods. For shared genes, mean RENIN r^2 was .068 and mean Pando r^2 was .042. **b.** r^2 calculated for RENIN- and CellOracle-predicted H-FR gene expression by training on Gerhardt 2023 multiomic dataset compared to target gene expression in an independent Kirita 2020 dataset for genes that were successfully modeled by both methods. For shared genes, mean RENIN r^2 was .012 and mean CellOracle r^2 was .0035. **c.** Number of H-FR differentially expressed genes modeled by each method by training on Gerhardt 2023 multiomic dataset.

As cis-regulatory elements are identified by ATAC-seq the functionality of the open chromatin regions remains unclear. In fact most of the predicted regulatory elements might not have any function at all. Normally additional experiments are performed to reduce the false positive rate eg. marker of active chromatin (H3K27ac) or closed chromatin, Pol2 and RNA-seq (active transcription), mediator binding, transcription factor binding,... This fact may diminishes the advantage of measuring ATAC- and RNA-seq in the same cell. In silico methods to reduce the false positive rate has not proofed very successful.

We agree that our in silico predictions would benefit validation and have performed a number of additional validation experiments. We have demonstrated that NFAT5 binds to open

chromatin regions in kidney tissue from healthy donors and donors with CKD. We describe our results in the response to the second reviewer critique.

Statistical inference of gene regulatory networks will benefit from detailed single cell profiling of epigenetic factors such as histone modifications. However present multimodal single cell technologies do not allow for all of these factors to be measured simultaneously. Alternatively, we could use bulk measurements, but this approach has some downsides as well. For example, bulk measurements may suffer from false discovery due to technical factors, sequencing depth, and transience of binding events. Most importantly, bulk measurements lack the advantages of single cell resolution, including individual cell datapoints that allow us to infer peak-gene connections and cell type specificity. Moreover, the FR population is a small minority of the cells in the kidney and would be underrepresented in a bulk analysis. For these reasons, it would be difficult to retain cell type and state resolution when studying FR development in bulk human kidney tissue. We believe that simultaneous RNA and ATAC-seq profiling are two measurements that are useful for CRE identification and have included some additional follow-up studies to further reduce false discovery.

We would like to highlight how some of the experiments we have performed validate our methods. For example, Figure 3G demonstrates that increasing CRE scores have increased partitioned heritability for CKD risk variants. As the vast majority of risk variants are non-coding and therefore likely modify disease risk through effects on transcription, the fact that H-FR CREs identified with our method are strongly enriched for CKD heritability demonstrates that our method has identified relevant and active CREs with some degree of false positive rate reduction. Figure 3A-F and Supplementary Figure 4 demonstrate how identified cis-regulatory elements with increasing scores enrich for histone modifications associated with active transcriptional regulation, H3K27ac and H3K4me3 in RPTECs. This demonstrates that we have augmented simple ATAC-seq data with a metric that correlates with some of the mentioned markers of active transcriptional regulation and achieved some success at reduction of false positive rates with ATAC-seq data alone. This approach can therefore capture some of the regulatory information that would be provided by simultaneous single cell histone modification and RNA-seq profiling.

This analysis also demonstrated that some FR CREs and associated gene regulatory mechanisms are active in RPTECs, as demonstrated by their overlap with histone modifications, H3K27ac and H3K4me3. Furthermore, FR CREs had improved recall of these histone modifications relative to H CREs. This improved recall of RPTEC H3K27ac and H3K4me3 peaks with predicted FR CREs relative to H CREs is demonstrated by other similar multiomic multivariable regression methods, SCENIC+, F1gR, and DIRECT-NET (to a small extent). This suggests that the differential recall is not a quirk of our methodology and is rather more likely to be due to the presence of active FR gene regulatory mechanisms in RPTEC cell culture. While we do not think RPTECs are FR PT cells, we do think it provides evidence that RPTEC cell culture contains a heterogeneous mix of cells and/or regulatory processes, which we have clarified in the text.

In our model, we reduce the false positive rate by only retaining ATAC peaks with accessibility correlated to expression of linked H-FR genes. As ATAC accessibility and other epigenetic marks all suffer from false positivity, we view this accessibility-expression regression approach as a critical part of reducing false discovery. The estimator we used also incorporates measures to increase sparsity and reduce false discovery, which can arise from the single cell resolution of the ATAC and RNA-seq dataset. Additionally, our regression approach accounts for ATAC accessibility scores as a metric that encompasses factors such as histone modifications

and other factors that modify accessibility of an enhancer. We also think an important advantage of our approach, along with other similar regression strategies, is that the regression analysis provides a new data layer that allows linking of putative regulatory elements to target genes. This regression approach also allows the more effective use of single cell RNA-seq data to validate distal regulatory elements that are not, for example, easily linked to a target gene because they lie within that gene's promoter. Therefore, we think a major advantage of this approach, given the important points made, is that it allows the use of single cell RNA-seq to effectively reduce false positive rates of cis-regulatory elements prediction with single cell ATAC-seq data alone. By leveraging these advantages with simultaneous profiling of single cell ATAC- and RNA-seq, we achieved competitive results in four independent tests, using independent datasets for benchmarking, relative to other single cell methods.

We have clarified the text to add the limitations of in vitro systems while explaining the rationale of wanting to preserve cell type specificity with this single cell approach instead of relying on bulk profiling of all the mentioned epigenetic markers. We also eagerly look forward to the development of single cell approaches to assay multiple epigenomic profiles simultaneously with RNA expression and clarify that these will help reduce false positive rates of CRE prediction. We also discuss how, without these single cell epigenomic technologies, a major advantage of our computational approach is that we can reduce the false positive rate of ATAC-seq-guided CRE prediction by correlating accessibility with gene expression. This allows us to preserve the several important benefits that single cell sequencing provide, particularly for the cellularly complex kidney.

The quality of the CUT-RUN experiments remains unclear or rather it seems they are of poor quality? Do you see NFAT5 binding in open chromatin regions?

We think this comment may be referring to the bound motif analysis in the RPTEC CUT&RUN data. Motifs were identified with Signac's motifmatchr-based function using the chromVAR motifs dataset, and motif-containing peaks were identified in our multiome dataset. 3,945 of 17,499 NFAT5-containing peaks (22.5%) are bound by NFAT5 in our RPTEC CUT&RUN data. There are 1,543 peaks with both increased accessibility in FR PT cells (FR PT DARs) and at least one NFAT5 motif; 795 of these peaks are bound by NFAT5 (51.5%). A similar enrichment is found for HIVEP2 motifs: 2,664 of 12,759 HIVEP2 motif-containing peaks (20.9%) compared to 697 of 1,563 HIVEP2 motif-containing FR PT DARs (44.6%). These findings suggest that NFAT5 is binding open chromatin regions identified in our multiome dataset and that cell type specificity is a major important contributor to the finding that not all computational motifs are bound by NFAT5 or HIVEP2 in vitro (**New Supplementary Figure 24**). These findings also demonstrate that current PWM representations do not include all factors that contribute to TF binding, such as the presence of other cofactors or more complex representations of DNA binding sequences. As all multiome algorithms rely on computational motif representations, we expect that future modeling algorithms will benefit from improvements in motif representation. We have clarified the text describing this finding.

New Supplementary Figure 24. NFAT5 and HIVEP2 motif-containing peaks bound in RPTEC CUT&RUN. a. 3,945 of 17,499 NFAT5 motif-containing peaks (22.5%) are bound by NFAT5. 795 of 1,543 NFAT5 motif-containing FR PT DARs (51.5%) are bound by NFAT5. **b.** 2,664 of 12,759 HIVEP2 motif-containing peaks (20.9%) are bound by HIVEP2. 697 of 1,563 HIVEP2 motif-containing FR PT DARs (44.6%) are bound by HIVEP2.

We also did more analysis of our MACS2-called peaks in our NFAT5 CUT&RUN dataset. We confirmed that well-studied NFAT5 targets, *TauT/SLC6A6*, *SMIT/SLC5A3*, *AR/AKR1B1*, and *RNF183*, are all bound by NFAT5 with much higher signal than that of the IgG control (**Figure R1A-D**). MACS2-predicted NFAT5 binding of these known target genes supports the quality of the RPTEC CUT&RUN experiments.

Figure R1 (for review only). RPTEC NFAT5 CUT&RUN. Visualization with IGV browser of NFAT5 CUT&RUN signal, IgG CUT&RUN signal, and MACS2 called peaks for (a) AKR1B1 (b) RNF183 (c) SLC5A3 and (d) SLC6A6.

For this the CUT-RUN experiments should be performed in primary human tissue. This would also allow for comparison in RPTECs. This is also true for ATAC-seq only the other way around. (Perform ATAC-seq in RPTECs or any other marker of open/active chromatin). The chromatin landscape can be quite different between in-vivo and in-vitro systems. Therefore, RPTECs might not a good model to validate drivers of failed repair important in the human kidney in-vivo. NFAT5 is also expressed in VCAM1 negative, LTL negative tubules - expression in non-failed repair, non proximal tubules (Suppl. Fig. 24) What is the role of NFAT5 in these tubules. Authors should discuss. I am still not convinced that RPTECs are the best model for failed repair. Most of the methods can be performed in human kidney tissue, authors should do that.

We thank the reviewer for this feedback. We have now performed new experiments as suggested by the reviewer (see below). We have added discussion of the role of NFAT5 in the response to osmotic stress in these tubules. Additionally we have added new analysis and data along the reviewer's suggestions.

First, we performed NFAT5 CUT&RUN in primary healthy and CKD human kidney samples as suggested by the reviewer. We calculated the enrichment of FR PT features—genes, chromatin peaks, and predicted NFAT5 motifs—in NFAT5 CUT&RUN peaks relative to their representation in the whole dataset and NFAT5 peaks were significantly enriched in FR CRE. Chromatin peaks with increased accessibility in KIM1+ PT cells (FR differentially accessible regions, or FR PT DARs) represent 10,082 of the 193,787 peaks (5.2%) in our dataset. 437 of 1845 NFAT5-bound peaks that overlapped with our dataset peaks (23.7%, $p < 2.2e-16$ by Fisher's exact test) were FR PT DARs, again representing significant enrichment. Finally, there are 17,499 peaks in our multiome dataset with computationally predicted NFAT5 motifs. Of these, 1,543 lie within FR PT DARs (8.8%). 824 NFAT5 motif-containing peaks overlapped with NFAT5 CUT&RUN binding, and 198 of these were FR-PT DARs (24%, $p < 2.2e-16$ by Fisher's exact test). Given NFAT5's role in different distal tubular cell types it is likely that many of the remaining NFAT5-bound genes are not involved in the H-FR transition. Of 1070 genes bound by NFAT5, 237 were in the H-FR DEG list (22.2%, $p < 2.2e-16$ by Fisher's exact test), representing significant enrichment (**New Supplementary Figure 26A**). Therefore, although NFAT5 is involved in many processes in the different renal cell types, a significant portion of its binding activity and regulatory function is represented in H-FR regulatory processes, confirming its role in failed repair. We did not think that H3K27ac or H3K4me3 CUT&RUN in primary human kidney would be as helpful, because this would label regulatory elements for all genes expressed in the kidney, rather than those that are specifically involved in failed repair or are bound by NFAT5.

Second, we compared snATAC-seq PT and RPTEC chromatin accessibility profiles and demonstrate that they have significant overlap. We removed the bottom 25% of accessible peaks in the PCT, PST, and KIM1+ PT clusters to derive a whole kidney proximal tubule chromatin accessibility profile. We compared overlap with the H3K27ac and H3K4me3 data generated for this manuscript and RPTEC ATAC-seq data (Ref 11). In **New Supplementary Figure 4A**, 82,414 of 140,850 (58.5%) RPTEC ATAC peaks overlapped accessible chromatin in whole kidney PT accessible peaks. Additionally, 26,199 of 41,599 H3K27ac peaks (63.0%) and 18,547 of 21,407 H3K4me3 peaks (86.6%) overlapped accessible chromatin. The majority of assayed RPTEC chromatin peaks overlapping whole kidney accessible chromatin supports the use of RPTECs as an in vitro model of proximal tubule. Because these findings do suggest the presence of accessible chromatin and epigenetic modifications that are not identified in human kidney tissue, we also assessed overlap of snATAC-seq peaks in RPTEC ATAC peaks. We identified 18,492 healthy PT differentially accessible regions (DARs) as ATAC-seq peaks with increased accessibility in PCT and/or PST cells relative to other cell types using Seurat's FindMarkers function. Similarly, there were 10,082 KIM1+ PT DARs. Comparing these peak sets to RPTEC ATAC-seq data, we found 75388 of 145340 PT peaks (51.9%), 13467 of 18492 healthy PT DARs (72.8%), and 9179 of 10082 KIM1+ PT peaks (91.0%) overlapped RPTEC accessible chromatin (**New Supplementary Figure 4B**). While we do not have enough data to suggest nor do we think that RPTECs are failed repair cells, the increased representation of KIM1+ PT peaks relative to healthy PT peaks in RPTEC ATAC-seq data suggests that RPTEC culture is heterogeneous with both healthy and FR cells and/or gene regulatory mechanisms present. We therefore think this allows us to study the effects of predicted TFs by assessing whether components of FR gene regulatory mechanisms are affected, such as VCAM1 downregulation with NFAT5 siRNA knockdown. While there are many differences between in

vivo and in vitro models, given the preserved FR-associated gene regulatory elements in RPTECs, we thought that enriching for PT-specific signals outweighed the benefits of pursuing whole kidney approaches.

New Supplementary Figure 4. Comparison of snATAC-seq and RPTeC chromatin accessibility. **a.** RPTeC accessible peaks overlap with snATAC PT peaks. 82,414 of 140,850 (58.5%) RPTeC ATAC peaks, 26,199 of 41,599 H3K27ac peaks (63.0%), and 18,547 of 21,407 H3K4me3 peaks (86.6%) overlap accessible chromatin in whole kidney PT accessible peaks. **b.** snATAC-seq PT ATAC-seq overlap with RPTeC ATAC-seq. 75388 of 145340 PT peaks (51.9%), 13467 of 18492 healthy PT DARs (72.8%), and 9179 of 10082 KIM1+ PT peaks (91.0%) overlap RPTeC ATAC-seq peaks.

New Supplementary Figure 26. H-FR genes and regulatory elements are overrepresented in NFAT5-bound peaks in whole kidney CUT&RUN. **a.** 1,519 of 18,262 genes (8.3%) are H-FR DEGs. 237 of 1,070 NFAT5-bound genes are H-FR DEGs. **b.** 10,082 of 193,787 (5.2%) peaks have increased accessibility in FR PT cells (FR PT DARs). 437 of 1,845 NFAT5-bound peaks are FR PT DARs (23.7%). **c.** 1,543 of 17,499 NFAT5 motif-containing peaks are FR PT DARs (8.8%). 198 of 824 NFAT5-bound, NFAT5 motif-containing peaks are FR PT DARs (24%). Statistical significance determined by Fisher's exact test, * denotes $p < 2.2e-16$.

To summarize the new analysis added, we have demonstrated that the epigenetic landscape of RPTECs resembles the proximal tubule chromatin accessibility profile derived from our single cell multiome dataset. Furthermore, the FR PT accessibility profile shares more overlap with RPTEC accessibility than the healthy PT accessibility profile does. This suggests the presence of FR PT regulatory mechanisms, allowing us to use this in vitro system to confirm that NFAT5 promotes a failed repair phenotype through siRNA knockdown and CUT&RUN experiments. However, as in vitro experiments may suffer from false positives and other drawbacks compared to in vivo experiments, we also performed NFAT5 CUT&RUN in whole kidney. NFAT5-bound genes, regulatory elements, and motifs were significantly enriched for failed repair elements. These results demonstrate that one of NFAT5's roles in whole kidney is in the development of maladaptive repair.

Reviewer #2

I am satisfied with the answer to my concerns. The paper is much improved and merits publication in N.Comms.

Thank you for these positive comments.

1) Authors have argued somewhat convincingly that pseudobulk aggregation reduce noise and compared against aggregation with 1, 5 and 10 cells (S.F.12). Not sure what the jacquard similarity metrics represent in terms of inferential results. Are the results with 1 cell more accurate than with 10 ?

We thank the reviewer for their thoughts on pseudobulk aggregation. We agree with their initial point that individual cell modeling is best for most accurate inference of individual gene regulatory network composition. However, when summing the effect of a TF across all predicted target genes, rankings of top predicted TFs do not change significantly, which may facilitate faster exploratory analysis or analysis with computational resource limitations. However, we agree with the reviewer that investigators should not use aggregation when possible for simulation studies and inference of individual TF-gene regulatory relationships. We have clarified the text towards this point.

2) The authors have made the work of including additional experiments to validate GRN inference, such as:

- Additional knock-out experiments for CREB5, HIVEP2, PPARA,
- Added Cut&Run experiments for NFAT2,5, HIVEP2
- tried Cut&Run for CREB5 and PPARA but failed.
- together with previous siRNA experiments

and used to benchmark model's predictions against Pando, CellOracle, Scenic+ and FigR. Results on new figure 5c and S.F 22b shows credible improvements in their methodology

compared to competing approaches. Their results seem convincing and other reviewers commented positively on the analysis of the data sets and the significance for kidney biology.

3) They also addressed my comments throughout the text, deemphasize many of the claims when needed and modified the title to emphasize their focus on kidney biology.

4) To address another reviewer, they have added co-staining experiments showing that NFAT5 is associated with the repair state.

5) Additionally, they have changed the focus from cell type specificity to H-FR transition and added reanalysis of additional human data sets.

6) I like Figure 2 depicting some of the math and the link between CREs and genes.

We appreciate the reviewer's careful reading and suggestions to this manuscript.

Reviewer #3

I am mostly satisfied with the revision of the manuscript describing RENIN. The comparison of RENIN with additional methods for regulatory network inference reinforces the relevance of the proposed method. A few points still require some additional clarifications.

1. I was not able to find the example data to run RENIN. Authors should make its location explicit. Also, they should provide R objects (of complete data) at zenodo.

We have uploaded the R object of the multiomic dataset to Zenodo and clarified that it may be found there as well.

2. The finding of a small overlap between the CUT& RUN and the sentence "a finding that has been detailed before in the context of gene regulatory network modeling and may demonstrate cell type- specific TF activity⁶⁷" are puzzling. Authors should expand and explain this analysis with more details. How were the NFAT5 motifs exactly defined? Aren't these findings cell specific? It would be crucial to further clarify this observation.

We thank the reviewer for their careful reading. We wish to convey two contributors to this result: first, the effect of cell type specificity and second, that there may be other cofactor binding requirements that are not yet assayable with current single cell multiomic technology. Motifs were defined with Signac's motifmatchr-based function using the chromVAR motifs dataset. Regarding the first point, 3,945 of 17,499 NFAT5-containing peaks (22.5%) are bound by NFAT5 in our RPTEC CUT&RUN data. There are 1,543 peaks with both increased accessibility in FR PT cells (FR PT DARs) and at least one NFAT5 motif; 795 of these peaks are bound by NFAT5 (51.5%). A similar enrichment is found for HIVEP2 motifs: 2,664 of 12,759 HIVEP2 motif-containing peaks (20.9%) compared to 697 of 1,563 HIVEP2 motif-containing FR PT DARs (44.6%). These findings strongly suggest that cell type specificity is an important contributor to the finding that not all computational motifs are bound by NFAT5 or HIVEP2 in vitro (**New Supplementary Figure 24**). These findings also demonstrate that current PWM representations do not include all factors that contribute to TF binding, such as the presence of other cofactors or more complex representations of DNA binding sequences. Future modeling algorithms may benefit from improvements in motif representation. We have clarified the text describing this finding.

New Supplementary Figure 24. NFAT5 and HIVEP2 motif-containing peaks bound in RPTEC CUT&RUN. a. 3,945 of 17,499 NFAT5 motif-containing peaks (22.5%) are bound by NFAT5. 795 of 1,543 NFAT5 motif-containing FR PT DARs (51.5%) are bound by NFAT5. **b.** 2,664 of 12,759 HIVEP2 motif-containing peaks (20.9%) are bound by HIVEP2. 697 of 1,563 HIVEP2 motif-containing FR PT DARs (44.6%) are bound by HIVEP2.

Minor Points:

It is unclear why most of the novel analysis went to the supplement. Some of the interesting results (graph with AUCs of all methods in Sup. 4c), are a main part of the benchmarking and could be shown in figure 3.

We appreciate the highlighting of interesting results to move to the main body. We have moved Supplementary Figure 4C into **New Figure 3**.

New Figure 3. Cell type CREs identified with RENIN. **a.** ROC curve calculated for RENIN-predicted healthy-FR PT CREs against RPTEC H3K4me3 peaks identified with CUT&RUN. AUCs were 0.696 (FR) and 0.617 (Healthy). Dotted line indicates random performance (AUC = 0.5). **b.** ROC curve calculated for RENIN-predicted healthy-FR PT CREs against RPTEC H3K27ac peaks identified with CUT&RUN. FR (AUC = 0.694, CREs predicted to regulate a marker gene of the KIM1+ cluster; Healthy (AUC = 0.570), CREs predicted to regulate a marker gene of the PCT and/or PST cluster. **c.** ROC curve calculated for LinkPeaks-predicted healthy-FR PT CREs against same RPTEC H3K4me3 peak set as in **(a)**. AUCs were 0.350 (FR) and 0.388 (Healthy). **d.** ROC curve calculated for LinkPeaks-predicted healthy-FR PT CREs against same RPTEC H3K27ac peak set as in **(b)**. AUCs were 0.435 (FR) and 0.422 (Healthy). **e-f.** ROC curve calculated for DIRECT-NET-predicted healthy-FR PT CREs against RPTEC H3K4me3 (AUCs 0.630 and 0.621) **(e)** and H3K27ac (AUCs 0.649 and 0.644) **(f)** CUT&RUN peaks. **g.** AUCs for all tested methods against H3K27ac CUT&RUN peaks (left) and H3K4me3 CUT&RUN peaks (right). Each tested method's quantitative metric was used. FigR CREs were scored by rObs, SCENIC+ CREs were scored by summed R2G_importance_x_abs_rho across all target genes, Cicero CREs were sorted by summed coaccessibility score, and scMEGA CREs were sorted by the TStat metric summed across all target genes. **h.** Comparison of RENIN, LinkPeaks, and DIRECT-NET by enrichment of partitioned heritability of CKD in model-predicted healthy (PCT + PST) and FR (failed repair—KIM1+ PT) CREs. Statistical significance determined by two-tailed t-test of difference between enrichment, $p < 0.05$.

The reference to panel 3E (page 10) seems wrong. This possibly refer to 3H.

We thank the reviewer for their close reading and have made the correction.

Additional Changes

While benchmarking CellOracle, Pando, and RENIN with the Gerhardt 2023 and Kirita 2020 datasets, we realized our implementation of CellOracle was limiting the number of motifs it was able to use. With an adjusted implementation, performance relative to RENIN and Pando did not change, but the change increased the mean r^2 (.016 to .026) and the number of CellOracle-predicted NFAT5 targets, from 56 (45 bound in CUT&RUN) to 67 (53 bound in CUT&RUN). We have updated Figures 4 and 5 in the manuscript and include them below.

New Figure 4. Predictions of key TFs involved in healthy to failed repair PT transition. **a.** TFs sorted by regulatory score, computed as the sum of predicted regulatory coefficients for healthy-FR PT DEGs multiplied by mean TF expression in PT (PCT, PST, KIM1+ PT) clusters. Negative scores indicate FR-promoting TFs—positive regulation of DEGs upregulated in FR PT or negative regulation of DEGs downregulated in FR-PT—and positive scores indicate H-promoting TFs—positive regulation of DEGs upregulated in H PT (PCT and PST) or negative regulation of DEGs downregulated in H PT. **b.** Graph visualization of gene regulatory networks predicted by RENIN. TF node size represents centrality of TFs computed by PageRank, top 20 TFs are labeled. **c.** Top 25 TFs ranked by betweenness. **d.** Top 25 TFs ranked by PageRank. **e.** r^2 calculated for RENIN- and Pando-predicted H-FR gene expression compared to target gene expression in an independent KPMP snRNA-seq dataset for genes that were successfully modeled by both methods. For shared genes, mean RENIN r^2 was .080 and mean Pando r^2 was .065. **f.** r^2 calculated for RENIN- and CellOracle-predicted H-FR gene expression compared to target gene expression in an independent KPMP snRNA-seq dataset for genes that were successfully modeled by both methods. For shared genes, mean RENIN r^2 was .055 and mean CellOracle r^2 was .026. **g.** Number of H-FR differentially expressed genes modeled by each method.

New Figure 5. NFAT5 promotes FR expression phenotype in cultured RPTECs. **a.** Expression of *NFAT5* and *VCAM1* in cultured RPTECs treated with non-targeting (NT) or *NFAT5*-targeting siRNA. RNA levels measured by RT-qPCR and normalized to *GAPDH* expression. *NFAT5* siRNA-treated cells had 22% of the *NFAT5* RNA and 57% of the *VCAM1* RNA levels of non-targeting-siRNA-treated cells (*p* values of 0.018 and 0.013, respectively, unpaired 2-tailed *t*-test). **b.** Heatmap of select differentially expressed genes by RNA-seq in NT and *NFAT5* siRNA-treated RPTECs. **c.** Number of predicted *NFAT5* targets by each method, separated into target genes that were bound versus unbound on *NFAT5* CUT&RUN-seq. 379/445 *RENIN*-predicted targets, 104/116 *Pando*-predicted targets, 53/67 *CellOracle*-predicted targets, 274/305 *SCENIC+*-predicted targets, and 31/40 *FigR*-predicted targets were bound by *NFAT5* assessed by CUT&RUN-seq on RPTEC culture. **d.** Distal predicted CRE for *TPM1*, predicted to be *NFAT5* target *FR* gene and downregulated with siRNA *NFAT5* knockdown, bound by *NFAT5*. **e.** Immunofluorescent labeling of *NFAT5* in adult human kidney. DAPI is a nucleus marker and LTL is an apical proximal tubule marker. * denotes examples of tubules with low LTL intensity. Scale bars are 50 μ m in length.

Reviewer #1 (Remarks to the Author):

I thank the authors for addressing my comments. The manuscript is much improved and I want to congratulate the authors.

I would change the title to indicate the focus on failed repair in kidney epithelia cells.

eg.

Predicting drivers of failed repair in kidney epithelia cells through regularized regression analysis of single cell multiomic sequencing

Reviewer #2 (Remarks to the Author):

I am satisfied with the current revision of the manuscript.

I commend the authors for the recognition of their mistake while benchmarking CellOracle, Pando, and RENIN and the change in R^2 and their amendment.

Reviewer #3 (Remarks to the Author):

I am happy with the authors response and the current version of the manuscript

Reviewer #1 (Remarks to the Author):

I thank the authors for addressing my comments. The manuscript is much improved and I want to congratulate the authors.

I would change the title to indicate the focus on failed repair in kidney epithelia cells.

eg.

Predicting drivers of failed repair in kidney epithelia cells through regularized regression analysis of single cell multiomic sequencing

We've amended the title to better indicate the focus of our manuscript as suggested. In order to meet the 15 word limit, we have changed it to "Predicting proximal tubule failed repair drivers through regularized regression analysis of single cell multiomic sequencing." We thank the reviewer for all their helpful suggestions.

Reviewer #2 (Remarks to the Author):

I am satisfied with the current revision of the manuscript.

I commend the authors for the recognition of their mistake while benchmarking CellOracle, Pando, and RENIN and the change in R^2 and their amendment.

We thank the reviewer for all their helpful suggestions.

Reviewer #3 (Remarks to the Author):

I am happy with the authors response and the current version of the manuscript.

We thank the reviewer for all their helpful suggestions.